# Randomness Helps Rigor: A Probabilistic Learning Rate Scheduler Bridging Theory and Deep Learning Practice

## Abstract

Learning rate schedulers have shown great success in speeding up the convergence of learning algorithms in practice. However, their convergence to a minimum has not been proven theoretically. This difficulty mainly arises from the fact that, while traditional convergence analysis prescribes to monotonically decreasing (or constant) learning rates, schedulers opt for rates that often increase and decrease through the training epochs. In this work, we aim to bridge the gap by proposing a probabilistic learning rate scheduler (PLRS) that does not conform to the monotonically decreasing condition, with provable convergence guarantees. To cement the relevance and utility of our work in modern day applications, we show experimental results on deep neural network architectures such as ResNet, WRN, VGG, and DenseNet on CIFAR-10, CIFAR-100, and Tiny ImageNet datasets. We show that PLRS performs as well as or better than existing state-of-the-art learning rate schedulers in terms of convergence as well as accuracy. For example, while training ResNet-110 on the CIFAR-100 dataset, we outperform the state-of-the-art knee scheduler by $1.56\%$ in terms of classification accuracy. Furthermore, on the Tiny ImageNet dataset using ResNet-50 architecture, we show a significantly more stable convergence than the cosine scheduler and a better classification accuracy than the existing schedulers.

## 1 Introduction

Over the last two decades, there has been an increased interest in analyzing the convergence of gradient descent-based algorithms. This can be majorly attributed to their extensive use in the training of neural networks and their numerous derivatives. Stochastic Gradient Descent (SGD) and their adaptive variants such as Adagrad [8], Adadelta [31], and Adam [17] have been the choice of optimization algorithms for most machine learning practitioners, primarily due to their ability to process enormous amounts of data in batches. Even with the introduction of adaptive optimization techniques that use a default learning rate, the use of stochastic gradient descent with a tuned learning rate was quite prevalent, mainly due to its generalization properties [34]. However, tuning the learning rate of the network can be computationally intensive and time consuming.

Various methods to efficiently choose the learning rate without excessive tuning have been explored. One of the initial successes in this domain is the random search method [3]; here, a learning rate is randomly selected from a specified interval across multiple trials, and the best performing learning rate is ultimately chosen. Following this, more advanced methods such as Sequential Model-Based Optimization (SMBO) [4] for the choice of learning rate became prevalent in practice. SMBO represents a significant advancement over random search by tracking the effectiveness of learning rates from previous trials and using this information to build a model that suggests the next optimal

learning rate. A tuning method for shallow neural networks based on theoretical computation of the Hessian Lipschitz constant was proposed by Tholeti et al. [27].

Several works on training deep neural networks prescribed the use of a decaying Learning Rate (LR)[1] scheduler [10, 32, 26]. Recently, much attention has been paid to cyclically varying learning rates [24]. By varying learning rates in a triangular schedule within a predetermined range of values, the authors hypothesize that the optimal learning rate lies within the chosen range, and the periodic high learning rate helps escape saddle points. Although no theoretical backing has been provided, it was shown to be a valid hypothesis owing to the presence of many saddle points in a typical high dimensional learning task [6]. Many variants of the cyclic LR scheduler have henceforth been used in various machine learning tasks [12, 7, 1]. A cosine-based cyclic LR scheduler proposed by Loshchilov et al. [21] has also found several applications, including Transformers [30, 5]. Following the success of the cyclic LR schedulers, a one-cycle LR scheduler proposed by Smith et al. [25] has been observed to provide faster convergence empirically; this was attributed to the injection of 'good noise' by higher learning rates which helps in convergence. Although empirical validation and intuitions were provided to support the working of these LR schedulers, a theoretical convergence guarantee has not been provided to the best of our knowledge.

There is extensive research on the convergence behavior of perturbed SGD methods, where noise is added to the gradient during updates. In Jin et al. [15], the vanilla gradient descent is perturbed by samples from a ball whose radius is fixed using the optimization function-specific constants. They show escape from a saddle point by characterizing the distribution around a perturbed iterate as uniformly distributed over a perturbation ball along which the region corresponding to being stuck at a saddle point is shown to be very small. In Ge at al. [9], the saddle point escape for a perturbed stochastic gradient descent is proved using the second-order Taylor approximation of the optimization function, where the perturbation is applied from a unit ball to the stochastic gradient descent update. Following Ge at al. [9], several works prove the convergence of noisy stochastic gradient descent in the additive noise setting [33, 16, 2, 28]. In contrast to the above works which operate in the *additive* noise setting, our proposed LR scheduler results in *multiplicative* noise. Analyzing the convergence behavior under the new multiplicative noise setting is fairly challenging and results in a non-trivial addition to the literature.

## 1.1 Motivation

Traditional convergence analysis of gradient descent algorithms and its variants requires the use of a constant or a decaying learning rate [22]. However, with the introduction of LR schedulers, the learning rates are no longer monotonically decreasing. Rather, their values heavily fluctuate, with the occasional use of very large learning rates. Although there are ample justifications provided for the success of such methods, there are no theoretical results which prove that stochastic gradient descent algorithms with fluctuating learning rates converge to a local minimum in a non-convex setting. With the increase of emphasis on trustworthy artificial intelligence, we believe that it is important to no longer treat optimization algorithms as black-box models, and instead provide provable convergence guarantees while deviating from the proven classical implementation of the descent algorithms. In this work, we aim to bridge the gap by providing rigorous mathematical proof for the convergence of our proposed probabilistic LR scheduler with SGD.

## 1.2 Our contributions

1. We propose a new Probabilistic Learning Rate Scheduler (PLRS) where we model the learning rate as an instance of a random noise distribution.

2. We provide convergence proofs to show that SGD with our proposed PLRS converges to a local minimum in Section 4. To the best of our knowledge, we are the first to theoretically prove convergence of SGD with a LR scheduler that does not conform to constant or monotonically decreasing rates. We show how our LR scheduler, in combination with inherent SGD noise, speeds up convergence by escaping saddle points.

3. Our proposed probabilistic LR scheduler, while provably convergent, can be seamlessly ported into practice without the knowledge of theoretical constants (like gradient and Hessian-Lipschitz constants). We illustrate the efficacy of the PLRS through extensive

---

[1]We abbreviate learning rate only in the context of learning rate scheduler as LR scheduler.

experimental validation, where we compare the accuracies with state-of-the-art schedulers in Section 5. We show that the proposed method outperforms popular schedulers such as cosine annealing [21], one-cycle [25], knee [14] and the multi-step scheduler when used with ResNet-110 on CIFAR-100, VGG-16 on CIFAR-10 and ResNet-50 on Tiny ImageNet, while displaying comparable performances on other architectures like DenseNet-40-12 and WRN-28-10 when trained on CIFAR-10 and CIFAR-100 datasets respectively. We also observe lesser spikes in the training loss across epochs which leads to a faster and more stable convergence. We provide our base code with all the hyperparameters for reproducibility in the supplemental material.

## 2 Probabilistic learning rate scheduler

Let $f : \mathbb{R}^d \to \mathbb{R}$ be the function to be minimized. The unconstrained optimization, $\min_{\mathbf{x} \in \mathbb{R}^d} f(\mathbf{x})$, can be solved iteratively using stochastic gradient descent whose update equation at time step $t$ is given by

$$\mathbf{x}_{t+1} = \mathbf{x}_t - \eta_{t+1} g(\mathbf{x}_t). \tag{1}$$

Here, $\eta_{t+1} \in \mathbb{R}$ is the learning rate and $g(\mathbf{x}_t)$ is the stochastic gradient of $f(\mathbf{x})$ at time $t$. In this work, we propose a new LR scheduler, in which the learning rate $\eta_{t+1}$ is sampled from a uniform random variable,

$$\eta_{t+1} \sim \mathcal{U}[L_{min}, L_{max}], \quad 0 < L_{min} < L_{max} < 1. \tag{2}$$

Note that contrary to existing LR schedulers, which are deterministic functions, we propose that the learning rate at each time instant be a realization of a uniformly distributed random variable. Although the learning rate in our method is not scheduled, but is rather chosen as a random sample at every time step, we call our proposed method Probabilistic LR scheduler to keep in tune with the body of literature on LR schedulers. In order to represent our method in the conventional form of the stochastic gradient descent update, we split the learning rate $\eta_{t+1}$ into a constant learning rate $\eta_c$ and a random component, as $\eta_{t+1} = \eta_c + u_{t+1}$, where $u_{t+1} \sim \mathcal{U}[L_{min} - \eta_c, L_{max} - \eta_c]$. The stochastic gradient descent update using the proposed PLRS (referred to as SGD-PLRS) takes the form

$$\mathbf{x}_{t+1} = \mathbf{x}_t - (\eta_c + u_{t+1}) g(\mathbf{x}_t) = \mathbf{x}_t - \eta_c \nabla f(\mathbf{x}_t) - \mathbf{w}_t, \tag{3}$$

where we define $\mathbf{w}_t$ as

$$\mathbf{w}_t = \eta_c g(\mathbf{x}_t) - \eta_c \nabla f(\mathbf{x}_t) + u_{t+1} g(\mathbf{x}_t). \tag{4}$$

Here, $\nabla f(\mathbf{x}_t)$ refers to the true gradient, i.e., $\nabla f(\mathbf{x}_t) = \mathbb{E}[g(\mathbf{x}_t)]$. Note that in (3), the term $\mathbf{x}_t - \eta_c \nabla f(\mathbf{x}_t)$ resembles the vanilla gradient descent update and $\mathbf{w}_t$ encompasses the noise in the update; the noise is inclusive of both the randomness due to the stochastic gradient as well as the randomness from the proposed LR scheduler. We set $\eta_c = \frac{L_{min} + L_{max}}{2}$ so that the noise $\mathbf{w}_t$ is zero mean, which we prove later in Lemma 1.

**Remark 1.** *Note that a periodic LR scheduler such as triangular, or cosine annealing based scheduler can be considered as a single instance of our proposed PLRS. The range of values assigned to the learning rate $\eta_{t+1}$ is pre-determined in both cases. In fact, for any LR scheduler, the basic mechanism is to vary the learning rate between a low and a high value - the high learning rates help escape the saddle point by perturbing the iterate, whereas the low values help in convergence. This pattern of switching between high and low values can be achieved through both stochastic and deterministic mechanisms. While the current literature explores the deterministic route (without providing analysis), we propose and explore the stochastic variant here and also provide a detailed analysis.*

## 3 Preliminaries and definitions

We denote the Hessian of a function $f : \mathbb{R}^d \to \mathbb{R}$ at $\mathbf{x} \in \mathbb{R}^d$ as $\mathbf{H}(\mathbf{x}) := \nabla^2 f(\mathbf{x})$ and the minimum eigenvalue of the Hessian as $\lambda_{min}(\mathbf{H}(\mathbf{x})) := \lambda_{min}(\nabla^2 f(\mathbf{x}))$ respectively.

**Definition 1.** *A function $f : \mathbb{R}^d \to \mathbb{R}$ is said to be $\beta$-smooth (also referred to as $\beta$-gradient Lipschitz) if, $\exists \beta \geq 0$ such that,*

$$\|\nabla f(\mathbf{x}) - \nabla f(\mathbf{y})\| \leq \beta \|\mathbf{x} - \mathbf{y}\|, \quad \forall \mathbf{x}, \mathbf{y} \in \mathbb{R}^d. \tag{5}$$

**Definition 2.** *A function $f : \mathbb{R}^d \to \mathbb{R}$ is said to be $\rho$-Hessian Lipschitz if, $\exists \rho \geq 0$ such that,*

$$\|\boldsymbol{H}(\boldsymbol{x}) - \boldsymbol{H}(\boldsymbol{y})\| \leq \rho \|\mathbf{x} - \mathbf{y}\|, \quad \forall \mathbf{x}, \mathbf{y} \in \mathbb{R}^d. \tag{6}$$

Informally, a function is said to be gradient/Hessian Lipschitz, if the rate of change of the gradient/Hessian with respect to its input is bounded by a constant, i.e., the gradient/Hessian will not change rapidly. We now proceed to define approximate first and second-order stationary points of a given function $f$.

**Definition 3.** *For a function $f : \mathbb{R}^d \to \mathbb{R}$ that is differentiable, we say $\mathbf{x} \in \mathbb{R}^d$ is a $\nu$- first-order stationary point ($\nu$-FOSP), if for a small positive value of $\nu$, $\|\nabla f(\mathbf{x})\| \leq \nu$.*

Before we define an $\epsilon$-second order stationary point, we define a saddle point.

**Definition 4.** *For a $\rho$-Hessian Lipschitz function $f : \mathbb{R}^d \to \mathbb{R}$ that is twice differentiable, we say $\boldsymbol{x} \in \mathbb{R}^d$ is a saddle point if,*

$$\|\nabla f(\mathbf{x})\| \leq \nu \quad and \quad \lambda_{min}(\boldsymbol{H}(\boldsymbol{x})) \leq -\gamma,$$

*where $\nu, \gamma > 0$ are arbitrary constants.*

For a convex function, it is sufficient if the algorithm is shown to converge to the $\nu$-FOSP as it would be the global minimum. However, in the case of a non-convex function, a point satisfying the condition for a $\nu$-FOSP may not necessarily be a local minimum, but could be a saddle point or a local maximum. Hence, the Hessian of the function is required to classify it as a second-order stationary point, as defined below. Note that, in our analysis, we prove convergence of SGD-PLRS to the approximate second-order stationary point.

**Definition 5.** *For a $\rho$-Hessian Lipschitz function $f : \mathbb{R}^d \to \mathbb{R}$ that is twice differentiable, we say $\mathbf{x} \in \mathbb{R}^d$ is a $\nu$-second-order stationary point ($\nu$-SOSP) if,*

$$\|\nabla f(\mathbf{x})\| \leq \nu \quad and \quad \lambda_{min}(\boldsymbol{H}(\boldsymbol{x})) \geq -\gamma, \tag{7}$$

*where $\nu, \gamma > 0$ are arbitrary constants.*

**Definition 6.** *A function $f : \mathbb{R}^d \to \mathbb{R}$ is said to possess the strict saddle property at all $\boldsymbol{x} \in \mathbb{R}^d$ if $\boldsymbol{x}$ fulfills any one of the following conditions: (i) $\|\nabla f(\mathbf{x})\| \geq \nu$, (ii) $\lambda_{min}(\boldsymbol{H}(\boldsymbol{x})) \leq -\gamma$, (iii) $\mathbf{x}$ is close to a local minimum.*

The strict saddle property ensures that an iterate stuck at a saddle point has a direction of escape.

**Definition 7.** *A function $f : \mathbb{R}^d \to \mathbb{R}$ is $\alpha-$strongly convex if $\lambda_{min}(\boldsymbol{H}(\boldsymbol{x})) \geq \alpha \quad \forall \mathbf{x} \in \mathbb{R}^d$.*

We now provide the formal definitions of two common terms in time complexity.

**Definition 8.** *A function $f(s)$ is said to be $O(g(s))$ if $\exists$ a constant $c > 0$ such that $|f(s)| \leq c|g(s)|$. Here $s \in S$ which is the domain of the functions $f$ and $g$.*

**Definition 9.** *A function $f(s)$ is said to be $\Omega(g(s))$ if $\exists$ a constant $c > 0$ such that $|f(s)| \geq c|g(s)|$.*

In our analysis, we introduce the notations $\tilde{O}(.)$ and $\tilde{\Omega}(.)$ which hide all factors (including $\beta$, $\rho$, $d$, and $\alpha$) except $\eta_c$, $L_{min}$ and $L_{max}$ in $O$ and $\Omega$ respectively.

# 4 Proof of convergence

We present our convergence proofs to theoretically show that the proposed PLRS method converges to a $\nu$-SOSP in finite time. We first state the assumptions that are instrumental for our proofs.

**Assumptions 1.** *We now state the assumptions regarding the function $f : \mathbb{R}^d \to \mathbb{R}$ that we require for proving the theorems.*

> **A1** *The function $f$ is $\beta$-smooth.*

> **A2** *The function $f$ is $\rho$-Hessian Lipschitz.*

> **A3** *The norm of the stochastic gradient noise is bounded i.e, $\|g(\mathbf{x}_t) - \nabla f(\mathbf{x}_t)\| \leq Q \quad \forall t \geq 0$. Further, $\mathbb{E}[Q^2] \leq \sigma^2$.*

> **A4** *The function $f$ has strict saddle property.*

> **A5** *The function $f$ is bounded i.e., $|f(\mathbf{x})| \leq B, \ \forall \mathbf{x} \in \mathbb{R}^d$.*

173   ***A6*** *The function f is locally* $\alpha-$*strongly convex i.e, in the* $\delta$*-neighborhood of a locally optimal*
174    *point* $\mathbf{x}^*$ *for some* $\delta > 0$.

175 **Remark 2.** *If* $\nabla \tilde{f}(\tilde{\mathbf{x}}_t)$ *and* $\tilde{g}(\tilde{\mathbf{x}}_t)$ *are the gradient and stochastic gradient of the second order Taylor*
176 *approximation of f about the iterate* $\tilde{\mathbf{x}}_t$*, from Assumption **A3**, it is implied that* $\left\| \tilde{g}(\tilde{\mathbf{x}}_t) - \nabla \tilde{f}(\tilde{\mathbf{x}}_t) \right\| \leq$
177 $\tilde{Q}$*. Further,* $\mathbb{E}[\tilde{Q}^2] \leq \tilde{\sigma}^2$.

178 Note that these assumptions are similar to those in the perturbed gradient literature [9, 15, 16]. We
179 call attention to two significant differences in our approach compared to other perturbed gradient
180 methods such as [15, 9, 16]: (i) In contrast to the isotropic *additive* perturbation commonly added to
181 the SGD update, we introduce randomness in our learning rate, manifested as *multiplicative* noise
182 in the update. This makes the characterization of the total noise dependent on the gradient, making
183 the analysis challenging. (ii) The magnitude of noise injected is computed through the smoothness
184 constants in the work by Jin et al. [15, 16]; instead, we treat the parameters $L_{min}$ and $L_{max}$ as
185 hyperparameters to be tuned. This enables our PLRS method to be easily applied to training deep
186 neural networks where the computation of these smoothness constants could be infeasible due to
187 sheer computational complexity.

188 We reiterate the update equations of the proposed SGD-PLRS.

$$\mathbf{x}_{t+1} = \mathbf{x}_t - \eta_c \nabla f(\mathbf{x}_t) - \mathbf{w}_t. \tag{3}$$

189
$$\mathbf{w}_t = \eta_c g(\mathbf{x}_t) - \eta_c \nabla f(\mathbf{x}_t) + u_{t+1} g(\mathbf{x}_t). \tag{4}$$

190 Note that the term $\mathbf{w}_t$ has zero mean and we state this formally in the lemma below.

191 **Lemma 1** (Zero mean property). *The mean of* $\mathbf{w}_{t-1}$ $\forall t \geq 1$ *is* 0.

*Proof.*
$$\begin{aligned} \mathbb{E}[\mathbf{w}_{t-1}] &= \mathbb{E}\left[\eta_c g(\mathbf{x}_{t-1}) - \eta_c \nabla f(\mathbf{x}_{t-1})\right] + \mathbb{E}\left[u_t g(\mathbf{x}_{t-1})\right] \\ &= \mathbf{0} \qquad \forall t \geq 1. \end{aligned} \tag{8}$$

192 This follows as $\mathbb{E}[u_t] = \frac{L_{\min} + L_{\max} - 2\eta_c}{2} = 0$ and $\mathbb{E}\left[g(\mathbf{x}_{t-1})\right] = \nabla f(\mathbf{x}_{t-1})$.    $\square$

193 For a function satisfying the Assumptions **A1**-**A6**, there are three possibilities for the iterate $\mathbf{x}_t$ with
194 respect to the function's gradient and Hessian, namely, **B1:** Gradient is large; **B2:** Gradient is small
195 and iterate is around a saddle point; **B3:** Gradient is small and iterate is around a $\nu$-SOSP.

196 We now present three theorems corresponding to each of these cases. Our first result pertains to the
197 case **B1** where the gradient of the iterate is large.

198 **Theorem 1.** *Under the assumptions **A1** and **A3** with* $L_{max} < \frac{1}{\beta}$*, for any point* $\mathbf{x}_t$ *with* $\|\nabla f(\mathbf{x}_t)\| \geq$
199 $\sqrt{3\eta_c \beta \sigma^2}$ *where* $\sqrt{3\eta_c \beta \sigma^2} < \epsilon$*, after one iteration, we have*

$$\mathbb{E}[f(\mathbf{x}_{t+1})] - f(\mathbf{x}_t) \leq -\tilde{\Omega}(L_{max}^2).$$

200 This theorem suggests that, for any iterate $\mathbf{x}_t$ for which the gradient is large, the expected functional
201 value of the subsequent iterate $f(\mathbf{x}_{t+1})$ decreases, and the corresponding decrease $\mathbb{E}[f(\mathbf{x}_{t+1})] - f(\mathbf{x}_t)$
202 is in the order of $\tilde{\Omega}(L_{max}^2)$. The formal proof for this theorem can be found in Appendix A.

203 The next theorem corresponds to the case **B2** where the gradient is small and the Hessian is negative.

204 **Theorem 2.** *Consider f satisfying Assumptions **A1** - **A5**. Let* $\{\mathbf{x}_t\}$ *be the SGD iterates of the function*
205 *f using PLRS. Let* $\|\nabla f(\mathbf{x}_0)\| \leq \sqrt{3\eta_c \beta \sigma^2} < \epsilon$ *and* $\lambda_{min}(\mathbf{H}(\mathbf{x}_0)) \leq -\gamma$ *where* $\epsilon, \gamma > 0$*. Then, there*
206 *exists a* $T = \tilde{O}\left(L_{max}^{-1/4}\right)$ *such that with probability at least* $1 - \tilde{O}\left(L_{max}^{7/2}\right)$*,*

$$\mathbb{E}[f(\mathbf{x}_T) - f(\mathbf{x}_0)] \leq -\tilde{\Omega}\left(L_{max}^{3/4}\right).$$

207 The formal proof of this theorem is provided in Appendix C. The sketch of the proof is given below.
208

209 **Proof Sketch** This theorem shows that the iterates obtained using PLRS escape from a saddle point
210 $\mathbf{x}_0$ (where the gradient is small, and the Hessian has atleast one negative eigenvalue), i.e, it shows

the decrease in the expected value of the function $f$ after $T = \tilde{O}\left(L_{max}^{-1/4}\right)$ iterations. Note that for a $\rho-$Hessian smooth function,

$$f(\mathbf{x}_T) \leq f(\mathbf{x}_0) + \nabla f(\mathbf{x}_0)^T(\mathbf{x}_T - \mathbf{x}_0) + \frac{1}{2}(\mathbf{x}_T - \mathbf{x}_0)^T\mathbf{H}(\mathbf{x}_0)(\mathbf{x}_T - \mathbf{x}_0) + \frac{\rho}{6}\|\mathbf{x}_T - \mathbf{x}_0\|^3. \quad (9)$$

To evaluate $\mathbb{E}[f(\mathbf{x}_T) - f(\mathbf{x}_0)]$ from (9), we require an analytical expression for $\mathbf{x}_T - \mathbf{x}_0$, which is not tractable. Hence, we employ the second-order Taylor approximation of the function $f$, which we denote as $\tilde{f}$. We then apply SGD-PLRS on $\tilde{f}$ to obtain $\tilde{\mathbf{x}}_T$. Following this, we write $\mathbf{x}_T - \mathbf{x}_0 = (\mathbf{x}_T - \tilde{\mathbf{x}}_T) + (\tilde{\mathbf{x}}_T - \mathbf{x}_0)$ and derive expressions for upper bounds on $\tilde{\mathbf{x}}_T - \mathbf{x}_0$ and $\mathbf{x}_T - \tilde{\mathbf{x}}_T$ which hold with high probability in Lemmas 2 and 3, respectively (given in Appendix B.1 and B.2).

We split the quadratic term in (9) into two parts corresponding to $\tilde{\mathbf{x}}_T - \mathbf{x}_0$ and $\mathbf{x}_T - \tilde{\mathbf{x}}_T$. We further decompose the term, say $\mathcal{Y} = (\tilde{\mathbf{x}}_T - \mathbf{x}_0)^T\mathbf{H}(\mathbf{x}_0)(\tilde{\mathbf{x}}_T - \mathbf{x}_0)$ into its eigenvalue components along each dimension with corresponding eigenvalues $\lambda_1, \ldots, \lambda_d$ of $\mathbf{H}(\mathbf{x}_0)$. Our main result in this theorem proves that the term $\mathcal{Y}$ dominates over all the other terms of (9), and that it is bounded by a negative value, thereby, proving $\mathbb{E}[f(\mathbf{x}_T)] \leq f(\mathbf{x}_0)$. This main result uses a two-pronged proof. Firstly, we use our assumption that the initial iterate $\mathbf{x}_0$ is at a saddle point and hence at least one of $\lambda_i, \quad 1 \leq i \leq d$ is negative. We formally show that the eigenvector corresponding to this eigenvalue points to the direction of escape. Secondly, we use the second order statistics of our noise, to show that the magnitude of $\mathcal{Y}$ is large enough to dominate over the other terms of (9). Note that our noise term involves the stochasticity in the gradient and the probabilistic learning rate. Hence, we have shown that the negative eigenvalue of the Hessian at a saddle point and the unique characterization of the noise is sufficient to force a descent along the negative curvature safely out of the region of the saddle point within $T$ iterations. ∎

As each SGD-PLRS update is noisy, we need to ensure that once we escape a saddle point and move towards a local minimum (case **B3**), we do not overshoot the minimum but rather, stay in the $\delta-$neighborhood of an SOSP, with high probability. We formalize this in Theorem 3.

**Theorem 3.** *Consider $f$ satisfying the assumptions A1-A6. Let the initial iterate $\boldsymbol{x}_0$ be $\delta$ close to a local minimum $\boldsymbol{x}^*$ such that $\|\boldsymbol{x}_0 - \boldsymbol{x}^*\| \leq \tilde{O}(\sqrt{L_{max}}) < \delta$. With probability at least $1 - \xi$, $\forall t \leq T$ where $T = \tilde{O}\left(\frac{1}{L_{max}^2}\log\frac{1}{\xi}\right)$,*

$$\|\boldsymbol{x}_t - \boldsymbol{x}^*\| \leq \tilde{O}\left(\sqrt{L_{max}\log\frac{1}{L_{max}\xi}}\right) < \delta$$

This theorem deals with the case that the initial iterate $\mathbf{x}_0$ is $\delta$-close to a local minimum $\mathbf{x}^*$ (case **B3**). We prove that the subsequent iterates are also in the same neighbourhood, i.e., $\delta$ close to the local minimum, with high probability. In other words, we prove that the sequence $\{\|\mathbf{x}_t - \mathbf{x}^*\|\}$ is bounded by $\delta$ for $t \leq T$. In the neighbourhood of the local minimum, gradients are small and subsequently, the change in iterates, $\mathbf{x}_t - \mathbf{x}_{t-1}$ are minute. Therefore, the iterates stay near the local minimum with high probability. It is worth noting that the nature of the noise, which is comprised of stochastic gradients (whose stochasticity is bounded by $Q$) multiplied with a bounded uniform random variable (owing to PLRS), aids in proving our result. We provide the formal proof in Appendix D.

## 5 Empirical evaluation

We provide results on CIFAR-10, CIFAR-100 [18] and Tiny ImageNet [20] and compare with: (i) cosine annealing with warm restarts [21], (ii) one-cycle scheduler [25], (iii) knee scheduler [14], (iv) constant learning rate and (v) multi-step decay scheduler. We run experiments for 500 epochs for the CIFAR datasets and for 100 epochs for the Tiny ImageNet dataset using the SGD optimizer for all schedulers [2]. We also set all other regularization parameters, such as weight decay and dampening, to zero. We use a batch size of 64 for DenseNet-40-12, 50 for ResNet-50, and 128 for the others. We conduct all our experiments in a single NVIDIA GeForce RTX 2080 GPU card.

To determine the parameters $L_{min}$ and $L_{max}$ for PLRS, we perform a range test, where we observe the training loss for a range of learning rates as is done in state-of-the-art LR schedulers such as

---

[2]We provide results without momentum to be consistent with our theoretical framework. When we used the SGD optimizer with momentum for PLRS, we obtain results better than those reported without momentum.

| Architecture | Scheduler | Max acc. | Mean acc. (S.D) |
|---|---|---|---|
| VGG-16 | Cosine | 96.87 | 96.09 (0.78) |
| VGG-16 | Knee | 96.87 | **96.35** (0.45) |
| VGG-16 | One-cycle | 90.62 | 89.06 (1.56) |
| VGG-16 | Constant | 96.09 | 96.06 (0.05) |
| VGG-16 | Multi-step | 92.97 | 92.45 (0.90) |
| VGG-16 | PLRS (ours) | **97.66** | 96.09 (1.56) |
| WRN-28-10 | Cosine | 92.03 | 91.90 (0.13) |
| WRN-28-10 | Knee | **92.04** | 91.64 (0.63) |
| WRN-28-10 | One-cycle | 87.76 | 87.37 (0.35) |
| WRN-28-10 | Constant | **92.04** | **92.00** (0.08) |
| WRN-28-10 | Multi-step | 88.94 | 88.80 (0.21) |
| WRN-28-10 | PLRS (ours) | 92.02 | 91.43 (0.54) |

Table 1: Maximum and mean (with standard deviation) test accuracies over 3 runs for CIFAR-10.

one-cycle [25] and knee schedulers [14]. As the learning rate is gradually increased, we first observe a steady decrease in the training loss, then followed by a drastic increase. We note the learning rate at which there is an increase of training loss, say $\bar{L}$ and choose the maximum learning rate $L_{max}$ to be just below $\bar{L}$, where the loss is still decreasing. We then tune $L_{min}$ such that $0 < L_{min} < L_{max}$. We choose the parameters for the baseline schedulers as suggested in the original papers (further details of parameters are provided in Appendix F).

While there are ample works which prove the convergence of SGD with additive noise as in [9], they cannot be ported into practice for deep neural networks. They require smoothness constants [33, 16, 2] or functional bounds on the norms of the function derivatives [28] to be computed for the additive noise injection, which can not be obtained for the loss functions of neural networks or can only be approximated locally [19]. Further, the empirical convergence properties of noisy SGD are not demonstrated through examples in the majority of these analytical works which makes it hard to compare their convergence with PLRS. However, we compare our proposed PLRS against the noisy SGD mechanism proposed by Ge et al. [9] providing convergence results on the online tensor decomposition problem using the code provided by the authors in Appendix G.

## 5.1 Results for CIFAR-10

We consider VGG-16 [23] and WRN-28-10 [29] and use $L_{min} = 0.07$ and $L_{max} = 0.1$ for both the networks. We record the maximum and mean test accuracies across different LR schedulers in Table 1. The highest accuracy across schedulers is recorded in bold. For the VGG-16 network, we rank the highest in terms of maximum test accuracy. In terms of the mean test accuracy over 3 runs, the knee scheduler outperforms the rest. Note that the second highest mean test accuracy is achieved by both PLRS and the cosine annealing schedulers. Unsurprisingly, the constant scheduler has the lowest standard deviation. In the WRN-28-10 network, note that the maximum test accuracies for the cosine, knee, constant and the PLRS schedulers are very similar (difference in the order of $10^{-2}$). Their similar performance is also reflected in the mean test accuracies although the constant learning rate edges out the other schedulers marginally. To study the convergence of the schedulers we also plot the training loss across epochs in Figure 1. We observe that our proposed PLRS achieves one of the fastest rates of convergence in terms of the training loss compared across all the schedulers for both networks. Note that the cosine annealing scheduler records several spikes across the training.

## 5.2 Results for CIFAR-100

We consider networks ResNet-110 [10] and DenseNet-40-12 [13], and use $L_{min} = 0.07$ and $L_{max} = 0.1$ for the former, and $L_{min} = 0.1$ and $L_{max} = 0.2$ for the latter. The maximum and the mean test accuracies (with standard deviation) across 3 runs are provided in Table 2. For ResNet-110, PLRS performs best in terms of the maximum and the mean test accuracies. This is closely followed by the other state-of-the-art LR schedulers such as knee and cosine schedulers. For the DenseNet-40-12 network, PLRS comes to a close second to the multi-step LR scheduler in terms of the maximum and mean test accuracies. However, it is important to note that the multi-step scheduler records the

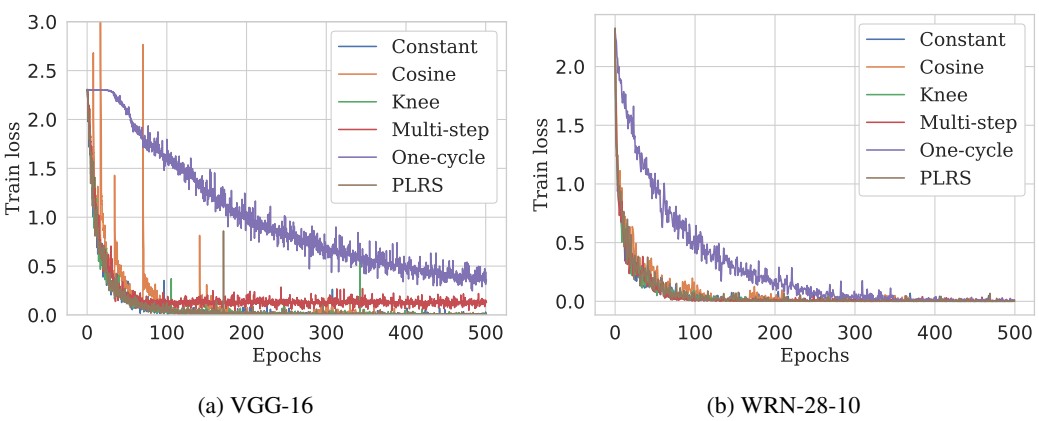

|     | (a) VGG-16 |     | (b) WRN-28-10 |
| --- | --- | --- | --- |

Figure 1: Training loss vs epochs for VGG-16 and WRN-28-10 for CIFAR-10.

| Architecture | Scheduler | Max acc. | Mean acc.(S.D) |
| --- | --- | --- | --- |
| ResNet-110 | Cosine | 74.22 | 72.66 (1.56) |
| ResNet-110 | Knee | 75.78 | 72.39 (2.96) |
| ResNet-110 | One-cycle | 71.09 | 70.05 (1.19) |
| ResNet-110 | Constant | 69.53 | 66.67 (2.51) |
| ResNet-110 | Multi-step | 63.28 | 61.20 (2.39) |
| ResNet-110 | PLRS (ours) | **77.34** | **74.61** (2.95) |
| DenseNet-40-12 | Cosine | 82.81 | 80.47 (2.07) |
| DenseNet-40-12 | Knee | 82.81 | 80.73 (2.39) |
| DenseNet-40-12 | One-cycle | 73.44 | 72.39 (0.90) |
| DenseNet-40-12 | Constant | 82.81 | 80.73 (2.39) |
| DenseNet-40-12 | Multi-step | **87.50** | **84.89** (2.39) |
| DenseNet-40-12 | PLRS (ours) | 84.37 | 83.33 (0.90) |

Table 2: Maximum and mean (with standard deviation) test accuracies over 3 runs for CIFAR-100.

least test accuracy with the ResNet-110 network. Hence, its performance is not consistent across the networks, while PLRS is consistently one of the best performing schedulers.

We plot the training loss in Figure 2. For ResNet-110, both PLRS and knee LR scheduler converge to a low training loss around $150$ epochs. While cosine annealing LR scheduler also seems to converge fast, it experiences sharp spikes along the curve during the restarts. For DenseNet-40-12, PLRS converges faster to a lower training loss compared to the other schedulers.

### 5.3 Results for Tiny ImageNet

We consider the Resnet-50 [10] architecture and use $L_{min} = 0.35$ and $L_{max} = 0.4$. We present the maximum and mean test accuracies in Table 3. We provide the plot of training loss in Figure 3. PLRS performs the best in terms of maximum test accuracy. In terms of mean test accuracy, it ranks second next to cosine annealing by a close margin. It can be observed that PLRS achieves the fastest convergence to the lowest training loss compared to others. Moreover, it exhibits stable convergence, especially when compared cosine annealing, which experiences multiple spikes due to warm restarts.

### 5.4 Limitations and broader impact

In line with all other works which focus on convergence proofs, our work too applies only to a restricted class of functions that meet the assumptions in Section 4. In contrast, our experiments are conducted on deep neural networks, which may not strictly satisfy these assumptions. While this is a limitation of our work, we note that many papers focused on theoretical convergence of SGD do not include empirical results, and many practice-oriented papers proposing new LR schedulers do

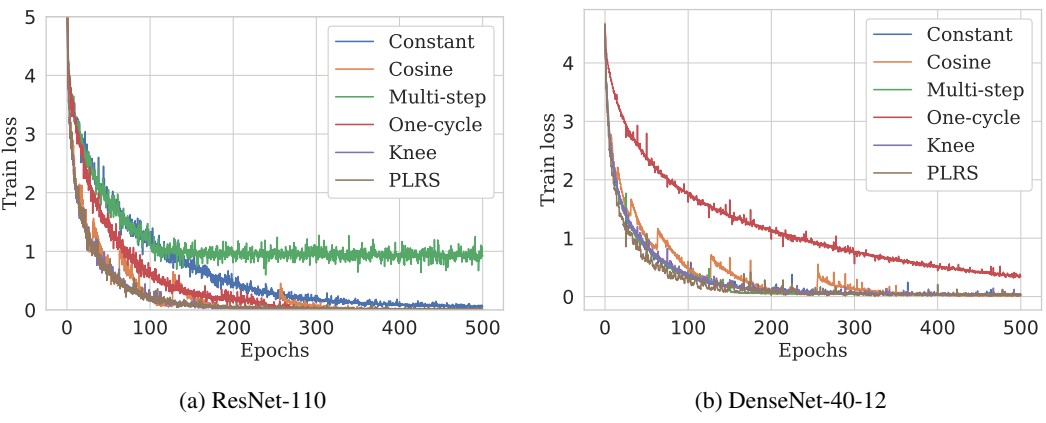

(a) ResNet-110          (b) DenseNet-40-12

Figure 2: Training loss vs epochs for ResNet-110 and DenseNet-40-12 on CIFAR-100.

| Scheduler | Max acc. | Mean acc. (S.D) |
|-----------|----------|-----------------|
| Cosine | 62.13 | **62.03** (0.15) |
| Knee | 61.93 | 61.50 (0.42) |
| One-cycle | 52.24 | 51.99 (0.22) |
| Constant | 61.59 | 61.11 (0.42) |
| Multi-step | 61.28 | 61.20 (0.08) |
| PLRS (ours) | **62.34** | 61.90 (0.73) |

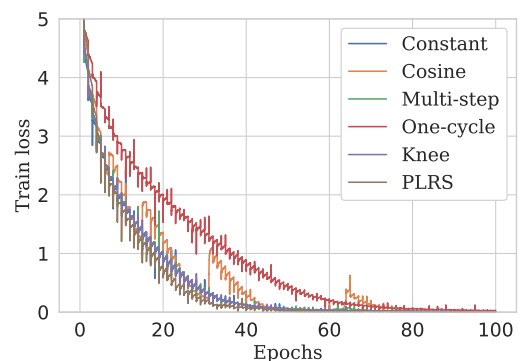

Table 3: Maximum and mean (with standard deviation) test accuracies over 3 runs for Tiny ImageNet.

Figure 3: Training loss vs epochs for ResNet-50 with Tiny ImageNet.

not include convergence proofs. Another limitation is that our experiments are limited to benchmark image datasets, even though our proposed scheduler is general and can be applied to other domains.

Our work contributes to the relatively underexplored theoretical understanding of LR schedulers, an area where most prior research has focused on empirical or application-driven results. As discussed earlier, commonly used periodic schedulers, such as triangular or cosine annealing, can be viewed as special cases of our proposed PLRS. This generalization opens new avenues for theoretical investigation, including the analysis of convergence properties across a broader class of schedulers. In practice, PLRS demonstrates improved stability and enables faster convergence, reducing the number of training epochs required. This efficiency translates to lower GPU usage and energy consumption, supporting more sustainable and resource-conscious AI development.

## 6 Concluding remarks

We have proposed the novel idea of a probabilistic LR scheduler. The probabilistic nature of the scheduler helped us provide the first theoretical convergence proofs for SGD using LR schedulers. In our opinion, this is a significant step in the right direction to bridge the gap between theory and practice in the LR scheduler domain. Our empirical results show that our proposed LR scheduler performs competitively with the state-of-the-art cyclic schedulers, if not better, on CIFAR-10, CIFAR-100, and Tiny ImageNet datasets for a wide variety of popular deep architectures. This leads us to hypothesize that the proposed probabilistic LR scheduler acts as a super-class of LR schedulers encompassing both probabilistic and deterministic schedulers. Future research directions include further exploration of this hypothesis.

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

# Appendix

## A    Proof of Theorem 1

**Theorem 4** (Theorem 1 restated). *Under the assumptions A1 and A3 with $L_{max} < \frac{1}{\beta}$, for any point $\mathbf{x}_t$ with $\|\nabla f(\mathbf{x}_t)\| \geq \sqrt{3\eta_c\beta\sigma^2}$ where $\sqrt{3\eta_c\beta\sigma^2} < \epsilon$ (satisfying B1), after one iteration we have,*

$$\mathbb{E}[f(\mathbf{x}_{t+1})] - f(\mathbf{x}_t) \leq -\tilde{\Omega}(L_{max}^2).$$

*Proof.* Using the second order Taylor series approximation for $f(\mathbf{x}_{t+1})$ around $\mathbf{x}_t$, where $\mathbf{x}_{t+1} = \mathbf{x}_t - \eta_c\nabla f(\mathbf{x}_t) - \mathbf{w}_t$, we have

$$f(\mathbf{x}_{t+1}) - f(\mathbf{x}_t) \leq \nabla f(\mathbf{x}_t)^T (\mathbf{x}_{t+1} - \mathbf{x}_t) + \frac{\beta}{2} \|\mathbf{x}_{t+1} - \mathbf{x}_t\|^2,$$

following the result from [22, Lemma 1.2.3]. Taking expectation w.r.t. $\mathbf{w}_t$,

$$\begin{aligned}
\mathbb{E}[f(\mathbf{x}_{t+1})] - f(\mathbf{x}_t) &\leq \nabla f(\mathbf{x}_t)^T \mathbb{E}[\mathbf{x}_{t+1} - \mathbf{x}_t] + \frac{\beta}{2}\mathbb{E}[\|\mathbf{x}_{t+1} - \mathbf{x}_t\|^2] \\
&= \nabla f(\mathbf{x}_t)^T \mathbb{E}[-\eta_c\nabla f(\mathbf{x}_t) - \mathbf{w}_t] + \frac{\beta}{2}\mathbb{E}[\|-\eta_c\nabla f(\mathbf{x}_t) - \mathbf{w}_t\|^2] \qquad (10) \\
&= -\eta_c \|\nabla f(\mathbf{x}_t)\|^2 + \frac{\beta}{2}\mathbb{E}[\eta_c^2 \|\nabla f(\mathbf{x}_t)\|^2 + \|\mathbf{w}_t\|^2],
\end{aligned}$$

since $\mathbb{E}[\mathbf{w}_t] = 0$ due to the zero mean property in Lemma 1. We focus on the last term in the next steps. Expanding $\|\mathbf{w}_t\|^2$,

$$\begin{aligned}
\|\mathbf{w}_t\|^2 &= (\eta_c g(\mathbf{x}_t) - \eta_c\nabla f(\mathbf{x}_t) + u_{t+1}g(\mathbf{x}_t))^T (\eta_c g(\mathbf{x}_t) - \eta_c\nabla f(\mathbf{x}_t) + u_{t+1}g(\mathbf{x}_t)) \\
&= \eta_c^2 \|g(\mathbf{x}_t)\|^2 - \eta_c^2 g(\mathbf{x}_t)^T\nabla f(\mathbf{x}_t) + \eta_c u_{t+1} \|g(\mathbf{x}_t)\|^2 - \eta_c^2\nabla f(\mathbf{x}_t)^T g(\mathbf{x}_t) + \eta_c^2 \|\nabla f(\mathbf{x}_t\|^2 \\
&\quad - \eta_c u_{t+1}\nabla f(\mathbf{x}_t)^T g(\mathbf{x}_t) + \eta_c u_{t+1} \|g(\mathbf{x}_t)\|^2 - \eta_c u_{t+1}g(\mathbf{x}_t)^T\nabla f(\mathbf{x}_t) + u_{t+1}^2 \|g(\mathbf{x}_t)\|^2.
\end{aligned}$$

Taking expectation with respect to $\mathbf{x}_t$ and noting that $\mathbb{E}[u_{t+1}] = 0$ and $\mathbb{E}[g(\mathbf{x}_t)] = \nabla f(\mathbf{x}_t)$,[3]

$$\mathbb{E}[\|\mathbf{w}_t\|^2] = \eta_c^2\mathbb{E}[\|g(\mathbf{x}_t)\|^2] - \eta_c^2 \|\nabla f(\mathbf{x}_t)\|^2 + \mathbb{E}[u_{t+1}^2]\mathbb{E}[\|g(\mathbf{x}_t)\|^2]. \qquad (11)$$

Now, as per assumption **A3**,

$$\begin{aligned}
\|g(\mathbf{x}_t) - \nabla f(\mathbf{x}_t)\|^2 &\leq Q^2 \\
\|g(\mathbf{x}_t)\|^2 + \|\nabla f(\mathbf{x}_t)\|^2 - 2g(\mathbf{x}_t)^T\nabla f(\mathbf{x}_t) &\leq Q^2 \\
\|g(\mathbf{x}_t)\|^2 &\leq Q^2 - \|\nabla f(\mathbf{x}_t)\|^2 + 2g(\mathbf{x}_t)^T\nabla f(\mathbf{x}_t) \\
\mathbb{E}[\|g(\mathbf{x}_t)\|^2] &\leq \mathbb{E}[Q^2] - \|\nabla f(\mathbf{x}_t)\|^2 + 2 \|\nabla f(\mathbf{x}_t)\|^2 \leq \sigma^2 + \|\nabla f(\mathbf{x}_t)\|^2, \qquad (12)
\end{aligned}$$

as $\mathbb{E}[Q^2] \leq \sigma^2$. Applying (12) to (11),

$$\begin{aligned}
\mathbb{E}[\|\mathbf{w}_t\|^2] &\leq \eta_c^2\sigma^2 + \eta_c^2 \|\nabla f(\mathbf{x}_t)\|^2 - \eta_c^2 \|\nabla f(\mathbf{x}_t)\|^2 + \mathbb{E}[u_{t+1}^2]\sigma^2 + \mathbb{E}[u_{t+1}^2] \|\nabla f(\mathbf{x}_t)\|^2 \\
&= \eta_c^2\sigma^2 + \mathbb{E}[u_{t+1}^2]\sigma^2 + \mathbb{E}[u_{t+1}^2] \|\nabla f(\mathbf{x}_t)\|^2 \qquad\qquad (13) \\
&= \eta_c^2\sigma^2 + \frac{(L_{max} - L_{min})^2\sigma^2}{12} + \frac{(L_{max} - L_{min})^2 \|\nabla f(\mathbf{x}_0)\|^2}{12},
\end{aligned}$$

---

[3]Note that there are two random variables in $\mathbf{w}_t$ which are the stochastic gradient $g(\mathbf{x}_t)$ and the uniformly distributed LR $u_{t+1}$ due to our proposed LR scheduler. Hence, the expectation is with respect to both these variables. Also note that $u_{t+1}$ and $g(\mathbf{x}_t)$ are independent of each other.

438 since the second moment of a uniformly distributed random variable in the interval $[L_{min}-\eta_c, L_{max}-$
439 $\eta_c]$ is given by $\frac{(L_{max}-L_{min})^2}{12}$. Using (13) in (10) and $\eta_c = \frac{L_{min}+L_{max}}{2}$,

$$\mathbb{E}[f(\mathbf{x}_{t+1})] - f(\mathbf{x}_t) \leq -\eta_c \|\nabla f(\mathbf{x}_t)\|^2 + \frac{\beta}{2}\eta_c^2 \|\nabla f(\mathbf{x}_t)\|^2 + \frac{\beta\eta_c^2\sigma^2}{2} + \frac{\beta(L_{max}-L_{min})^2\sigma^2}{24}$$
$$+ \frac{\beta(L_{max}-L_{min})^2 \|\nabla f(\mathbf{x}_t)\|^2}{24}$$
$$\leq -\eta_c \|\nabla f(\mathbf{x}_t)\|^2 + \frac{\beta}{2}\eta_c^2 \|\nabla f(\mathbf{x}_t)\|^2 + \frac{\beta\eta_c^2\sigma^2}{2} + \frac{\beta\eta_c^2\sigma^2}{6} + \frac{\beta\eta_c^2 \|\nabla f(\mathbf{x}_0)\|^2}{6}$$
$$= -\|\nabla f(\mathbf{x}_t)\|^2 \left(\eta_c - \frac{2\beta\eta_c^2}{3}\right) + \frac{2\beta\eta_c^2\sigma^2}{3}$$

440 Now, applying our initial assumption that $\|\nabla f(\mathbf{x}_t)\| \geq \sqrt{3\eta_c\beta\sigma^2}$, we have,

$$\mathbb{E}[f(\mathbf{x}_{t+1})] - f(\mathbf{x}_t) \leq -3\eta_c\beta\sigma^2 \left(\eta_c - \frac{2\beta\eta_c^2}{3}\right) + \frac{2\beta\eta_c^2\sigma^2}{3} = -3\eta_c^2\beta\sigma^2 + \frac{6\beta^2\eta_c^3\sigma^2}{3} + \frac{2\beta\eta_c^2\sigma^2}{3}$$

441 Since $L_{max} < \frac{1}{\beta}$ and $\eta_c = \frac{L_{min}+L_{max}}{2}$, we have $\eta_c\beta < L_{max}\beta < 1$. Finally,

$$\mathbb{E}[f(\mathbf{x}_{t+1})] - f(\mathbf{x}_t) \leq -3\eta_c^2\beta\sigma^2 + \frac{6\beta\eta_c^2\sigma^2}{3} + \frac{2\beta\eta_c^2\sigma^2}{3} = -\frac{\beta\eta_c^2\sigma^2}{3}$$
$$= -\tilde{\Omega}(\eta_c^2),$$

442 which proves the theorem. □

## B  Additional results needed to prove Theorem 2

444 Here, we state and prove two lemmas that are instrumental in the proof of Theorem 2.

### B.1  Proof of Lemma 2

446 In the following Lemma, we prove that the gradients of a second order approximation of $f$ are
447 probabilistically bounded for all $t \leq T$ and its iterates as we apply SGD-PLRS are also bounded
448 when the initial iterate $\mathbf{x}_0$ is a saddle point.

449 **Lemma 2.** *Let $f$ satisfy Assumptions A1 - A4. Let $\tilde{f}$ be the second order Taylor approximation*
450 *of $f$ and let $\tilde{\mathbf{x}}_t$ be the iterate at time step $t$ obtained using the SGD update equation as in* (3)
451 *on $\tilde{f}$; let $\tilde{\mathbf{x}}_0 = \mathbf{x}_0$, $\|\nabla f(\mathbf{x}_0)\| \leq \epsilon$ and the minimum eigenvalue of the Hessian of $f$ at $\mathbf{x}_0$ be*
452 *$\lambda_{min}(\boldsymbol{H}(\boldsymbol{x}_0)) = -\gamma_o$ where $\gamma_o > 0$. With probability at least $1 - \tilde{O}(L_{max}^{15/4})$, we have*

$$\left\|\nabla\tilde{f}(\tilde{\mathbf{x}}_t)\right\| \leq \tilde{O}\left(\frac{1}{L_{max}^{0.5}}\right), \quad \|\tilde{\mathbf{x}}_t - \boldsymbol{x}_0\| \leq \tilde{O}\left(L_{max}^{3/8}\log\left(\frac{1}{L_{max}}\right)\right) \quad \forall t \leq T = \tilde{O}\left(L_{max}^{-1/4}\right).$$

453 *Proof.* As $\tilde{f}$ is the second order Taylor series approximation of $f$, we have

$$\tilde{f}(\tilde{\mathbf{x}}) = f(\mathbf{x}_0) + \nabla f(\mathbf{x}_0)^T(\tilde{\mathbf{x}} - \mathbf{x}_0) + \frac{1}{2}(\tilde{\mathbf{x}} - \mathbf{x}_0)^T\mathbf{H}(\mathbf{x}_0)(\tilde{\mathbf{x}} - \mathbf{x}_0).$$

454 Taking derivative w.r.t. $\tilde{\mathbf{x}}$, we have $\nabla\tilde{f}(\tilde{\mathbf{x}}) = \nabla f(\mathbf{x}_0) + \mathbf{H}(\mathbf{x}_0)(\tilde{\mathbf{x}} - \mathbf{x}_0)$. Now, note that $\nabla\tilde{f}(\tilde{\mathbf{x}}_{t-1}) =$
455 $\nabla f(\mathbf{x}_0) + \mathbf{H}(\mathbf{x}_0)(\tilde{\mathbf{x}}_{t-1} - \mathbf{x}_0) = K(\mathbf{x}_0) + \mathbf{H}(\mathbf{x}_0)\tilde{\mathbf{x}}_{t-1}$, where $K(\mathbf{x}_0) = \nabla f(\mathbf{x}_0) - \mathbf{H}(\mathbf{x}_0)\mathbf{x}_0 =$
456 $\nabla\tilde{f}(\tilde{\mathbf{x}}_{t-1}) - \mathbf{H}(\mathbf{x}_0)\tilde{\mathbf{x}}_{t-1}$. Therefore,

$$\nabla\tilde{f}(\tilde{\mathbf{x}}_t) = K(\mathbf{x}_0) + \mathbf{H}(\mathbf{x}_0)\tilde{\mathbf{x}}_t = \nabla\tilde{f}(\tilde{\mathbf{x}}_{t-1}) - \mathbf{H}(\mathbf{x}_0)\tilde{\mathbf{x}}_{t-1} + \mathbf{H}(\mathbf{x}_0)\tilde{\mathbf{x}}_t$$
$$= \nabla\tilde{f}(\tilde{\mathbf{x}}_{t-1}) + \mathbf{H}(\mathbf{x}_0)(\tilde{\mathbf{x}}_t - \tilde{\mathbf{x}}_{t-1}). \tag{14}$$

457 Next, using the SGD-PLRS update and rearranging,

$$\nabla\tilde{f}(\tilde{\mathbf{x}}_t) = \nabla\tilde{f}(\tilde{\mathbf{x}}_{t-1}) - \mathbf{H}(\mathbf{x}_0)(\eta_c\nabla\tilde{f}(\tilde{\mathbf{x}}_{t-1}) + \tilde{\mathbf{w}}_{t-1})$$
$$= (I - \eta_c\mathbf{H}(\mathbf{x}_0))\nabla\tilde{f}(\tilde{\mathbf{x}}_{t-1}) - \mathbf{H}(\mathbf{x}_0)\tilde{\mathbf{w}}_{t-1}, \tag{15}$$

where $I$ denotes the $d \times d$ identity matrix. Next, unrolling the term $\nabla \tilde{f}(\tilde{\mathbf{x}}_{t-1})$ recursively,

$$\nabla \tilde{f}(\tilde{\mathbf{x}}_t) = (I - \eta_c \mathbf{H}(\mathbf{x}_0))^t \nabla \tilde{f}(\tilde{\mathbf{x}}_0) - \mathbf{H}(\mathbf{x}_0) \sum_{\tau=0}^{t-1} (I - \eta_c \mathbf{H}(\mathbf{x}_0))^{t-\tau-1} \tilde{\mathbf{w}}_\tau. \tag{16}$$

Using the triangle and Cauchy-Schwartz inequalities,

$$\left\| \nabla \tilde{f}(\tilde{\mathbf{x}}_t) \right\| \leq \left\| (I - \eta_c \mathbf{H}(\mathbf{x}_0))^t \nabla \tilde{f}(\tilde{\mathbf{x}}_0) \right\| + \left\| \mathbf{H}(\mathbf{x}_0) \sum_{\tau=0}^{t-1} (I - \eta_c \mathbf{H}(\mathbf{x}_0))^{t-\tau-1} \tilde{\mathbf{w}}_\tau \right\|$$
$$\leq \left\| (I - \eta_c \mathbf{H}(\mathbf{x}_0))^t \right\| \left\| \nabla \tilde{f}(\tilde{\mathbf{x}}_0) \right\| + \| \mathbf{H}(\mathbf{x}_0) \| \left\| \sum_{\tau=0}^{t-1} (I - \eta_c \mathbf{H}(\mathbf{x}_0))^{t-\tau-1} \tilde{\mathbf{w}}_\tau \right\| \tag{17}$$

Note that the norm over the matrices refers to the matrix-induced norm. Since $\mathbf{H}(\mathbf{x}_0)$ is a real symmetric matrix, the induced norm gives the maximum eigenvalue of $\mathbf{H}(\mathbf{x}_0)$ i.e, $\lambda_{max}(\mathbf{H}(\mathbf{x}_0)) \leq \beta$ by our $\beta$-smoothness assumption **A1**. In the case of $(I - \eta_c \mathbf{H}(\mathbf{x}_0))$ the induced norm gives $(1 - \eta_c \lambda_{min}(\mathbf{H}(\mathbf{x}_0)))$ which is $(1 + \eta_c \gamma_o)$ as per our assumption that $\lambda_{min}(\mathbf{H}(\mathbf{x}_0)) = -\gamma_o$. Also recall that $\left\| \nabla \tilde{f}(\tilde{\mathbf{x}}_0) \right\| \leq \epsilon$. Now (17) becomes,

$$\left\| \nabla \tilde{f}(\tilde{\mathbf{x}}_t) \right\| \leq (1 + \eta_c \gamma_o)^t \epsilon + \beta \left\| \sum_{\tau=0}^{t-1} (I - \eta_c \mathbf{H}(\mathbf{x}_0))^{t-\tau-1} \tilde{\mathbf{w}}_\tau \right\|,$$
$$\leq (1 + \eta_c \gamma_o)^t \epsilon + \beta \sum_{\tau=0}^{t-1} (1 + \eta_c \gamma_o)^{t-\tau-1} \| \tilde{\mathbf{w}}_\tau \| . \tag{18}$$

Now, expanding the noise term $\tilde{\mathbf{w}}_\tau$,

$$\left\| \nabla \tilde{f}(\tilde{\mathbf{x}}_t) \right\| = (1 + \eta_c \gamma_o)^t \epsilon + \beta \sum_{\tau=0}^{t-1} (1 + \eta_c \gamma_o)^{t-\tau-1} \left\| \eta_c \tilde{g}(\tilde{\mathbf{x}}_\tau) - \eta_c \nabla \tilde{f}(\tilde{\mathbf{x}}_\tau) + u_{\tau+1} \tilde{g}(\tilde{\mathbf{x}}_\tau) \right\|$$

Now recall from our assumption **A3** that $\left\| \tilde{g}(\tilde{\mathbf{x}}_\tau) - \nabla \tilde{f}(\tilde{\mathbf{x}}_\tau) \right\| \leq \tilde{Q}$. Hence,

$$\left\| \nabla \tilde{f}(\tilde{\mathbf{x}}_t) \right\| \leq (1 + \eta_c \gamma_o)^t \epsilon + \beta \sum_{\tau=0}^{t-1} (1 + \eta_c \gamma_o)^{t-\tau-1} \left( \eta_c \tilde{Q} + |u_{\tau+1}| \left\| \tilde{g}(\tilde{\mathbf{x}}_\tau) - \nabla \tilde{f}(\tilde{\mathbf{x}}_\tau) + \nabla \tilde{f}(\tilde{\mathbf{x}}_\tau) \right\| \right)$$
$$\leq (1 + \eta_c \gamma_o)^t \epsilon + \beta \sum_{\tau=0}^{t-1} (1 + \eta_c \gamma_o)^{t-\tau-1} \left( \eta_c \tilde{Q} + |u_{\tau+1}| \left( \tilde{Q} + \left\| \nabla \tilde{f}(\tilde{\mathbf{x}}_\tau) \right\| \right) \right)$$

Using $\left\| \nabla \tilde{f}(\tilde{\mathbf{x}}_0) \right\| \leq \epsilon$ and $\left\| \nabla \tilde{f}(\tilde{\mathbf{x}}_1) \right\| \leq (1 + \eta_c \gamma_o) \epsilon + \epsilon + 2\tilde{Q}$, it can be proved by induction that the general expression for $t \geq 2$ is given by,

$$\left\| \nabla \tilde{f}(\tilde{\mathbf{x}}_t) \right\| \leq 10\tilde{Q} \sum_{\tau=0}^{\frac{t(t-1)}{2}} (1 + \eta_c \gamma_o)^\tau \tag{19}$$

We give the proof of (19) by induction in Appendix E. Next, we prove the bound on $\tilde{\mathbf{x}}_t - \tilde{\mathbf{x}}_0$. Using the SGD-PLRS update,

$$\tilde{\mathbf{x}}_t - \tilde{\mathbf{x}}_0 = - \sum_{\tau=0}^{t-1} \left( \eta_c \nabla \tilde{f}(\tilde{\mathbf{x}}_\tau) + \tilde{\mathbf{w}}_\tau \right)$$

$$= - \sum_{\tau=0}^{t-1} \left( \eta_c \left( (I - \eta_c \mathbf{H}(\mathbf{x}_0))^\tau \nabla \tilde{f}(\tilde{\mathbf{x}}_0) - \mathbf{H}(\mathbf{x}_0) \sum_{\tau'=0}^{\tau-1} (I - \eta_c \mathbf{H}(\mathbf{x}_0))^{\tau - \tau' - 1} \tilde{\mathbf{w}}_{\tau'} \right) + \tilde{\mathbf{w}}_\tau \right) \tag{20a}$$

$$= - \sum_{\tau=0}^{t-1} \eta_c (I - \eta_c \mathbf{H}(\mathbf{x}_0))^\tau \nabla f(\mathbf{x}_0) - \sum_{\tau=0}^{t-1} (I - \eta_c \mathbf{H}(\mathbf{x}_0))^{t-\tau-1} \tilde{\mathbf{w}}_\tau, \tag{20b}$$

where the equation (20a) is obtained by using (16). We obtain (20b) by using the summation of geometric series as $\mathbf{H}(\mathbf{x}_0)$ is invertible by the strict saddle property. As $\tilde{\mathbf{x}}_0 = \mathbf{x}_0$, we can write $\nabla \tilde{f}(\tilde{\mathbf{x}}_0) = \nabla f(\mathbf{x}_0)$. Taking norm,

$$
\begin{aligned}
\|\tilde{\mathbf{x}}_t - \tilde{\mathbf{x}}_0\| &\le \left\| \sum_{\tau=0}^{t-1} \eta_c (I - \eta_c \mathbf{H}(\mathbf{x}_0))^\tau \nabla f(\mathbf{x}_0) \right\| + \left\| \sum_{\tau=0}^{t-1} (I - \eta_c \mathbf{H}(\mathbf{x}_0))^{t-\tau-1} \tilde{\mathbf{w}}_\tau \right\| \\
&\le \sum_{\tau=0}^{t-1} \|\eta_c (I - \eta_c \mathbf{H}(\mathbf{x}_0))^\tau \nabla f(\mathbf{x}_0)\| + \sum_{\tau=0}^{t-1} \left\| (I - \eta_c \mathbf{H}(\mathbf{x}_0))^{t-\tau-1} \tilde{\mathbf{w}}_\tau \right\| \qquad (21) \\
&\le \eta_c \epsilon \sum_{\tau=0}^{t-1} (1 + \eta_c \gamma_o)^\tau + \sum_{\tau=0}^{t-1} (1 + \eta_c \gamma_o)^{t-\tau-1} \|\tilde{\mathbf{w}}_\tau\|.
\end{aligned}
$$

In (21), it can be seen that the first term is arbitrarily small by the initial assumption and that the second term decides the order of $\|\tilde{\mathbf{x}}_t - \tilde{\mathbf{x}}_0\|$. Hence, in order to bound $\|\tilde{\mathbf{x}}_t - \tilde{\mathbf{x}}_0\|$ probabilistically, it is sufficient to bound the second term, $\sum_{\tau=0}^{t-1} (1 + \eta_c \gamma_o)^{t-\tau-1} \|\tilde{\mathbf{w}}_\tau\|$. Now,

$$
\begin{aligned}
\sum_{\tau=0}^{t-1} (1 + \eta_c \gamma_o)^{t-\tau-1} \|\tilde{\mathbf{w}}_\tau\| &= \sum_{\tau=0}^{t-1} (1 + \eta_c \gamma_o)^{t-\tau-1} \left\| \eta_c \tilde{g}(\tilde{\mathbf{x}}_\tau) - \eta_c \nabla \tilde{f}(\tilde{\mathbf{x}}_\tau) + u_{\tau+1} \tilde{g}(\tilde{\mathbf{x}}_\tau) \right\| \\
&= \sum_{\tau=0}^{t-1} (1 + \eta_c \gamma_o)^{t-\tau-1} \left( \eta_c \tilde{Q} + |u_{\tau+1}| \left\| \tilde{g}(\tilde{\mathbf{x}}_\tau) - \nabla \tilde{f}(\tilde{\mathbf{x}}_\tau) + \nabla \tilde{f}(\tilde{\mathbf{x}}_\tau) \right\| \right) \\
&= \sum_{\tau=0}^{t-1} (1 + \eta_c \gamma_o)^{t-\tau-1} \tilde{Q} (\eta_c + |u_{\tau+1}|) + \sum_{\tau=0}^{t-1} (1 + \eta_c \gamma_o)^{t-\tau-1} |u_{\tau+1}| \left\| \nabla \tilde{f}(\tilde{\mathbf{x}}_\tau) \right\|
\end{aligned}
$$

Now, using $\left\| \nabla \tilde{f}(\tilde{\mathbf{x}}_0) \right\| \le \epsilon$, $\left\| \nabla \tilde{f}(\tilde{\mathbf{x}}_1) \right\| \le (1 + \eta_c \gamma_o) \epsilon + \epsilon + 2\tilde{Q}$ and (19) we write,

$$
\begin{aligned}
\sum_{\tau=0}^{t-1} (1 + \eta_c \gamma_o)^{t-\tau-1} \|\tilde{\mathbf{w}}_\tau\| &\le \sum_{\tau=0}^{t-1} (1 + \eta_c \gamma_o)^{t-\tau-1} \tilde{Q} (\eta_c + |u_{\tau+1}|) + (1 + \eta_c \gamma_o)^{t-1} |u_1| \epsilon + \\
&\quad (1 + \eta_c \gamma_o)^{t-2} |u_2| \left( (1 + \eta_c \gamma_o) \epsilon + \epsilon + 2\tilde{Q} \right) + \sum_{\tau=2}^{t-1} (1 + \eta_c \gamma_o)^{t-\tau-1} |u_{\tau+1}| 10\tilde{Q} \sum_{\tau'=0}^{\frac{\tau(\tau-1)}{2}} (1 + \eta_c \gamma_o)^{\tau'}
\end{aligned}
$$
$$(22)$$

It can be observed from (22) that the last term dominates the expression of and hence, it determines the order of $\|\tilde{\mathbf{x}}_t - \tilde{\mathbf{x}}_0\|$. We now apply Hoeffding's inequality to derive a probabilistic bound on $\|\tilde{\mathbf{x}}_t - \tilde{\mathbf{x}}_0\|$. According to Hoeffding's inequality for any summation $S_n = X_1 + \cdots + X_n$ such that $a_i \le X_i \le b_i$, $\mathbb{P}(S_n - \mathbb{E}[S_n] \ge \delta) \le \exp\left( \frac{-2\delta^2}{\sum_{i=1}^n (b_i - a_i)^2} \right)$. Now, setting $T = \tilde{O}\left( L_{max}^{-1/4} \right)$ from (41) and assuming $\eta_c \le \eta_{max} \le \frac{\sqrt{2}-1}{\gamma'}$, $\gamma_o \le \gamma'$, the squared bound of the summation $\sum_{\tau=2}^{t-1} (1 + \eta_c \gamma_o)^{t-\tau-1} |u_{\tau+1}| 10\tilde{Q} \sum_{\tau'=0}^{\frac{\tau(\tau-1)}{2}} (1 + \eta_c \gamma_o)^{\tau'} \le \tilde{O}\left( L_{max}^{3/4} \right)$, Setting $\delta = \tilde{O}\left( \sqrt{L_{max}^{3/4} \log\left( \frac{1}{L_{max}} \right)} \right)$, for some $t \le T$,

$$
\begin{aligned}
\mathbb{P}\left( \sum_{\tau=2}^{t-1} (1 + \eta_c \gamma_o)^{t-\tau-1} |u_{\tau+1}| 10\tilde{Q} \sum_{\tau'=0}^{\frac{\tau(\tau-1)}{2}} (1 + \eta_c \gamma_o)^{\tau'} \ge \tilde{O}\left( L_{max}^{3/8} \log\left( \frac{1}{L_{max}} \right) \right) \right) \\
\le \tilde{O}(L_{max}^4).
\end{aligned}
$$

Taking the union bound over all $t \le T$,

$$
\begin{aligned}
\mathbb{P}\left( \forall t \le T, \quad \sum_{\tau=2}^{t-1} (1 + \eta_c \gamma_o)^{t-\tau-1} |u_{\tau+1}| 10\tilde{Q} \sum_{\tau'=0}^{\frac{\tau(\tau-1)}{2}} (1 + \eta_c \gamma_o)^{\tau'} \ge \tilde{O}\left( L_{max}^{3/8} \log\left( \frac{1}{L_{max}} \right) \right) \right) \\
\le \tilde{O}\left( L_{max}^{15/4} \right),
\end{aligned}
$$

which completes our proof. $\qquad\square$

 ## B.2   Proof of Lemma 3

488 This lemma is used to derive an expression for a high probability upper bound of $\|\mathbf{x}_t - \tilde{\mathbf{x}}_t\|$ and
489 $\left\|\nabla f(\mathbf{x}_t) - \nabla \tilde{f}(\tilde{\mathbf{x}}_t)\right\|$.

490 **Lemma 3.** *Let $f : \mathbb{R}^d \to \mathbb{R}$ satisfy Assumptions **A1** - **A4**. Let $\tilde{f}$ be the second order Taylor's*
491 *approximation of $f$ and let $\boldsymbol{x}_t$, $\tilde{\boldsymbol{x}}_t$ be the iterates at time step $t$ obtained using the SGD-PLRS update*
492 *on $f$, $\tilde{f}$ respectively; let $\tilde{\boldsymbol{x}}_0 = \boldsymbol{x}_0$ and $\|\nabla f(\boldsymbol{x}_0)\| \leq \epsilon$. Let the minimum eigenvalue of the Hessian at*
493 *$\boldsymbol{x}_0$ be $\lambda_{min}(\nabla^2(f(\boldsymbol{x}_0))) = -\gamma_o$, where $\gamma_o > 0$. Then $\forall t \leq T = O\left(L_{max}^{-1/4}\right)$, with a probability of*
494 *at least $1 - \tilde{O}(L_{max}^{7/2})$,*

$$\|\boldsymbol{x}_t - \tilde{\boldsymbol{x}}_t\| \leq O\left(L_{max}^{3/4}\right) \quad and \quad \left\|\nabla f(\boldsymbol{x}_t) - \nabla \tilde{f}(\tilde{\boldsymbol{x}}_t)\right\| \leq O\left(L_{max}^{3/8} \log \frac{1}{L_{max}}\right).$$

495 *Proof.* The expression for $\mathbf{x}_t - \tilde{\mathbf{x}}_t$ can be written as,

$$\mathbf{x}_t - \tilde{\mathbf{x}}_t = (\mathbf{x}_t - \mathbf{x}_0) - (\tilde{\mathbf{x}}_t - \mathbf{x}_0)$$
$$= -\sum_{\tau=0}^{t-1} \left(\eta_c \nabla f(\mathbf{x}_\tau) + \mathbf{w}_\tau\right) - \left(-\sum_{\tau=0}^{t-1} \left(\eta_c \nabla \tilde{f}(\tilde{\mathbf{x}}_\tau) + \tilde{\mathbf{w}}_\tau\right)\right) = -\sum_{\tau=0}^{t-1} \left(\eta_c \Delta_\tau + (\mathbf{w}_\tau - \tilde{\mathbf{w}}_\tau)\right). \tag{23}$$

496 where we define $\Delta_t = \nabla f(\mathbf{x}_t) - \nabla \tilde{f}(\tilde{\mathbf{x}}_t)$. Now in order to bound $\|\mathbf{x}_t - \tilde{\mathbf{x}}_t\|$, we derive expressions
497 for both $\mathbf{w}_\tau - \tilde{\mathbf{w}}_\tau$ and $\Delta_\tau$. We initially focus on the term $\mathbf{w}_\tau - \tilde{\mathbf{w}}_\tau$.

$$\mathbf{w}_\tau - \tilde{\mathbf{w}}_\tau = \eta_c g(\mathbf{x}_\tau) - \eta_c \nabla f_\tau + u_{\tau+1} g(\mathbf{x}_\tau) - \left(\eta_c \tilde{g}(\tilde{\mathbf{x}}_\tau) - \eta_c \nabla \tilde{f}(\tilde{\mathbf{x}}_\tau) + u_{\tau+1} \tilde{g}(\tilde{\mathbf{x}}_\tau)\right)$$
$$= (u_{\tau+1} + \eta_c) \left(\left(g(\mathbf{x}_\tau) - \nabla f(\mathbf{x}_\tau)\right) - \left(\tilde{g}(\tilde{\mathbf{x}}_\tau) - \nabla \tilde{f}(\tilde{\mathbf{x}}_\tau)\right)\right) + u_{\tau+1} \Delta_\tau. \tag{24}$$

498 Taking norm on both sides,

$$\|\mathbf{w}_\tau - \tilde{\mathbf{w}}_\tau\| \leq |u_{\tau+1} + \eta_c| \left(Q + \tilde{Q}\right) + |u_{\tau+1}| \|\Delta_\tau\| \tag{25}$$

499 Using (24) and (25) in (23), and assumption **A3** that stochastic noise is bounded, and applying norm,

$$\|\mathbf{x}_t - \tilde{\mathbf{x}}_t\| = \left\|-\sum_{\tau=0}^{t-1} \left(\eta_c \Delta_\tau + (\mathbf{w}_\tau - \tilde{\mathbf{w}}_\tau)\right)\right\| \leq \sum_{\tau=0}^{t-1} \|\eta_c \Delta_\tau + (\mathbf{w}_\tau - \tilde{\mathbf{w}}_\tau)\|$$
$$\leq \sum_{\tau=0}^{t-1} (\eta_c + |u_{\tau+1}|) \left(\|\Delta_\tau\| + Q + \tilde{Q}\right) \tag{26}$$

500 Next, we focus on providing a bound for $\|\Delta_t\|$. Recall that $\Delta_t = \nabla f(\mathbf{x}_t) - \nabla \tilde{f}(\tilde{\mathbf{x}}_t)$. The gradient
501 can be written as [22],

$$\nabla f(\mathbf{x}_t) = \nabla f(\mathbf{x}_{t-1}) + (\mathbf{x}_t - \mathbf{x}_{t-1}) \left(\int_0^1 \mathbf{H}(\mathbf{x}_{t-1} + v(\mathbf{x}_t - \mathbf{x}_{t-1})) dv\right)$$
$$= \nabla f(\mathbf{x}_{t-1}) + (\mathbf{x}_t - \mathbf{x}_{t-1}) \left(\int_0^1 \left(\mathbf{H}(\mathbf{x}_{t-1} + v(\mathbf{x}_t - \mathbf{x}_{t-1})) + \mathbf{H}(\mathbf{x}_{t-1}) - \mathbf{H}(\mathbf{x}_{t-1})\right) dv\right)$$
$$= \nabla f(\mathbf{x}_{t-1}) + \mathbf{H}(\mathbf{x}_{t-1})(\mathbf{x}_t - \mathbf{x}_{t-1}) + \theta_{t-1},$$

502 where $\theta_{t-1} = \left(\int_0^1 \left(\mathbf{H}(\mathbf{x}_{t-1} + v(\mathbf{x}_t - \mathbf{x}_{t-1})) - \mathbf{H}(\mathbf{x}_{t-1})\right) dv\right) (\mathbf{x}_t - \mathbf{x}_{t-1})$. Let $H'_{t-1} = \mathbf{H}(\mathbf{x}_{t-1}) -$
503 $\mathbf{H}(\mathbf{x}_0)$. Using the SGD-PLRS update,

$$\nabla f(\mathbf{x}_t) = \nabla f(\mathbf{x}_{t-1}) - (H'_{t-1} + \mathbf{H}(\mathbf{x}_0))(\eta_c \nabla f(\mathbf{x}_{t-1}) + \mathbf{w}_{t-1}) + \theta_{t-1}$$
$$= \nabla f(\mathbf{x}_{t-1})(I - \eta_c \mathbf{H}(\mathbf{x}_0)) - \mathbf{H}(\mathbf{x}_0)\mathbf{w}_{t-1} - \eta_c H'_{t-1}\nabla f(\mathbf{x}_{t-1}) - H'_{t-1}\mathbf{w}_{t-1} + \theta_{t-1}, \tag{27}$$

From (14) in the proof of Lemma 2,

$$\nabla \tilde{f}(\tilde{\mathbf{x}}_t) = \nabla \tilde{f}(\tilde{\mathbf{x}}_{t-1}) + \mathbf{H}(\mathbf{x}_0)(\tilde{\mathbf{x}}_t - \tilde{\mathbf{x}}_{t-1}). \tag{28}$$

Subtracting (28) from (27), we obtain $\Delta_t$ as,

$$
\begin{aligned}
\Delta_t &= \nabla f(\mathbf{x}_{t-1})(I - \eta_c \mathbf{H}(\mathbf{x}_0)) - \mathbf{H}(\mathbf{x}_0)\mathbf{w}_{t-1} - \eta_c H'_{t-1}\nabla f(\mathbf{x}_{t-1}) - H'_{t-1}\mathbf{w}_{t-1} + \theta_{t-1} \\
&\qquad\qquad - \nabla \tilde{f}(\tilde{\mathbf{x}}_{t-1}) - \mathbf{H}(\mathbf{x}_0)(\tilde{\mathbf{x}}_t - \tilde{\mathbf{x}}_{t-1}) \\
&= (I - \eta_c \mathbf{H}(\mathbf{x}_0))\Delta_{t-1} - \mathbf{H}(\mathbf{x}_0)\left(\mathbf{w}_{t-1} - \tilde{\mathbf{w}}_{t-1}\right) - H'_{t-1}\left(\eta_c \Delta_{t-1} + \eta_c \nabla \tilde{f}(\tilde{\mathbf{x}}_{t-1})\right) \\
&\qquad\qquad - H'_{t-1}\mathbf{w}_{t-1} + \theta_{t-1},
\end{aligned}
\tag{29}
$$

We now have an expression for $\Delta_t$. However, the derived expression is recursive and contains $\Delta_{t-1}$. We focus on eliminating the recursive dependence and obtain a stand-alone bound for $\|\Delta_t\| \; \forall t \leq T$. Now, we bound each of the five terms (we term them $T_1, \cdots, T_5$) of (29). First, let us define the events,

$$
R_t = \left\{ \forall \tau \leq t, \quad \left\| \nabla \tilde{f}(\tilde{\mathbf{x}}_\tau) \right\| \leq \tilde{O}\left(\frac{1}{\sqrt{L_{max}}}\right), \quad \|\tilde{\mathbf{x}}_\tau - \mathbf{x}_0\| \leq \tilde{O}\left(L_{max}^{3/8} \log\left(\frac{1}{L_{max}}\right)\right) \right\}
$$

$$
C_t = \left\{ \forall \tau \leq t, \quad \|\Delta_\tau\| \leq \mu L_{max}^{3/8} \log\left(\frac{1}{L_{max}}\right) \right\}.
$$

It can be seen that $R_t \subset R_{t-1}$ and $C_t \subset C_{t-1}$. Note that, from Lemma 2, we know the probabilistic characterization of $R_t$. We comment on the parameter $\mu$ later in the proof. Now, we derive bounds for each term of $\Delta_t$ *conditioned* on the event $R_{t-1} \cap C_{t-1}$ for time $t \leq T = O\left(L_{max}^{-1/4}\right)$.

$$
\begin{aligned}
T_1 : \quad \|(I - \eta_c \mathbf{H}(\mathbf{x}_0))\Delta_{t-1}\| &\leq \|\Delta_{t-1}\| + \|-\eta_c \mathbf{H}(\mathbf{x}_0)\Delta_{t-1}\| \\
&\leq \mu L_{max}^{3/8} \log\left(\frac{1}{L_{max}}\right) + \tilde{O}\left(\mu L_{max}^{11/8} \log\left(\frac{1}{L_{max}}\right)\right) \\
&= \tilde{O}\left(\mu L_{max}^{3/8} \log\left(\frac{1}{L_{max}}\right)\right),
\end{aligned}
\tag{30}
$$

where (30) follows from the definition of event $C_{t-1}$. Note that the first term in (30) governs the order of the expression (as $0 \leq L_{max} \leq 1$).

$$
\begin{aligned}
T_2 : \quad \|\mathbf{H}(\mathbf{x}_0)\left(\mathbf{w}_{t-1} - \tilde{\mathbf{w}}_{t-1}\right)\| &\leq \|\mathbf{H}(\mathbf{x}_0)\| \, \|\mathbf{w}_{t-1} - \tilde{\mathbf{w}}_{t-1}\| \\
&\leq \|\mathbf{H}(\mathbf{x}_0)\| \left( |u_{\tau+1}| + \eta_c| \left(Q + \tilde{Q}\right) + |u_{\tau+1}| \, \|\Delta_\tau\| \right) \\
&\leq \tilde{O}(L_{max}) + \tilde{O}\left(\mu L_{max}^{11/8} \log\left(\frac{1}{L_{max}}\right)\right) = \tilde{O}(L_{max}),
\end{aligned}
$$

where the substitution follows from (25). To bound $T_3$ and $T_4$, we first bound $H'_{t-1}$,

$$
\left\| H'_{t-1} \right\| = \|\mathbf{H}(\mathbf{x}_{t-1}) - \mathbf{H}(\mathbf{x}_0)\| \leq \rho \|\mathbf{x}_{t-1} - \mathbf{x}_0\| \tag{31a}
$$

$$
\leq \rho \left( \|\mathbf{x}_{t-1} - \tilde{\mathbf{x}}_{t-1}\| + \|\tilde{\mathbf{x}}_{t-1} - \mathbf{x}_0\| \right)
$$

$$
\leq \rho \left( \sum_{\tau=0}^{t-1} (\eta_c + |u_{\tau+1}|) \left( \|\Delta_\tau\| + Q + \tilde{Q} \right) \right) + \rho\tilde{O}\left( L_{max}^{3/8} \log\frac{1}{L_{max}} \right) \tag{31b}
$$

$$
= \tilde{O}\left(\frac{1}{L_{max}^{1/4}}\right) \tilde{O}\left(\mu L_{max}^{11/8} \log\frac{1}{L_{max}}\right) + \tilde{O}\left(\frac{1}{L_{max}^{1/4}}\right) \tilde{O}(L_{max}) + \tilde{O}\left( L_{max}^{3/8} \log\frac{1}{L_{max}} \right) \tag{31c}
$$

$$
\leq \tilde{O}(L_{max}^{3/4}) + \tilde{O}\left( L_{max}^{3/8} \log\frac{1}{L_{max}} \right) \leq \tilde{O}\left( L_{max}^{3/8} \log\frac{1}{L_{max}} \right), \tag{31d}
$$

where (31a) follows from the assumption **A2** while (31b) follows from (26). We use the bounds defined for events $R_{t-1} \cap C_{t-1}$ in (31b) and (31c). Now, using the bound for $\left\| H'_{t-1} \right\|$, $T_3$ can be

bounded as follows.

$$T_3: \quad \left\| H'_{t-1} \eta_c (\Delta_{t-1} + \nabla \tilde{f}(\tilde{\mathbf{x}}_{t-1})) \right\| \leq \eta_c \left\| H'_{t-1} \Delta_{t-1} \right\| + \eta_c \left\| H'_{t-1} \nabla \tilde{f}(\tilde{\mathbf{x}}_{t-1}) \right\|$$

$$\leq O(L_{max}) \tilde{O} \left( L_{max}^{3/8} \log \frac{1}{L_{max}} \right) \mu L_{max}^{3/8} \log \frac{1}{L_{max}}$$

$$+ O(L_{max}) \tilde{O} \left( L_{max}^{3/8} \log \frac{1}{L_{max}} \right) \tilde{O} \left( \frac{1}{\sqrt{L_{max}}} \right)$$

$$= \tilde{O} \left( L_{max}^{7/8} \log \frac{1}{L_{max}} \right),$$

where we use the bounds in the event $R_{t-1} \cap C_{t-1}$ and (31d).

$$T_4: \quad \left\| H'_{t-1} \mathbf{w}_{t-1} \right\| \leq \left\| H'_{t-1} \right\| \|\mathbf{w}_{t-1}\| = \left\| H'_{t-1} \right\| \|\eta_c g(\mathbf{x}_{t-1}) - \eta_c \nabla f(\mathbf{x}_{t-1} + u_t g(\mathbf{x}_t))\|$$

$$\leq \left\| H'_{t-1} \right\| (\eta_c Q + |u_t| Q + |u_t| \|\nabla f(\mathbf{x}_{t-1})\|) \tag{32a}$$

$$= (\eta_c + |u_t|) Q \left\| H'_{t-1} \right\| + |u_t| \left\| H'_{t-1} \right\| \|\Delta_{t-1}\| + |u_t| \left\| H'_{t-1} \right\| \left\| \nabla \tilde{f}(\tilde{\mathbf{x}}_{t-1}) \right\|$$

$$= \tilde{O} \left( L_{max}^{11/8} \log \frac{1}{L_{max}} \right) + \tilde{O} \left( \mu L_{max}^{14/8} \log^2 \frac{1}{L_{max}} \right) + \tilde{O} \left( L_{max}^{7/8} \log \frac{1}{L_{max}} \right) \tag{32b}$$

$$= \tilde{O} \left( L_{max}^{7/8} \log \frac{1}{L_{max}} \right),$$

where we use assumption **A3** in (32a) and the bounds of $R_{t-1} \cap C_{t-1}$ and (31d) in (32b).

$$T_5: \quad \|\theta_{t-1}\| = \left\| \left( \int_0^1 \left( \mathbf{H}(\mathbf{x}_{t-1} + v(\mathbf{x}_t - \mathbf{x}_{t-1})) - \mathbf{H}(\mathbf{x}_{t-1}) \right) dv \right) (\mathbf{x}_t - \mathbf{x}_{t-1}) \right\|$$

$$\leq \left( \int_0^1 \rho \|\mathbf{x}_{t-1} + v(\mathbf{x}_t - \mathbf{x}_{t-1}) - \mathbf{x}_{t-1}\| \, dv \right) \|\mathbf{x}_t - \mathbf{x}_{t-1}\| \tag{33a}$$

$$\leq \frac{\rho}{2} \|\mathbf{x}_t - \mathbf{x}_{t-1}\|^2 \leq \frac{\rho}{2} \|-\eta_c \nabla f(\mathbf{x}_{t-1}) - \mathbf{w}_{t-1}\|^2$$

$$\leq \frac{\rho}{2} \|-\eta_c \nabla f(\mathbf{x}_{t-1}) - \eta_c g(\mathbf{x}_{t-1}) + \eta_c \nabla f(\mathbf{x}_{t-1}) - u_t g(\mathbf{x}_{t-1})\|^2$$

$$\leq \frac{\rho |\eta_c + u_t|^2}{2} \left( Q^2 + \|\nabla f(\mathbf{x}_{t-1})\|^2 + 2Q \|\nabla f(\mathbf{x}_{t-1})\| \right)$$

$$= \frac{\rho |\eta_c + u_t|^2}{2} \left( Q^2 + \|\Delta_{t-1}\|^2 + \left\| \nabla \tilde{f}(\tilde{\mathbf{x}}_{t-1}) \right\|^2 + 2 \|\Delta_{t-1}\| \left\| \nabla \tilde{f}(\tilde{\mathbf{x}}_{t-1}) \right\| \right.$$

$$\left. + 2Q \|\Delta_{t-1}\| + 2Q \left\| \nabla \tilde{f}(\tilde{\mathbf{x}}_{t-1}) \right\| \right)$$

$$= \tilde{O}(L_{max}^2) + \tilde{O} \left( \mu^2 L_{max}^{11/4} \log^2 \frac{1}{L_{max}} \right) + \tilde{O}(L_{max}) + \tilde{O} \left( \mu L_{max}^{15/8} \log \frac{1}{L_{max}} \right)$$

$$+ \tilde{O} \left( \mu L_{max}^{19/8} \log \frac{1}{L_{max}} \right) + \tilde{O}(L_{max}^{3/2}) = \tilde{O}(L_{max}). \tag{33b}$$

Here, we use assumption **A3** and the bounds of the event $R_{t-1} \cap C_{t-1}$ in (33b). Note that we have derived bounds so far conditioned on the event $R_{t-1} \cap C_{t-1}$. We now include this conditioning explicitly in our notations going forward.

To characterize $\|\Delta_t\|^2$, we construct a supermartingale process; and to do so, we focus on finding $\mathbb{E}[\|\Delta_t\|^2 \mathbf{1}_{R_{t-1} \cap C_{t-1}}]$ using the bounds derived for the terms $T_1, \cdots, T_5$. Later, we use the Azuma-

Hoeffding inequality to obtain a probabilistic bound of $\|\Delta_t\|$.

$$\mathbb{E}[\|\Delta_t\|^2\, \mathbf{1}_{R_{t-1}\cap C_{t-1}}|S_{t-1}] \leq \left[(1+\eta_c\gamma_o)^2\,\|\Delta_{t-1}\|^2 + \tilde{O}\left(\mu L_{max}^{3/8}\log\frac{1}{L_{max}}\right)\tilde{O}\left(L_{max}^{7/8}\log\frac{1}{L_{max}}\right)\right.$$
$$+ \tilde{O}\left(\mu L_{max}^{3/8}\log\frac{1}{L_{max}}\right)\tilde{O}(L_{max}) + \tilde{O}(L_{max}^2)$$
$$\left.+ \tilde{O}\left(L_{max}^{7/8}\log\frac{1}{L_{max}}\right)\tilde{O}(L_{max}) + \tilde{O}\left(L_{max}^{7/4}\log^2\frac{1}{L_{max}}\right)\right]\mathbf{1}_{R_{t-1}\cap C_{t-1}}$$
$$\leq \left[(1+\eta_c\gamma_o)^2\,\|\Delta_{t-1}\|^2 + \tilde{O}\left(\mu L_{max}^{7/8}\log\frac{1}{L_{max}}\right)\right]\mathbf{1}_{R_{t-1}\cap C_{t-1}} \tag{34}$$

Now, let

$$G_t = (1+\eta_c\gamma_o)^{-2t}\left[\|\Delta_t\|^2 + \tilde{O}\left(\mu L_{max}^{7/8}\log\frac{1}{L_{max}}\right)\right]. \tag{35}$$

Now, in order to prove the process $G_t\mathbf{1}_{R_{t-1}\cap C_{t-1}}$ is a supermartingale, we prove that $\mathbb{E}[G_t\mathbf{1}_{R_{t-1}\cap C_{t-1}}|S_{t-1}] \leq G_{t-1}\mathbf{1}_{R_{t-2}\cap C_{t-2}}$. We define a filtration $S_t = s\{\mathbf{w}_0,\ldots,\mathbf{w}_{t-1}\}$ where $s\{.\}$ denotes a sigma-algebra field.

$$\mathbb{E}[G_t\mathbf{1}_{R_{t-1}\cap C_{t-1}}|S_{t-1}]$$
$$\leq (1+\eta_c\gamma_o)^{-2t}\left((1+\eta_c\gamma_o)^2\,\|\Delta_{t-1}\|^2 + 2\tilde{O}\left(\mu L_{max}^{7/8}\log\frac{1}{L_{max}}\right)\right)\mathbf{1}_{R_{t-1}\cap C_{t-1}} \tag{36a}$$
$$\leq (1+\eta_c\gamma_o)^{-2t}\left((1+\eta_c\gamma_o)^2\,\|\Delta_{t-1}\|^2 + 2(1+\eta_c\gamma_o)^2\tilde{O}\left(\mu L_{max}^{7/8}\log\frac{1}{L_{max}}\right)\right)\mathbf{1}_{R_{t-1}\cap C_{t-1}} \tag{36b}$$
$$= (1+\eta_c\gamma_o)^{-2(t-1)}\left(\|\Delta_{t-1}\|^2 + \tilde{O}\left(\mu L_{max}^{7/8}\log\frac{1}{L_{max}}\right)\right)\mathbf{1}_{R_{t-1}\cap C_{t-1}}$$
$$= G_{t-1}\mathbf{1}_{R_{t-1}\cap C_{t-1}} \leq G_{t-1}\mathbf{1}_{R_{t-2}\cap C_{t-2}}.$$

To obtain (36a), we use (34) to find $\mathbb{E}[G_t\mathbf{1}_{R_{t-1}\cap C_{t-1}}|S_{t-1}]$. In (36b), we upper bound by the multiplication of a positive term $(1+\eta_c\gamma_o)^2$. Therefore, $G_t\mathbf{1}_{R_{t-1}\cap C_{t-1}}$ is a supermartingale.

$$\|\Delta_t\|^2 - \mathbb{E}[\|\Delta_t\|^2|S_{t-1}]\mathbf{1}_{R_{t-1}\cap C_{t-1}} \leq -2\,\|(I-\eta_c\mathbf{H}(\mathbf{x}_0))\Delta_{t-1}\|\,\|\mathbf{H}(\mathbf{x}_0)(\mathbf{w}_{t-1}-\tilde{\mathbf{w}}_{t-1})\|$$
$$- 2\,\|(I-\eta_c\mathbf{H}(\mathbf{x}_0))\Delta_{t-1}\|\,\left\|H_{t-1}^{'}\mathbf{w}_{t-1}\right\| + 2\,\|(I-\eta_c\mathbf{H}(\mathbf{x}_0))\Delta_{t-1}\|\,\|\theta_{t-1}\|$$
$$+ \|\mathbf{H}(\mathbf{x}_0)(\mathbf{w}_{t-1}-\tilde{\mathbf{w}}_{t-1})\|^2 + \left\|H_{t-1}^{'}\mathbf{w}_{t-1}\right\|^2 + 2\,\|\mathbf{H}(\mathbf{x}_0)(\mathbf{w}_{t-1}-\tilde{\mathbf{w}}_{t-1})\|\,\left\|H_{t-1}^{'}\mathbf{w}_{t-1}\right\|$$
$$+ 2\,\|\mathbf{H}(\mathbf{x}_0)(\mathbf{w}_{t-1}-\tilde{\mathbf{w}}_{t-1})\|\,\left\|H_{t-1}^{'}(\eta_c\Delta_{t-1}+\eta_c\nabla\tilde{f}(\tilde{\mathbf{x}}_{t-1}))\right\|$$
$$- 2\,\|\mathbf{H}(\mathbf{x}_0)(\mathbf{w}_{t-1}-\tilde{\mathbf{w}}_{t-1})\|\,\|\theta_{t-1}\| + 2\,\left\|H_{t-1}^{'}(\eta_c\Delta_{t-1}+\eta_c\nabla\tilde{f}(\tilde{\mathbf{x}}_{t-1}))\right\|\,\left\|H_{t-1}^{'}\mathbf{w}_{t-1}\right\|$$
$$- 2\,\left\|H_{t-1}^{'}(\eta_c\Delta_{t-1}+\eta_c\nabla\tilde{f}(\tilde{\mathbf{x}}_{t-1}))\right\|\,\|\theta_{t-1}\| - 2\,\left\|H_{t-1}^{'}\mathbf{w}_{t-1}\right\|\,\|\theta_{t-1}\| + \|\theta_{t-1}\|^2$$
$$= \tilde{O}\left(\mu L_{max}^{11/8}\log\frac{1}{L_{max}}\right) + \tilde{O}\left(\mu L_{max}^{10/8}\log^2\frac{1}{L_{max}}\right) + \tilde{O}(L_{max}^2) + \tilde{O}\left(L_{max}^{15/8}\log\frac{1}{L_{max}}\right)$$
$$+ \tilde{O}\left(L_{max}^{7/4}\log^2\frac{1}{L_{max}}\right) \leq \tilde{O}\left(\mu L_{max}^{7/8}\log\frac{1}{L_{max}}\right)$$

Note that the above expression is obtained by the observation that the only random terms of $\Delta_t$ conditioned on the filtration $S_{t-1} = s\{\mathbf{w}_0,\mathbf{w}_1,\ldots,\mathbf{w}_{t-2}\}$ are $\mathbf{H}(\mathbf{x}_0)(\mathbf{w}_{t-1}-\tilde{\mathbf{w}}_{t-1})$, $H_{t-1}^{'}\mathbf{w}_{t-1}$ and $\theta_{t-1}$(see (33a)). Hence, we cancel out the deterministic terms in $\|\Delta_t\|^2$ and $\mathbb{E}\|\Delta_t\|^2$ and neglect the negative terms while upper bounding.

The Azuma-Hoeffding inequality for martingales and supermartingales [11] states that if $\{G_t\mathbf{1}_{R_{t-1}\cap C_{t-1}}\}$ is a supermartingale and $|G_t\mathbf{1}_{R_{t-1}\cap C_{t-1}} - G_{t-1}\mathbf{1}_{R_{t-2}\cap C_{t-2}}| \leq c_t$ almost surely,

then for all positive integers $t$ and positive reals $\delta$,

$$\mathbb{P}(G_t \mathbf{1}_{R_{t-1} \cap C_{t-1}} - G_0 \mathbf{1}_{R_{-1} \cap C_{-1}} \geq \delta) \leq \exp\left(-\frac{\delta^2}{2 \sum_{\tau=0}^{t-1} c_\tau^2}\right).$$

The bound of $|G_t \mathbf{1}_{R_{t-1} \cap C_{t-1}} - G_{t-1} \mathbf{1}_{R_{t-2} \cap C_{t-2}}|$ can be obtained using the definition of the process $G_t$ in (35). Recollecting our assumption that $\eta_c \leq \eta_{max} \leq \frac{\sqrt{2}-1}{\gamma'}, \gamma_o \leq \gamma'$, we see that $(1 + \eta_c \gamma_o)^{-2t} \leq \tilde{O}(1)$. Therefore,

$$|G_t \mathbf{1}_{R_{t-1} \cap C_{t-1}} - \mathbb{E}[G_t \mathbf{1}_{R_{t-1} \cap C_{t-1}} | S_{t-1}]| = (1 + \eta_c \gamma_o)^{-2t} \left| \|\Delta_t\|^2 - \mathbb{E}[\|\Delta_t\|^2 | S_{t-1}] \right| \mathbf{1}_{R_{t-1} \cap C_{t-1}}$$

$$\leq \tilde{O}\left(\mu L_{max}^{7/8} \log \frac{1}{L_{max}}\right).$$

We denote the bound obtained for $|G_t \mathbf{1}_{R_{t-1} \cap C_{t-1}} - \mathbb{E}[G_t \mathbf{1}_{R_{t-1} \cap C_{t-1}} | S_{t-1}]|$ as $c_{t-1}$. Now, let $\delta = \sqrt{\sum_{\tau=0}^{t-1} c_\tau^2 \log \frac{1}{L_{max}}}$ in the Azuma-Hoeffding inequality. Now, for any $t \leq T = O\left(L_{max}^{-1/4}\right)$,

$$\delta = \sqrt{O\left(\frac{1}{L_{max}^{1/4}}\right) \tilde{O}\left(\mu^2 L_{max}^{7/4} \log^2 \frac{1}{L_{max}}\right) \log \frac{1}{L_{max}}} = \tilde{O}\left(\mu L_{max}^{3/4} \log^2 \frac{1}{L_{max}}\right).$$

$$\mathbb{P}\left(G_t \mathbf{1}_{R_{t-1} \cap C_{t-1}} - G_0.1 \geq \tilde{O}\left(\mu L_{max}^{3/4} \log^2 \frac{1}{L_{max}}\right)\right) \leq \exp\left(-\tilde{\Omega}\left(\log^2 \frac{1}{L_{max}}\right)\right)$$

$$\leq \tilde{O}(L_{max}^4).$$

After taking union bound $\forall\, t \leq T$,

$$\mathbb{P}\left(\forall\, t \leq T,\ G_t \mathbf{1}_{R_{t-1} \cap C_{t-1}} - G_0 \geq \tilde{O}\left(\mu L_{max}^{3/4} \log^2 \frac{1}{L_{max}}\right)\right) \leq \tilde{O}(L_{max}^{15/4}).$$

We represent the hidden constants in $\tilde{O}\left(\mu L_{max}^{3/4} \log^2 \frac{1}{L_{max}}\right)$ by $\tilde{c}$ and choose $\mu$ such that $\mu < \tilde{c}$. Then, the following equation holds true.

$$\mathbb{P}\left(G_t \mathbf{1}_{R_{t-1} \cap C_{t-1}} - G_0 \geq \mu^2 L_{max}^{3/4} \log^2 \frac{1}{L_{max}}\right) \leq \tilde{O}(L_{max}^{15/4}).$$

Hence we can write,

$$\mathbb{P}\left(R_{t-1} \cap C_{t-1} \cap \left\{\|\Delta_t\| \geq \mu L_{max}^{3/8} \log \frac{1}{L_{max}}\right\}\right) \leq \tilde{O}(L_{max}^{15/4}). \tag{37}$$

We need the probability of the event $C_t, \forall t \leq T$ in order to prove the lemma. From Lemma 2, we get the probability of the event $\bar{R}_t$ as $\tilde{O}(L_{max}^{15/4})$. Then,

$$\mathbb{P}\left(C_{t-1} \cap \left\{\|\Delta_t\| \geq \mu L_{max}^{3/8} \log \frac{1}{L_{max}}\right\}\right) = \mathbb{P}\left(R_{t-1} \cap C_{t-1} \cap \left\{\|\Delta_t\| \geq \mu L_{max}^{3/8} \log \frac{1}{L_{max}}\right\}\right)$$

$$+ \mathbb{P}\left(\bar{R}_{t-1} \cap C_{t-1} \cap \left\{\|\Delta_t\| \geq \mu L_{max}^{3/8} \log \frac{1}{L_{max}}\right\}\right)$$

$$\leq \tilde{O}(L_{max}^{15/4}) + \mathbb{P}(\bar{R}_{t-1}) \leq \tilde{O}(L_{max}^{15/4}),$$

$$\tag{38}$$

where the first term of (38) follows from (37). The second term of (38) can be bounded by $\mathbb{P}(\bar{R}_{t-1})$ which is known by Lemma 2. Finally,

$$\mathbb{P}(\bar{C}_t) = \mathbb{P}\left(C_{t-1} \cap \left\{\|\Delta_t\| \geq \mu L_{max}^{3/8} \log \frac{1}{L_{max}}\right\}\right) + \mathbb{P}(\bar{C}_{t-1}) \leq \tilde{O}(L_{max}^{15/4}) + \mathbb{P}(\bar{C}_{t-1}).$$

The probability $\mathbb{P}(\bar{C}_{t-1})$ can be found as,

$$\mathbb{P}(\bar{C}_{t-1}) = \mathbb{P}\left(C_{t-2} \cap \left\{\|\Delta_{t-1}\| \geq \mu L_{max}^{3/8} \log \frac{1}{L_{max}}\right\}\right) + \mathbb{P}(\bar{C}_{t-2})$$

$$= \mathbb{P}\left(C_{t-2} \cap \left\{\|\Delta_{t-1}\| \geq \mu L_{max}^{3/8} \log \frac{1}{L_{max}}\right\}\right) + \dots$$

$$+ \mathbb{P}\left(C_0 \cap \left\{\|\Delta_1\| \geq \mu L_{max}^{3/8} \log \frac{1}{L_{max}}\right\}\right) + \mathbb{P}(\bar{C}_0).$$

As $T = O\left(L_{max}^{-1/4}\right)$, $\mathbb{P}(\bar{C}_T) \le \tilde{O}\left(L_{max}^{7/2}\right)$. From (26),

$$\|\mathbf{x}_t - \tilde{\mathbf{x}}_t\| \le \sum_{\tau=0}^{t-1} (\eta_c + |u_{\tau+1}|)\left(\|\Delta_\tau\| + Q + \tilde{Q}\right)$$

$$\le O\left(\frac{1}{L_{max}^{1/4}}\right)\left(\tilde{O}(L_{max})\mu L_{max}^{3/8}\log\frac{1}{L_{max}} + \tilde{O}(L_{max})\right)$$

$$= O\left(\mu L_{max}^{9/8}\log\frac{1}{L_{max}}\right) + \tilde{O}(L_{max}^{3/4}) \le \tilde{O}(L_{max}^{3/4})$$

This completes our proof. $\qquad\square$

## C  Proof of Theorem 2

**Theorem 5.** *(Theorem 2 restated) Consider $f$ satisfying Assumptions **A1** - **A5**. Let $\tilde{f}$ be the second order Taylor approximation of $f$; let $\{\boldsymbol{x}_t\}$ and $\{\tilde{\boldsymbol{x}}_t\}$ be the corresponding SGD iterates using PLRS, with $\tilde{\boldsymbol{x}}_0 = \boldsymbol{x}_0$. Let $\boldsymbol{x}_0$ correspond to **B2**, i.e., $\|\nabla f(\boldsymbol{x}_0)\| \le \epsilon$ and $\lambda_{min}(\boldsymbol{H}(\boldsymbol{x}_0)) \le -\gamma$ where $\epsilon, \gamma > 0$. Then, there exists a $T = \tilde{O}\left(L_{max}^{-1/4}\right)$ such that with probability at least $1 - \tilde{O}\left(L_{max}^{7/2}\right)$,*

$$\mathbb{E}[f(\boldsymbol{x}_T) - f(\boldsymbol{x}_0)] \le -\tilde{\Omega}\left(L_{max}^{3/4}\right).$$

*Proof.* In this proof, we consider the case when the initial iterate $\mathbf{x}_0$ is at a saddle point (corresponding to **B2**). This theorem shows that the SGD-PLRS algorithm escapes the saddle point in $T$ steps where $T = \tilde{O}\left(L_{max}^{-1/4}\right)$.

We use the Taylor series approximation in order to make the problem tractable. Similar to the SGD-PLRS updates for the function $f$, the SGD update on the function $\tilde{f}$ can be given as,

$$\tilde{\mathbf{x}}_t = \tilde{\mathbf{x}}_{t-1} - \eta_c\nabla\tilde{f}(\tilde{\mathbf{x}}_{t-1}) - \tilde{\mathbf{w}}_{t-1}, \quad \tilde{\mathbf{w}}_{t-1} = \eta_c\tilde{g}(\tilde{\mathbf{x}}_{t-1}) - \eta_c\nabla\tilde{f}(\tilde{\mathbf{x}}_{t-1}) + u_t\tilde{g}(\tilde{\mathbf{x}}_{t-1}).$$

As the function $f$ is $\rho$-Hessian, using [22, Lemma 1.2.4] and the Taylor series expansion one obtains, $f(\mathbf{x}) \le f(\mathbf{x}_0) + \nabla f(\mathbf{x}_0)^T(\mathbf{x} - \mathbf{x}_0) + \frac{1}{2}(\mathbf{x} - \mathbf{x}_0)^T\mathbf{H}(\mathbf{x}_0)(\mathbf{x} - \mathbf{x}_0) + \frac{\rho}{6}\|\mathbf{x} - \mathbf{x}_0\|^3$. Let $\tilde{\boldsymbol{\kappa}} = \tilde{\mathbf{x}}_T - \mathbf{x}_0$, $\boldsymbol{\kappa} = \mathbf{x}_T - \tilde{\mathbf{x}}_T$. Note that $\tilde{\boldsymbol{\kappa}} + \boldsymbol{\kappa} = \mathbf{x}_T - \mathbf{x}_0$. Then, replacing $\mathbf{x}$ by $\mathbf{x}_T$,

$$f(\mathbf{x}_T) - f(\mathbf{x}_0) \le \nabla f(\mathbf{x}_0)^T(\mathbf{x}_T - \mathbf{x}_0) + \frac{1}{2}(\mathbf{x}_T - \mathbf{x}_0)^T\mathbf{H}(\mathbf{x}_0)(\mathbf{x}_T - \mathbf{x}_0) + \frac{\rho}{6}\|\mathbf{x}_T - \mathbf{x}_0\|^3$$

$$= \nabla f(\mathbf{x}_0)^T(\tilde{\boldsymbol{\kappa}} + \boldsymbol{\kappa}) + \frac{1}{2}(\tilde{\boldsymbol{\kappa}} + \boldsymbol{\kappa})^T\mathbf{H}(\mathbf{x}_0)(\tilde{\boldsymbol{\kappa}} + \boldsymbol{\kappa}) + \frac{\rho}{6}\|\tilde{\boldsymbol{\kappa}} + \boldsymbol{\kappa}\|^3$$

$$= \left(\nabla f(\mathbf{x}_0)^T\tilde{\boldsymbol{\kappa}} + \frac{1}{2}\tilde{\boldsymbol{\kappa}}^T\mathbf{H}(\mathbf{x}_0)\tilde{\boldsymbol{\kappa}}\right) + \left(\nabla f(\mathbf{x}_0)^T\boldsymbol{\kappa} + \tilde{\boldsymbol{\kappa}}^T\mathbf{H}(\mathbf{x}_0)\boldsymbol{\kappa} + \frac{1}{2}\boldsymbol{\kappa}^T\mathbf{H}(\mathbf{x}_0)\boldsymbol{\kappa}\right.$$

$$\left. + \frac{\rho}{6}\|\tilde{\boldsymbol{\kappa}} + \boldsymbol{\kappa}\|^3\right).$$

Let the first term be $\tilde{\zeta} = \nabla f(\mathbf{x}_0)^T\tilde{\boldsymbol{\kappa}} + \frac{1}{2}\tilde{\boldsymbol{\kappa}}^T\mathbf{H}(\mathbf{x}_0)\tilde{\boldsymbol{\kappa}}$ and the second term be $\zeta = \nabla f(\mathbf{x}_0)^T\boldsymbol{\kappa} + \tilde{\boldsymbol{\kappa}}^T\mathbf{H}(\mathbf{x}_0)\boldsymbol{\kappa} + \frac{1}{2}\boldsymbol{\kappa}^T\mathbf{H}(\mathbf{x}_0)\boldsymbol{\kappa} + \frac{\rho}{6}\|\tilde{\boldsymbol{\kappa}} + \boldsymbol{\kappa}\|^3$. Hence $f(\mathbf{x}_T) - f(\mathbf{x}_0) \le \tilde{\zeta} + \zeta$. In order to prove the theorem, we require an upper bound on $\mathbb{E}[f(\mathbf{x}_T) - f(\mathbf{x}_0)]$.

Now, we introduce two mutually exclusive events $C_t$ and $\bar{C}_t$ so that $\mathbb{E}[f(\mathbf{x}_T) - f(\mathbf{x}_0)]$ can be written in terms of events $C_t$ and $\bar{C}_t$ as,

$$\mathbb{E}[f(\mathbf{x}_T) - f(\mathbf{x}_0)] = \mathbb{E}[f(\mathbf{x}_T) - f(\mathbf{x}_0)](\mathbb{E}[\mathbf{1}_{C_T}] + \mathbb{E}[\mathbf{1}_{\bar{C}_T}])$$

$$= \mathbb{E}[(f(\mathbf{x}_T) - f(\mathbf{x}_0))\mathbf{1}_{C_T}] + \mathbb{E}[(f(\mathbf{x}_T) - f(\mathbf{x}_0))\mathbf{1}_{\bar{C}_T}]$$

$$\le \mathbb{E}[\tilde{\zeta}\mathbf{1}_{C_T}] + \mathbb{E}[\zeta\mathbf{1}_{C_T}] + \mathbb{E}[(f(\mathbf{x}_T) - f(\mathbf{x}_0))\mathbf{1}_{\bar{C}_T}]$$

$$= \mathbb{E}[\tilde{\zeta}] + \mathbb{E}[\zeta\mathbf{1}_{C_T}] + \mathbb{E}[(f(\mathbf{x}_T) - f(\mathbf{x}_0))\mathbf{1}_{\bar{C}_T}] - \mathbb{E}[\tilde{\zeta}\mathbf{1}_{\bar{C}_T}].$$

Let $K_1 = \mathbb{E}[\tilde{\zeta}]$, $K_2 = \mathbb{E}[\zeta\mathbf{1}_{C_T}]$ and $K_3 = \mathbb{E}[(f(\mathbf{x}_T) - f(\mathbf{x}_0))\mathbf{1}_{\bar{C}_T}] - \mathbb{E}[\tilde{\zeta}\mathbf{1}_{\bar{C}_T}]$. In the remainder of the proof, we focus on deriving the bounds for individual terms, $K_1$, $K_2$ and $K_3$, and then finally put them together to obtain the result of the theorem.

## C.1 Bounding $K_1$

Using (20b) from the proof of Lemma 2 in Appendix B.1, we obtain the bound for the term $K_1 = \mathbb{E}[\tilde{\zeta}]$ as,

$$\mathbb{E}[\tilde{\zeta}] = \mathbb{E}\left[\nabla f(\mathbf{x}_0)^T(\tilde{\mathbf{x}}_T - \mathbf{x}_0) + \frac{1}{2}(\tilde{\mathbf{x}}_T - \mathbf{x}_0)^T\mathbf{H}(\mathbf{x}_0)(\tilde{\mathbf{x}}_T - \mathbf{x}_0)\right]$$

$$= \mathbb{E}\left[\nabla f(\mathbf{x}_0)^T\left(-\sum_{\tau=0}^{T-1}\eta_c(I - \eta_c\mathbf{H}(\mathbf{x}_0))^\tau\nabla f(\mathbf{x}_0) - \sum_{\tau=0}^{T-1}(I - \eta_c\mathbf{H}(\mathbf{x}_0))^{T-\tau-1}\tilde{\mathbf{w}}_\tau\right)\right]$$

$$+ \frac{1}{2}\mathbb{E}\left[\left(-\sum_{\tau=0}^{T-1}\eta_c(I - \eta_c\mathbf{H}(\mathbf{x}_0))^\tau\nabla f(\mathbf{x}_0) - \sum_{\tau=0}^{T-1}(I - \eta_c\mathbf{H}(\mathbf{x}_0))^{T-\tau-1}\tilde{\mathbf{w}}_\tau\right)^T\mathbf{H}(\mathbf{x}_0)\right.$$

$$\left.\left(-\sum_{\tau=0}^{T-1}\eta_c(I - \eta_c\mathbf{H}(\mathbf{x}_0))^\tau\nabla f(\mathbf{x}_0) - \sum_{\tau=0}^{T-1}(I - \eta_c\mathbf{H}(\mathbf{x}_0))^{T-\tau-1}\tilde{\mathbf{w}}_\tau\right)\right].$$

Since $\tilde{\mathbf{w}}_\tau = \mathbf{0}$, all the terms with $\mathbb{E}[\tilde{\mathbf{w}}_\tau]$ will go to zero. Hence we obtain,

$$\mathbb{E}[\tilde{\zeta}] = \nabla f(\mathbf{x}_0)^T\left(-\sum_{\tau=0}^{T-1}\eta_c(I - \eta_c\mathbf{H}(\mathbf{x}_0))^\tau\nabla f(\mathbf{x}_0)\right) +$$

$$\frac{1}{2}\left(-\sum_{\tau=0}^{T-1}\eta_c(I - \eta_c\mathbf{H}(\mathbf{x}_0))^\tau\nabla f(\mathbf{x}_0)\right)^T\mathbf{H}(\mathbf{x}_0)\left(-\sum_{\tau=0}^{T-1}\eta_c(I - \eta_c\mathbf{H}(\mathbf{x}_0))^\tau\nabla f(\mathbf{x}_0)\right)$$

$$+ \frac{1}{2}\mathbb{E}\left[\left(-\sum_{\tau=0}^{T-1}(I - \eta_c\mathbf{H}(\mathbf{x}_0))^{T-\tau-1}\tilde{\mathbf{w}}_\tau\right)^T\mathbf{H}(\mathbf{x}_0)\left(-\sum_{\tau=0}^{T-1}(I - \eta_c\mathbf{H}(\mathbf{x}_0))^{T-\tau-1}\tilde{\mathbf{w}}_\tau\right)\right].$$

Let $\lambda_1, \ldots, \lambda_d$ be the eigenvalues of the Hessian matrix at $\mathbf{x}_0$, $\mathbf{H}(\mathbf{x}_0)$. Now, we simplify similar to Ge et al. [9] as,

$$\mathbb{E}[\tilde{\zeta}] = -\sum_{i=1}^{d}\sum_{\tau=0}^{T-1}\eta_c(1 - \eta_c\lambda_i)^\tau|\nabla_i f(\mathbf{x}_0)|^2 + \frac{1}{2}\sum_{i=1}^{d}\lambda_i\sum_{\tau=0}^{T-1}\eta_c^2(1 - \eta_c\lambda_i)^{2\tau}|\nabla_i f(\mathbf{x}_0)|^2$$

$$+ \frac{1}{2}\sum_{i=1}^{d}\lambda_i\sum_{\tau=0}^{T-1}(1 - \eta_c\lambda_i)^{2(T-\tau-1)}\mathbb{E}[|\tilde{\mathbf{w}}_{\tau,i}|^2].$$

Note that for the case of very small gradients (as per our initial conditions), $|\nabla_i f(\mathbf{x}_0)|^2 \leq \|\nabla f(\mathbf{x}_0)\| \leq \epsilon$. Therefore, the first and second terms can be made arbitrarily small so that they do not contribute to the order of the equation. Hence, we focus on the third term. We first characterize $\mathbb{E}[|\tilde{\mathbf{w}}_{\tau,i}|^2]$ as follows. Since the norm of the stochastic noise is bounded as per the assumption **A3**, we assume that $\tilde{g}_i(\tilde{\mathbf{x}}_t) - \nabla_i\tilde{f}(\tilde{\mathbf{x}}_t) \leq \tilde{q}$ and $\mathbb{E}[\tilde{q}] \leq \tilde{\sigma}^2$.

$$\tilde{\mathbf{w}}_{\tau,i} = \eta_c\tilde{g}_i(\tilde{\mathbf{x}}_t) - \eta_c\nabla_i\tilde{f}(\tilde{\mathbf{x}}_t) + u_{t+1}\tilde{g}_i(\tilde{\mathbf{x}}_t)$$

$$\leq \eta_c\tilde{q} + u_{t+1}\left(\tilde{g}_i(\tilde{\mathbf{x}}_t) - \nabla_i\tilde{f}(\tilde{\mathbf{x}}_t) + \nabla_i\tilde{f}(\tilde{\mathbf{x}}_t)\right)$$

$$\leq \tilde{q}(\eta_c + u_{t+1}) + u_{t+1}\nabla_i\tilde{f}(\tilde{\mathbf{x}}_t)$$

$$|\tilde{\mathbf{w}}_{\tau,i}|^2 \leq \left(\tilde{q}(\eta_c + u_{t+1}) + u_{t+1}\nabla_i\tilde{f}(\tilde{\mathbf{x}}_t)\right)^2$$

$$= \tilde{q}^2(\eta_c^2 + 2\eta_c u_{t+1} + u_{t+1}^2) + 2\tilde{q}\eta_c u_{t+1}\nabla_i\tilde{f}(\tilde{\mathbf{x}}_t) + 2\tilde{q}u_{t+1}^2\nabla_i\tilde{f}(\tilde{\mathbf{x}}_t) + u_{t+1}^2\left|\nabla_i\tilde{f}(\tilde{\mathbf{x}}_t)\right|^2.$$

Taking expectation with respect to $\tilde{q}$ and the uniformly distributed random variable $u_{t+1}$ and recalling that $\mathbb{E}[u_{t+1}] = 0$, we set expectation over linear functions of $u_{t+1}$ to zero.

$$
\begin{aligned}
\mathbb{E}[|\tilde{\mathbf{w}}_{\tau,i}|^2] &\leq \tilde{\sigma}^2 \eta_c^2 + \tilde{\sigma}^2 \mathbb{E}[u_{t+1}^2] + 2\tilde{\sigma}^2 \mathbb{E}[u_{t+1}^2]\nabla_i \tilde{f}(\tilde{\mathbf{x}}_t) + \mathbb{E}[u_{t+1}^2]\left|\nabla_i \tilde{f}(\tilde{\mathbf{x}}_t)\right|^2 \\
&\leq \tilde{O}(L_{max}^2) + \tilde{O}(L_{max}^2) + \tilde{O}(L_{max}^2)\tilde{O}\left(\frac{1}{\sqrt{L_{max}}}\right) + \tilde{O}(L_{max}^2)\tilde{O}\left(\frac{1}{L_{max}}\right) \\
&= \tilde{O}(L_{max}^2) + \tilde{O}(L_{max}^{1.5}) + \tilde{O}(L_{max}) = \tilde{O}(L_{max}).
\end{aligned}
\tag{39}
$$

Here, we use $\mathbb{E}[u_{t+1}^2] = \frac{(L_{max} - L_{min})^2}{12} = \tilde{O}(L_{max}^2)$. From (19) in the proof of Lemma 2 (Appendix B.1), $\left\|\nabla \tilde{f}(\tilde{\mathbf{x}}_t)\right\| \leq 10\tilde{Q}\sum_{\tau=0}^{\frac{t(t-1)}{2}}(1 + \eta_c\gamma_o)^\tau = \tilde{O}\left(\frac{1}{\sqrt{L_{max}}}\right)$ as $t \leq T = \tilde{O}\left(L_{max}^{-1/4}\right)$. Also, note that $\tilde{q}$ and $u_{t+1}$ are independent of each other. As $\lambda_{min}(\mathbf{H}(\mathbf{x}_0)) = -\gamma_o$,

$$
\begin{aligned}
&\frac{1}{2}\sum_{i=1}^{d}\lambda_i \sum_{\tau=0}^{T-1}(1 - \eta_c\lambda_i)^{2(T-\tau-1)}\mathbb{E}[|\tilde{\mathbf{w}}_{\tau,i}|^2] \\
&\leq \frac{1}{2}\sum_{i=1}^{d}\lambda_i \sum_{\tau=0}^{T-1}(1 + \eta_c\gamma_o)^{2\tau}\mathbb{E}[|\tilde{\mathbf{w}}_{\tau,i}|^2] \leq \frac{\tilde{O}(L_{max})}{2}\sum_{i=1}^{d}\lambda_i \sum_{\tau=0}^{T-1}(1 + \eta_c\gamma_o)^{2\tau} \\
&= \frac{\tilde{O}(L_{max})}{2}\left(-\gamma_o \sum_{\tau=0}^{T-1}(1 + \eta_c\gamma_o)^{2\tau} + (d-1)\lambda_{max}(\mathbf{H}(\mathbf{x}_0))\sum_{\tau=0}^{T-1}(1 + \eta_c\gamma_o)^{2\tau}\right),
\end{aligned}
\tag{40a, 40b}
$$

where we use the upper bound of $\mathbb{E}[|\tilde{\mathbf{w}}_{\tau,i}|^2]$ obtained from (39) in (40a). We use the fact that one of the eigenvalues of $\mathbf{H}(\mathbf{x}_0)$ is $-\gamma_o$ and then upper bound the other eigenvalues by the maximum eigenvalue $\lambda_{max}(\mathbf{H}(\mathbf{x}_0))$ in (40b).

Let $\eta_c \leq \eta_{max} \leq \frac{\sqrt{2}-1}{\gamma'}$ where $\gamma \leq \gamma_o \leq \gamma'$. As $\sum_{\tau=0}^{T-1}(1 + \eta_c\gamma_o)^{2\tau}$ is a monotonically increasing sequence, we choose the smallest $T$ that satisfies $\frac{d}{\eta_c^{1/4}\gamma_o} \leq \sum_{\tau=0}^{T-1}(1 + \eta_c\gamma_o)^{2\tau}$. Therefore, $\sum_{\tau=0}^{T-2}(1 + \eta_c\gamma_o)^{2\tau} \leq \frac{d}{\eta_c^{1/4}\gamma_o}$. Now,

$$
\sum_{\tau=0}^{T-1}(1 + \eta_c\gamma_o)^{2\tau} = 1 + (1 + \eta_c\gamma_o)^2 \sum_{\tau=0}^{T-2}(1 + \eta_c\gamma_o)^{2\tau} \leq 1 + \frac{2d}{\eta_c^{1/4}\gamma_o},
$$

which follows from our constraints that $\eta_c < \frac{\sqrt{2}-1}{\gamma'}$ and $\gamma_o \leq \gamma'$ making $(1 + \eta_c\gamma)^2 \leq \left(1 + \frac{\sqrt{2}-1}{\gamma'}\gamma'\right)^2 \leq 2$. Further using $\eta_c\gamma_o \leq \eta_c^{1/4}\gamma_o \leq \frac{\sqrt{2}-1}{\gamma'}\gamma' < d$,

$$
\frac{d}{\eta_c^{1/4}\gamma_o} \leq \sum_{\tau=0}^{T-1}(1 + \eta_c\gamma_o)^{2\tau} \leq 1 + \frac{2d}{\eta_c^{1/4}\gamma_o} \leq \frac{3d}{\eta_c^{1/4}\gamma_o}
\tag{41}
$$

Hence the order of $T$ is given by $T = O\left(\frac{\log d}{L_{max}^{1/4}\gamma_o}\right)$. We hide the dependence on $d$ when we use $T = \tilde{O}\left(L_{max}^{-1/4}\right)$. Using (41) it can be proved that,

$$
\frac{1}{2}\sum_{i=1}^{d}\lambda_i \sum_{\tau=0}^{T-1}(1 - \eta_c\lambda_i)^{2(T-\tau-1)}\mathbb{E}[|\tilde{\mathbf{w}}_{\tau,i}|^2] \leq -\tilde{O}(L_{max}^{3/4}).
$$

## C.2 Bounding $K_2$ and $K_3$

We define the event $C_T$ as, $C_T = \left\{\forall t \leq T, \|\tilde{\boldsymbol{\kappa}}\| \leq \tilde{O}\left(L_{max}^{3/8}\log\frac{1}{L_{max}}\right), \|\boldsymbol{\kappa}\| \leq \tilde{O}(L_{max}^{3/4})\right\}$. From Lemma 2 and Lemma 3 in Appendix B.1 and B.2 respectively, we know that with probability $\mathbb{P}(C_T) \geq 1 - \tilde{O}\left(L_{max}^{7/2}\right)$, the term $\|\tilde{\boldsymbol{\kappa}}\|$ can be bounded by $\tilde{O}\left(L_{max}^{3/8}\log\frac{1}{L_{max}}\right)$ and $\|\boldsymbol{\kappa}\|$ can be bounded by $\tilde{O}(L_{max}^{3/4})$, $\forall t \leq T = O\left(L_{max}^{-1/4}\right)$.

Now, to complete the proof of Theorem 2, we need to show that the term $K_1$ dominates both $K_2$ and $K_3$. Hence, we obtain the bound for the term $K_2$ as,

$$\mathbb{E}[\zeta \mathbf{1}_{C_T}] = \mathbb{E}\left[\nabla f(\mathbf{x}_0)^T \boldsymbol{\kappa} + \tilde{\boldsymbol{\kappa}}^T \mathbf{H}(\mathbf{x}_0)\boldsymbol{\kappa} + \frac{1}{2}\boldsymbol{\kappa}^T \mathbf{H}(\mathbf{x}_0)\boldsymbol{\kappa} + \frac{\rho}{6}\|\tilde{\boldsymbol{\kappa}} + \boldsymbol{\kappa}\|^3\right]\mathbb{P}(C_T)$$

$$\leq \tilde{O}\left(L_{max}^{3/8}\log\frac{1}{L_{max}}\right)\tilde{O}(L_{max}^{3/4})\mathbb{P}(C_T) = \tilde{O}\left(L_{max}^{9/8}\log\frac{1}{L_{max}}\right)\mathbb{P}(C_T).$$

Finally, we bound the term $K_3$ as follows.

$$\mathbb{E}[(f(\mathbf{x}_T) - f(\mathbf{x}_0))\mathbf{1}_{\bar{C}_T}] - \mathbb{E}[\tilde{\zeta}\mathbf{1}_{\bar{C}_T}] \leq \tilde{O}(1)\mathbb{P}(\bar{C}_T) \leq \tilde{O}\left(L_{max}^{7/2}\right),$$

where the inequality arises from the boundedness of the function. Comparing the bounds of the terms $K_1$, $K_2$, and $K_3$, we find that $K_1$ dominates, which completes the proof. $\qquad\square$

# D   Proof of Theorem 3

**Theorem 6.** *(Theorem 3 restated) Consider $f$ satisfying the assumptions **A1-A6**. Let the initial iterate $\boldsymbol{x}_0$ be $\delta$ close to a local minimum $\boldsymbol{x}^*$ such that $\|\boldsymbol{x}_0 - \boldsymbol{x}^*\| \leq \tilde{O}(\sqrt{L_{max}}) < \delta$. With probability at least $1 - \xi$, $\forall t \leq T$ where $T = \tilde{O}\left(\frac{1}{L_{max}^2}\log\frac{1}{\xi}\right)$,*

$$\|\boldsymbol{x}_t - \boldsymbol{x}^*\| \leq \tilde{O}\left(\sqrt{L_{max}\log\frac{1}{L_{max}\xi}}\right) < \delta$$

*Proof.* This theorem handles the case when the iterate is close to the local minimum (case **B3**). We aim to show that the iterate does not leave the neighbourhood of the minimum for $t \leq \tilde{O}\left(\frac{1}{L_{max}^2}\log\frac{1}{\xi}\right)$. By assumption **A6**, if $\mathbf{x}_t$ is $\delta$ close to the local minimum $\mathbf{x}^*$, the function is locally $\alpha$- strongly convex. We define event $D_t = \{\forall \tau \leq t, \|\mathbf{x}_\tau - \mathbf{x}^*\| \leq \mu\sqrt{L_{max}\log\frac{1}{L_{max}\xi}} < \delta\}$. Let $L_{max} < \frac{r}{\log\xi^{-1}}$ where $r < \log\xi^{-1}$. It can be seen that $D_{t-1} \subset D_t$. Conditioned on event $D_t$, and using $\alpha-$strong convexity of $f$, $(\nabla f(\mathbf{x}_t) - \nabla f(\mathbf{x}^*))^T(\mathbf{x}_t - \mathbf{x}^*)\mathbf{1}_{D_t} \geq \alpha\|\mathbf{x}_t - \mathbf{x}^*\|^2\mathbf{1}_{D_t}$. As $\nabla f(\mathbf{x}^*) = 0$, it becomes, $\nabla f(\mathbf{x}_t)^T(\mathbf{x}_t - \mathbf{x}^*)\mathbf{1}_{D_t} \geq \alpha\|\mathbf{x}_t - \mathbf{x}^*\|^2\mathbf{1}_{D_t}$. We define a filtration $S_t = s\{\mathbf{w}_0, \ldots, \mathbf{w}_{t-1}\}$ in order to construct a supermartingale and use the Azuma-Hoeffding inequality where $s\{.\}$ denotes a sigma-algebra field. Now, assuming $L_{max} < \frac{\alpha}{\beta^2}$,

$$\mathbb{E}[\|\mathbf{x}_t - \mathbf{x}^*\|^2 \mathbf{1}_{D_{t-1}}|S_{t-1}] = \mathbb{E}[\|\mathbf{x}_{t-1} - \eta_c\nabla f(\mathbf{x}_{t-1}) - \mathbf{w}_{t-1} - \mathbf{x}^*\|^2|S_{t-1}]\mathbf{1}_{D_{t-1}}$$

$$= \mathbb{E}[\|(\mathbf{x}_{t-1} - \mathbf{x}^*) - \eta_c\nabla f(\mathbf{x}_{t-1}) - \mathbf{w}_{t-1}\|^2|S_{t-1}]\mathbf{1}_{D_{t-1}}$$

$$= [\|\mathbf{x}_{t-1} - \mathbf{x}^*\|^2 - 2\eta_c(\mathbf{x}_{t-1} - \mathbf{x}^*)^T\nabla f(\mathbf{x}_{t-1}) + \eta_c^2\|\nabla f(\mathbf{x}_{t-1})\|^2 + \mathbb{E}[\|\mathbf{w}_{t-1}\|^2]]\mathbf{1}_{D_{t-1}} \quad (42a)$$

$$\leq [\|\mathbf{x}_{t-1} - \mathbf{x}^*\|^2 - 2\eta_c\alpha\|\mathbf{x}_{t-1} - \mathbf{x}^*\|^2 + \eta_c^2\beta^2\|\mathbf{x}_{t-1} - \mathbf{x}^*\|^2 + \mathbb{E}[\|\mathbf{w}_{t-1}\|^2]]\mathbf{1}_{D_{t-1}} \quad (42b)$$

We use $\mathbb{E}[\mathbf{w}_t] = 0$ in (42a). We use the $\beta$-smoothness and $\alpha-$convexity assumptions of $f$ in (42b). Now, using $\mathbf{w}_{t-1} = \eta_c g(\mathbf{x}_{t-1}) - \eta_c\nabla f(\mathbf{x}_{t-1}) + u_t g(\mathbf{x}_{t-1})$, we compute $\mathbb{E}[\|\mathbf{w}_{t-1}\|^2]$ as,

$$\mathbb{E}[\|\mathbf{w}_{t-1}\|^2]$$

$$= \mathbb{E}\left[\eta_c^2\|g(\mathbf{x}_{t-1}) - \nabla f(\mathbf{x}_{t-1})\|^2 + 2\eta_c u_t\left(g(\mathbf{x}_{t-1}) - \nabla f(\mathbf{x}_{t-1})\right)^T g(\mathbf{x}_{t-1}) + u_t^2\|g(\mathbf{x}_{t-1})\|^2\right]$$

$$\leq \eta_c^2\sigma^2 + \mathbb{E}[u_t^2]\mathbb{E}[\|g(\mathbf{x}_{t-1})\|^2] \leq \eta_c^2\sigma^2 + \mathbb{E}[u_t^2](\sigma^2 + \|\nabla f(\mathbf{x}_{t-1})\|^2)$$

$$\leq \eta_c^2\sigma^2 + \mathbb{E}[u_t^2]\sigma^2 + \mathbb{E}[u_t^2]\beta^2\|\mathbf{x}_{t-1} - \mathbf{x}^*\|^2$$

$$\leq \sigma^2\left(\eta_c^2 + \frac{2L_{max}^2}{3} - \frac{2L_{max}\eta_c}{3}\right) + \beta^2\|\mathbf{x}_{t-1} - \mathbf{x}^*\|^2\left(\frac{2L_{max}^2}{3} - \frac{2L_{max}\eta_c}{3}\right).$$

$$(43)$$

630 As $\eta_c = \frac{L_{min}+L_{max}}{2}$, $L_{min} = 2\eta_c - L_{max}$. Hence, we write $\mathbb{E}[u_t^2] = \frac{(L_{max}-L_{min})^2}{12} =$
631 $\frac{4(L_{max}-\eta_c)^2}{12} = \frac{L_{max}^2+\eta_c^2-2L_{max}\eta_c}{3} < \frac{2L_{max}^2}{3} - \frac{2L_{max}\eta_c}{3}$ in (43). Using (43) in (42b),

$$
\begin{aligned}
\mathbb{E}[\|\mathbf{x}_t - \mathbf{x}^*\|^2 \mathbf{1}_{D_{t-1}}|S_{t-1}] &\leq \left[\|\mathbf{x}_{t-1} - \mathbf{x}^*\|^2 \left(1 - 2\eta_c\alpha + \eta_c^2\beta^2 + \frac{2L_{max}^2\beta^2}{3} - \frac{2L_{max}\eta_c\beta^2}{3}\right)\right.\\
&\quad \left. +\sigma^2\left(\eta_c^2 + \frac{2L_{max}^2}{3} - \frac{2L_{max}\eta_c}{3}\right)\right]\mathbf{1}_{D_{t-1}}\\
&\leq \left[\|\mathbf{x}_{t-1} - \mathbf{x}^*\|^2\left(1 + \eta_c\alpha + \frac{2L_{max}\alpha}{3}\right) + \sigma^2\left(L_{max}^2 + \frac{2L_{max}^2}{3}\right)\right]\mathbf{1}_{D_{t-1}}\\
&\leq \left[\|\mathbf{x}_{t-1} - \mathbf{x}^*\|^2\left(1 + L_{max}\alpha + \frac{2L_{max}\alpha}{3}\right) + \sigma^2\left(L_{max}^2 + \frac{2L_{max}^2}{3}\right)\right]\mathbf{1}_{D_{t-1}}\\
&= \left[\|\mathbf{x}_{t-1} - \mathbf{x}^*\|^2\left(1 + \frac{5L_{max}\alpha}{3}\right) + \frac{5L_{max}^2\sigma^2}{3}\right]\mathbf{1}_{D_{t-1}}.
\end{aligned}
$$

632 We use $L_{max} < \frac{\alpha}{\beta^2}$. Let $J_t = \left(1 + \frac{5\alpha L_{max}}{3}\right)^{-t}\left(\|\mathbf{x}_t - \mathbf{x}^*\|^2 + \frac{L_{max}\sigma^2}{\alpha}\right)$. We prove $J_t\mathbf{1}_{D_{t-1}}$ is a
633 supermartingale process as follows.
634

$$
\begin{aligned}
\mathbb{E}\left[\left(1 + \frac{5\alpha L_{max}}{3}\right)^{-t}\left(\|\mathbf{x}_t - \mathbf{x}^*\|^2 + \frac{L_{max}\sigma^2}{\alpha}\right)\Big|S_{t-1}\right]\mathbf{1}_{D_{t-1}} &\leq \\
\left(1 + \frac{5\alpha L_{max}}{3}\right)^{-t}\left[\|\mathbf{x}_{t-1} - \mathbf{x}^*\|^2\left(1 + \frac{5L_{max}\alpha}{3}\right) + \frac{5L_{max}^2\sigma^2}{3} + \frac{L_{max}\sigma^2}{\alpha}\right]\mathbf{1}_{D_{t-1}} &\\
= \left(1 + \frac{5\alpha L_{max}}{3}\right)^{-(t-1)}\left[\|\mathbf{x}_{t-1} - \mathbf{x}^*\|^2 + \frac{L_{max}\sigma^2}{\alpha}\right]\mathbf{1}_{D_{t-1}} = J_{t-1}\mathbf{1}_{D_{t-1}} &\leq J_{t-1}\mathbf{1}_{D_{t-2}}.
\end{aligned}
$$

635 Hence $J_t\mathbf{1}_{D_{t-1}}$ is a supermartingale. In order to use the Azuma-Hoeffding inequality, we bound
636 $|J_t\mathbf{1}_{D_{t-1}} - \mathbb{E}[J_t\mathbf{1}_{D_{t-1}}|S_{t-1}]|$ as,
637

$$
\begin{aligned}
&|J_t\mathbf{1}_{D_{t-1}} - \mathbb{E}[J_t\mathbf{1}_{D_{t-1}}|S_{t-1}]| = \left(1 + \frac{5\alpha L_{max}}{3}\right)^{-t}\left[\|\mathbf{x}_t - \mathbf{x}^*\|^2 - \mathbb{E}[\|\mathbf{x}_t - \mathbf{x}^*\|^2|S_{t-1}]\right]\mathbf{1}_{D_{t-1}}\\
&\leq \left(1 + \frac{5\alpha L_{max}}{3}\right)^{-t}\left[2\|\mathbf{x}_{t-1} - \eta_c\nabla f(\mathbf{x}_{t-1}) - \mathbf{x}^*\|\,\|\mathbf{w}_{t-1}\| + \|\mathbf{w}_{t-1}\|^2 +\right.\\
&\left. \sigma^2\left(\eta_c^2 + \frac{2L_{max}^2}{3} - \frac{2L_{max}\eta_c}{3}\right) + \beta^2\|\mathbf{x}_{t-1} - \mathbf{x}^*\|^2\left(\frac{2L_{max}^2}{3} - \frac{2L_{max}\eta_c}{3}\right)\right]\mathbf{1}_{D_{t-1}},
\end{aligned}
$$
(44)

638 where we use (43) in (44) for the term $\mathbb{E}[\|\mathbf{w}_{t-1}\|^2]$. Now, we compute $\|\mathbf{w}_{t-1}\|$ using assumption **A3**
639 as follows.

$$
\begin{aligned}
\|\mathbf{w}_{t-1}\| &= \|\eta_c g(\mathbf{x}_{t-1}) - \eta_c\nabla f(\mathbf{x}_{t-1}) + u_t g(\mathbf{x}_{t-1})\|\\
&\leq \eta_c Q + |u_t|(Q + \|\nabla f(\mathbf{x}_{t-1})\|) \leq Q(\eta_c + |u_t|) + |u_t|\beta\|\mathbf{x}_{t-1} - \mathbf{x}^*\|.
\end{aligned}
$$
(45)

640    Using (45) in (44) and the bound of the event $D_{t-1}$,

$$|J_t \mathbf{1}_{D_{t-1}} - \mathbb{E}[J_t \mathbf{1}_{D_{t-1}} | S_{t-1}]|$$

$$\leq \left(1 + \frac{5\alpha L_{max}}{3}\right)^{-t} \left[ 2 \|\mathbf{x}_{t-1} - \mathbf{x}^*\| \left(Q(\eta_c + |u_t|) + |u_t|\beta \|\mathbf{x}_{t-1} - \mathbf{x}^*\|\right) \right.$$

$$+ \left(Q(\eta_c + |u_t|) + |u_t|\beta \|\mathbf{x}_{t-1} - \mathbf{x}^*\|\right)^2 + \sigma^2 \left(\eta_c^2 + \frac{2L_{max}^2}{3} - \frac{2L_{max}\eta_c}{3}\right)$$

$$+ \beta^2 \|\mathbf{x}_{t-1} - \mathbf{x}^*\|^2 \left(\frac{2L_{max}^2}{3} - \frac{2L_{max}\eta_c}{3}\right) \bigg] \mathbf{1}_{D_{t-1}}$$

$$= \left(1 + \frac{5\alpha L_{max}}{3}\right)^{-t} \left[ \tilde{O}\left(\mu L_{max}^{1.5} \log^{0.5} \frac{1}{L_{max}\xi}\right) + \tilde{O}\left(\mu^2 L_{max}^2 \log \frac{1}{L_{max}\xi}\right) + 2\tilde{O}(L_{max}^2) \right.$$

$$+ \tilde{O}\left(\mu L_{max}^{2.5} \log^{0.5} \frac{1}{L_{max}\xi}\right) + 2\tilde{O}\left(\mu^2 L_{max}^3 \log \frac{1}{L_{max}\xi}\right) \bigg]$$

$$\leq \left(1 + \frac{5\alpha L_{max}}{3}\right)^{-t} \tilde{O}\left(\mu L_{max}^{1.5} \log^{0.5} \frac{1}{L_{max}\xi}\right) = d_t$$

641    We denote the bound of $|J_t \mathbf{1}_{D_{t-1}} - \mathbb{E}[J_t \mathbf{1}_{D_{t-1}} | S_{t-1}]|$ as $d_t$.

642    Let $b_t = \sqrt{\sum_{\tau=1}^t d_\tau^2} = \sqrt{\sum_{\tau=1}^t \left(1 + \frac{5\alpha L_{max}}{3}\right)^{-2\tau} \tilde{O}\left(\mu L_{max}^{1.5} \log^{0.5} \frac{1}{L_{max}\xi}\right)}$. Now,

$$\sqrt{\sum_{\tau=1}^t \left(1 + \frac{5\alpha L_{max}}{3}\right)^{-2\tau} \tilde{O}\left(\mu L_{max}^{1.5} \log^{0.5} \frac{1}{L_{max}\xi}\right)}$$

$$\leq \sqrt{\frac{1}{1 - \left(1 + \frac{5\alpha L_{max}}{3}\right)^{-2}}} \tilde{O}\left(\mu L_{max}^{1.5} \log^{0.5} \frac{1}{L_{max}\xi}\right)$$

$$= \sqrt{\frac{\tilde{O}(1)}{\tilde{O}(L_{max})}} \tilde{O}\left(\mu L_{max}^{1.5} \log^{0.5} \frac{1}{L_{max}\xi}\right) = \tilde{O}\left(\mu L_{max} \log^{0.5} \frac{1}{L_{max}\xi}\right).$$

643    Hence $b_t$ is of the order $\tilde{O}\left(\mu L_{max} \log^{0.5} \frac{1}{L_{max}\xi}\right)$. By the Azuma Hoeffding inequality,

$$\mathbb{P}\left(J_t \mathbf{1}_{D_{t-1}} - J_0 \geq b_t \log^{0.5} \frac{1}{L_{max}\xi}\right) \leq \exp\left(-\tilde{\Omega}\left(\log \frac{1}{L_{max}\xi}\right)\right) \leq \tilde{O}(L_{max}^3 \xi),$$

644    which leads to,

$$\mathbb{P}\left(J_t \mathbf{1}_{D_{t-1}} - J_0 \geq \tilde{O}\left(\mu L_{max} \log \frac{1}{L_{max}\xi}\right)\right) \leq \tilde{O}(L_{max}^3 \xi).$$

645    Hence we can write,

$$\mathbb{P}\left(D_{t-1} \cap \left\{\|\mathbf{x}_t - \mathbf{x}^*\|^2 \geq \tilde{O}\left(\mu L_{max} \log \frac{1}{L_{max}\xi}\right)\right\}\right) \leq \tilde{O}(L_{max}^3 \xi)$$

646    For some constant $\tilde{b}$ independent of $L_{max}$ and $\xi$ we can write,

$$\mathbb{P}\left(D_{t-1} \cap \left\{\|\mathbf{x}_t - \mathbf{x}^*\|^2 \geq \tilde{b}\mu L_{max} \log \frac{1}{L_{max}\xi}\right\}\right) \leq \tilde{O}(L_{max}^3 \xi)$$

647    By choosing $\mu < \tilde{b}$,

$$\mathbb{P}\left(D_{t-1} \cap \left\{\|\mathbf{x}_t - \mathbf{x}^*\| \geq \mu\sqrt{L_{max} \log \frac{1}{L_{max}\xi}}\right\}\right) \leq \tilde{O}(L_{max}^3 \xi)$$

648

$$\mathbb{P}(\bar{D}_t) = \mathbb{P}\left(D_{t-1} \cap \left\{\|\mathbf{x}_t - \mathbf{x}^*\| \geq \mu\sqrt{L_{max} \log \frac{1}{L_{max}\xi}}\right\}\right) + \mathbb{P}(\bar{D}_{t-1})$$

$$\leq \tilde{O}(L_{max}^3 \xi) + \mathbb{P}(\bar{D}_{t-1})$$

649    Iteratively unrolling the above equation, we obtain $\mathbb{P}(\bar{D}_t) \leq t\tilde{O}(L_{max}^3 \xi)$. Choosing $t =$
650    $\tilde{O}\left(\frac{1}{L_{max}^2} \log \frac{1}{\xi}\right)$, $\mathbb{P}(\bar{D}_t) \leq \tilde{O}\left(L_{max} \xi \log \frac{1}{\xi}\right)$. As $L_{max} < \tilde{O}\left(\frac{1}{\log \frac{1}{\xi}}\right)$, $\mathbb{P}(\bar{D}_t) \leq \tilde{O}(\xi)$.    $\square$

## 651    E    Proof using induction

652 In the proof of Lemma 2 in Appendix B.1, we state that (19) can be proved by induction for $t \geq 2$.
653 We restate the equation here and provide the corresponding proof by induction.

$$\text{Induction hypothesis:} \quad \left\| \nabla \tilde{f}(\tilde{\mathbf{x}}_t) \right\| \leq 10\tilde{Q} \sum_{\tau=0}^{\frac{t(t-1)}{2}} (1 + \eta_c \gamma_o)^\tau. \tag{46}$$

654 Recollect from that (15) that $\nabla \tilde{f}(\tilde{\mathbf{x}}_t) = (I - \eta_c \mathbf{H}(\mathbf{x}_0)) \nabla \tilde{f}(\tilde{\mathbf{x}}_{t-1}) - \mathbf{H}(\mathbf{x}_0) \tilde{\mathbf{w}}_{t-1}$. Taking matrix
655 induced norm on both sides,

$$\left\| \nabla \tilde{f}(\tilde{\mathbf{x}}_{t+1}) \right\| \leq (1 + \eta_c \gamma_o) \left\| \nabla \tilde{f}(\tilde{\mathbf{x}}_t) \right\| + \beta \|\tilde{\mathbf{w}}_t\|$$

$$= ((1 + \eta_c \gamma_o) + \beta |u_{t+1}|) \left\| \nabla \tilde{f}(\tilde{x}_t) \right\| + \beta \tilde{Q}(\eta_c + |u_{t+1}|), \tag{47}$$

656 since, $\left\| \tilde{g}(\tilde{\mathbf{x}}_t) - \nabla \tilde{f}(\tilde{\mathbf{x}}_t) \right\| \leq \tilde{Q}$. Note that $\left\| \nabla \tilde{f}(\tilde{\mathbf{x}}_t) \right\| \leq \epsilon$, $|u_t| \leq L_{max}$ and $\beta L_{max} < 1$ hold for all
657 $t$. Therefore, at $t = 1$,

$$\left\| \nabla \tilde{f}(\tilde{\mathbf{x}}_1) \right\| \leq ((1 + \eta_c \gamma_o) + \beta |u_1|) \epsilon + \beta \tilde{Q}(\eta_c + |u_1|) \leq (1 + \eta_c \gamma_o)\epsilon + \epsilon + 2\tilde{Q}.$$

658 Now, we prove the hypothesis in (46) for $t = 2$. From (47), for an arbitrarily small $\epsilon$,

$$\left\| \nabla \tilde{f}(\tilde{\mathbf{x}}_2) \right\| \leq ((1 + \eta_c \gamma_o) + \beta |u_2|) \left\| \nabla \tilde{f}(\tilde{\mathbf{x}}_1) \right\| + \beta \tilde{Q}(\eta_c + |u_2|)$$

$$\leq (1 + \eta_c \gamma_o)^2 \epsilon + 2(1 + \eta_c \gamma_o)\epsilon + \epsilon + 2\tilde{Q}(1 + \eta_c \gamma_o) + 4\tilde{Q}$$

$$\leq 2\epsilon \sum_{\tau=0}^{2} (1 + \eta_c \gamma_o)^\tau + 4\tilde{Q} \sum_{\tau=0}^{1} (1 + \eta_c \gamma_o)^\tau \leq 10\tilde{Q} \sum_{\tau=0}^{\frac{2(2-1)}{2}} (1 + \eta_c \gamma_o)^\tau.$$

659 We have shown that the induction hypothesis holds for $t = 2$. Now, assuming that it holds for any $t$,
660 we need to prove that it holds for $t + 1$. We know from (47), when the hypothesis is assumed to hold
661 for $t$,

$$\left\| \nabla \tilde{f}(\tilde{\mathbf{x}}_{t+1}) \right\| \leq ((1 + \eta_c \gamma_o) + \beta |u_{t+1}|) 10\tilde{Q} \sum_{\tau=0}^{\frac{t(t-1)}{2}} (1 + \eta_c \gamma_o)^\tau + \beta \tilde{Q}(\eta_c + |u_{t+1}|)$$

$$\leq (1 + \eta_c \gamma_o) 10\tilde{Q} \sum_{\tau=0}^{\frac{t(t-1)}{2}} (1 + \eta_c \gamma_o)^\tau + 10\tilde{Q} \sum_{\tau=0}^{\frac{t(t-1)}{2}} (1 + \eta_c \gamma_o)^\tau + \beta \tilde{Q}(\eta_c + |u_{t+1}|)$$

$$\leq 20\tilde{Q} \sum_{\tau=0}^{\frac{t(t-1)}{2}+1} (1 + \eta_c \gamma_o)^\tau$$

662 If we prove $20\tilde{Q} \sum_{\tau=0}^{\frac{t(t-1)}{2}+1} (1 + \eta_c \gamma_o)^\tau \leq 10\tilde{Q} \sum_{\tau=0}^{\frac{t(t+1)}{2}} (1 + \eta_c \gamma_o)^\tau$, the induction proof is complete.
663 Now, we need to prove

$$20\tilde{Q} \sum_{\tau=0}^{\frac{t^2-t}{2}+1} (1 + \eta_c \gamma_o)^\tau \leq 10\tilde{Q} \sum_{\tau=0}^{\frac{t^2+t}{2}} (1 + \eta_c \gamma_o)^\tau$$

$$\leq 10\tilde{Q} \sum_{\tau=0}^{\frac{t^2-t}{2}+1} (1 + \eta_c \gamma_o)^\tau + 10\tilde{Q} \sum_{\tau=\frac{t^2-t}{2}+2}^{\frac{t^2+t}{2}} (1 + \eta_c \gamma_o)^\tau.$$

664 Therefore we need to show that,

$$\underbrace{\sum_{\tau=0}^{\frac{t^2-t}{2}+1} (1 + \eta_c \gamma_o)^\tau}_{S_1} \leq \underbrace{\sum_{\tau=\frac{t^2-t}{2}+2}^{\frac{t^2+t}{2}} (1 + \eta_c \gamma_o)^\tau}_{S_2}. \tag{48}$$

Now, summing up the geometric series $S_1$, $\sum_{\tau=0}^{\frac{t^2-t}{2}+1}(1+\eta_c\gamma_o)^\tau = \frac{(1+\eta_c\gamma_o)^{\frac{t^2-t}{2}+2}-1}{\eta_c\gamma_o}$. Using change of variable in $S_2$ of (48) as $m = \tau - \left(\frac{t^2-t}{2}+2\right)$,

$$\sum_{m=0}^{t-2}(1+\eta_c\gamma_o)^{\frac{t^2-t}{2}+m+2} = (1+\eta_c\gamma_o)^{\frac{t^2-t}{2}+2}\frac{(1+\eta_c\gamma_o)^{t-1}-1}{\eta_c\gamma_o}.$$

Therefore, we now need to prove,

$$(1+\eta_c\gamma_o)^{\frac{t^2-t}{2}+2}-1 \le (1+\eta_c\gamma_o)^{\frac{t^2-t}{2}+2}\left((1+\eta_c\gamma_o)^{t-1}-1\right)$$
$$\Rightarrow 2(1+\eta_c\gamma_o)^{\frac{t^2-t}{2}+2} \le (1+\eta_c\gamma_o)^{\frac{t^2-t}{2}+t+1}+1 \tag{49}$$

We further prove (49) by induction as follows. For $t=2$, $2(1+\eta_c\gamma_o)^3 \le (1+\eta_c\gamma_o)^4+1$. Let us assume the following expression holds for time step $t$.

$$2(1+\eta_c\gamma_o)^{\frac{t^2-t}{2}+2} \le (1+\eta_c\gamma_o)^{\frac{t^2-t}{2}+t+1} \tag{50}$$

Now, we prove for the time step $t+1$,

$$2(1+\eta_c\gamma_o)^{\frac{t(t+1)}{2}+2} = 2(1+\eta_c\gamma_o)^{\frac{t(t-1)}{2}+t+2} \le (1+\eta_c\gamma_o)^{\frac{t^2-t}{2}+t+1+t}$$
$$= (1+\eta_c\gamma_o)^{\frac{t(t+1)}{2}+t+1} \le (1+\eta_c\gamma_o)^{\frac{t(t+1)}{2}+t+2}, \tag{51}$$

where we use $\frac{t(t-1)}{2}+t = \frac{t(t+1)}{2}$ and apply our assumption (50) in (51). We have proved $2(1+\eta_c\gamma_o)^{\frac{t^2-t}{2}+2} \le (1+\eta_c\gamma_o)^{\frac{t^2-t}{2}+t+1} \le (1+\eta_c\gamma_o)^{\frac{t^2-t}{2}+t+1}+1$. This concludes our proof of (46).

# F   Choice of parameters for other LR schedulers

1. Cosine annealing [21]: There are 3 parameters namely, initial restart interval, a multiplicative factor and minimum learning rate. The authors propose an initial restart interval of 1, a factor of 2 for subsequent restarts, with a minimum learning rate of $1e-4$, which we use in our comparisons.

2. Knee [14]: The total number of epochs is divided into those that correspond to the "explore" epochs and "exploit" epochs. During the explore epochs, the learning rate is kept at a constant high value, while from the beginning of the exploit epochs, it is linearly decayed. We use the suggested setting of 100 initial explore epochs with a learning rate of 0.1 followed by a linear decay for the rest of the epochs.

3. One cycle [25]: We perform the learning rate range test for our networks as suggested by the authors. For the range test, the learning rate is gradually increased during which the training loss explodes. The learning rate at which it explodes is noted and the maximum learning rate (the learning rate at the middle of the triangular cycle) is fixed to be before that. We linearly increase the learning rate for the initial $45\%$ of the total epochs up to the maximum learning rate determined by the range test, followed by a linear decay for the next $45\%$ of the total epochs. We then decay it further up to a divisive factor of 10 for the rest of the epochs, which is the suggested setting. Note that the one cycle LR scheduler relies heavily on regularization parameters like weight decay and momentum.

4. Constant: To compare with a constant learning rate, we choose $0.05$ for the VGG-16 architecture and $0.1$ for the remaining architectures as done in our other baselines[24, 21].

5. Multi step: For the multi-step decay scheduler, our choice of the decay rate and time is based on the standard repositories for the architectures. [4]. Specifically, we decay the learning rate by a factor of 10 at the the epochs 100 and 150 for ResNet-110 and ResNet-50. In the case of DenseNet-40-12, we decay by a factor of 10 at the epochs 150 and 225. For VGG-16, we decay by a factor of 10 every 30 epochs. In the case of WRN, we fix a learning rate of $0.2$ for the initial 60 epochs, decay it by $0.2^2$ for the next 60 epochs, and by $0.2^3$ for the rest of the epochs.

---

[4]ResNet:https://github.com/akamaster/pytorch_resnet_cifar10,
DenseNet:https://github.com/andreasveit/densenet-pytorch,
VGG:https://github.com/chengyangfu/pytorch-vgg-cifar10,
WRN:https://github.com/meliketoy/wide-resnet.pytorch

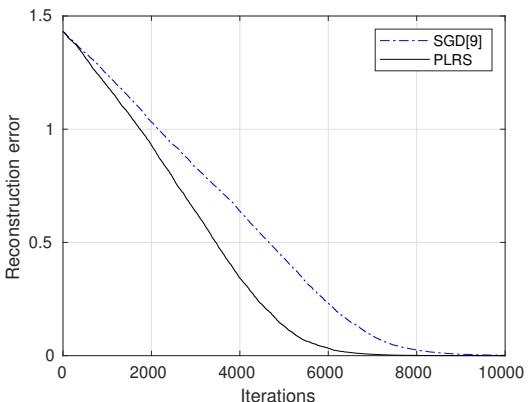

Figure 4: Reconstruction error for online tensor decomposition

## G  Online tensor decomposition

We follow the experimental setup in [9], where their proposed projected noisy gradient descent is applied to orthogonal tensor decomposition. A brief description of the online tensor decomposition problem is given below.

Consider a tensor $T$ which has an orthogonal decomposition,

$$T = \sum_{i=1}^{d} a_i^{\otimes 4}, \tag{52}$$

where $a_i$'s are orthonormal vectors. The goal of performing the tensor decomposition is to find the orthonormal components, given the tensor. The objective function is defined to reduce the correlation between the components:

$$\min_{\forall i, \|u_i\|=1} \sum_{i \neq j} T(u_i, u_i, u_j, u_j) \tag{53}$$

We plot the normalized reconstruction error, $\left\| T - \sum_{i=1}^{d} u_i^{\otimes 4} \right\|_F^2 / \|T\|_F^2$ in Figure 4, where $\|.\|_F$ denotes the Frobenius norm. We tune the learning rate parameters $L_{min}$ and $L_{max}$ to 0.007 and 0.01 respectively to obtain the convergence plot with PLRS. We compare against the plot in Figure 1.a of [9]. We note that the proposed Uniform LR produces faster and smoother convergence when compared to the unit sphere noise proposed in the Noisy SGD algorithm. As mentioned in [9], the plot may vary depending on the instance of initialization; however, it converges consistently across all runs.

Additionally, we implemented stochastic gradient descent with additive noise in the neural network setting. However, its performance was suboptimal even with extensive tuning of hyperparameters.

