# OpenReview forum: "Randomness Helps Rigor: A Probabilistic Learning Rate Scheduler Bridging Theory and Deep Learning Practice"
_NeurIPS.cc/2025/Conference — Submitted to NeurIPS 2025_

### Official Review · Reviewer_69TR · 2025-06-04

**Clarity:** 4
**Significance:** 3
**Originality:** 4
**Rating:** 5
**Confidence:** 4

**Summary:**

This paper proposes PLRS, which uses randomness to do gradient descent with good empirical performance. The proof is good but there is a gap between the proof and the empirical performance, which is fine. I think it should be the case that convergence happens because the function is strongly convex, and when it's hit with a uniform noise this accomplished the same thing as a decreasing learning rate no?

**Questions:**

From a quick glance the proof looks good. Is it the case that submartingale or martingale analysis was used in the proof?

**Ethical Concerns:**

["NO or VERY MINOR ethics concerns only"]

**Final Justification:**

I keep my score. Please take a look at collective feedback on how to improve the draft.

**Quality:**

3

**Strengths And Weaknesses:**

Strengths:
The coverage of related work in the introduction is good.
excellent novelty
good proofs
good writing

Once section 2 starts, the tempo of the rest of the paper is good.

The empirical validation is sufficient for a theory work.

Weaknesses:
The introduction does not sufficiently give a high level overview of the problem that is being solved. Although the paper is easy to understand for experts within this subarea, it is not well explained what is the problem being solved to people even slightly out of the subarea.

Separating the introduction into subsections hurts the flow of the paper.

In particular, I would like to understand the difference between additive vs. multiplicative noise and how that assumption/decision affects the analysis. There needs to be a discussion regarding robustness as I believe the multiple orders of magnitude covered by the noise aids in robustness against a diverse variety of datasets and deep learning architectures.

This line is the only motivation line in the work, it needs to fleshed out more and be more front and center:

"With
72 the increase of emphasis on trustworthy artificial intelligence, we believe that it is important to no
73 longer treat optimization algorithms as black-box models, and instead provide provable convergence
74 guarantees while deviating from the proven classical implementation of the descent algorithms"

Empirical validation is not too much.

" for SGD using LR schedulers.
324 In our opinion, this is a significant step in the right direction to bridge the gap between theory and
325"
it is not opinion if its shown theoretically and empirically.

---

> ### Author Rebuttal · Authors · 2025-07-30
>
> We thank the reviewer for their comments and constructive suggestions. We appreciate your thoughtful feedback and are glad that you found merit in our work.
>
> **Difference between additive and multiplicative noise and how it affects the analysis**
> We have now elaborated on the characteristics of the multiplicative noise.
>
> In the small-gradient regime, such as near a local minimum or a saddle point, the intuition is that occasional sampling of a high learning rate (close to $L_{\text{max}}$) causes the stochastic gradient to be scaled significantly, enabling the iterate to move away from the saddle point. Conversely, when the learning rate sampled from the uniform distribution $U[L_{\text{min}}, L_{\text{max}}]$ is low, the iterate may undergo only minor changes. However, due to the randomness in the learning rate, repeated application of this multiplicative noise results, with high probability, in the iterate escaping the saddle point. By bringing in multiplicative noise using a probabilistic LR scheduler, in this work we analytically show that under certain assumptions, the iterate escapes a saddle point region with high probability.
>
> We believe that incorporating this discussion in the paper will offer clearer context on the role and implications of multiplicative noise. We appreciate the reviewer’s suggestion to expand on this point.
>
> **Empirical validation**
> In addition to the ones we have provided in the paper, we have now run additional experiments on the ImageNet-1K dataset with Resnet-50 architecture and have obtained a maximum top-1 accuracy of 68.25. Also, we have fine-tuned BERT-large with the SQuAD V1.1 dataset and obtained an F1 score of **87.69**.  For more details on the base settings for the experiments, please refer to our response to Reviewer KReu.
>
> **Robustness of the proposed approach:**
> We thank the reviewer for this insight. We agree that the added multiplicative noise will aid in greater robustness and will offer better generalization for a wide variety of datasets and architectures. In line with your observation, we observe consistently strong performance not only in our original experiments but also in the additional, larger-scale tasks (described above) we conducted for this rebuttal, across different datasets and model types.  We will be sure to add this as an advantage of our method to the discussion.
>
> **Type of martingale analysis:**
>
> We use supermartingale analysis to prove convergence using the Azuma Hoeffding inequality for martingales and supermartingales (stated in lines 537-539 of our paper). Specifically, we construct a martingale process based on the norm of the difference between the function's gradient and its Taylor approximation at each iterate, and then prove that it is a supermartingale process. We then use Azuma Hoeffding inequality to prove that they remain coupled with high probability. Similarly, we prove that the SGD iterates of the function remain coupled with those of its Taylor approximation. Finally, we establish expected descent of the function value using Taylor approximation and the high-probability bounds we derive.
>
> **Fleshing out motivation**
>
> We appreciate the reviewer’s suggestion to flesh this out more. We agree that it is central to the motivation of our work. We have expanded upon this as follows and would love the opportunity to incorporate it into our Introduction.
>
> *As machine learning systems become increasingly embedded in high-stakes domains, the demand for trustworthy and reliable AI* *continues to grow. This trustworthiness extends beyond model predictions and fairness metrics to the every **algorithms used to train these models**. Despite their foundational role, optimization algorithms are still often treated as black-box tools: selected based on* *empirical performance, tuned heuristically, and frequently lacking theoretical transparency in modern settings.*
> *We argue that this is no longer sufficient. The optimization process plays a crucial role in the final model behavior, including its* *generalization, robustness, and even fairness properties. Therefore, advancing trustworthy AI requires not only scrutinizing model* *architectures and training data, but also **bringing rigor and interpretability to the optimization algorithms themselves.***
>
> *In this work, we take a step in that direction. We present a learning rate scheduler that is grounded in a principled theoretical* *framework. Our approach offers provable convergence guarantees, closing the gap between practical performance and theoretical* *soundness. By designing algorithms with transparent dynamics and analyzable behavior, we aim to bridge the divide between* *empirical success and theoretical robustness, contributing to the broader vision of reliable, trustworthy machine learning.*

---

### Official Review · Reviewer_KReu · 2025-07-02

**Clarity:** 3
**Significance:** 2
**Originality:** 3
**Rating:** 3
**Confidence:** 2

**Summary:**

This work proposes a randomized learning rate scheduler that samples a learning rate uniformly from $[L_\mathrm{min}, L_\mathrm{max}]$ at each step.
The main hypoethesis is that the randomness can help overcoming local minima and sedle points.
The authors provide theoretical guarantees for the convergence of the method on a restricted class of functions and report some empirical results on popular vision datasets and architectures.

**Questions:**

Don't the references cited in lines 60-61 go against the statement of contribution 2 (starting at line 80)?

**Ethical Concerns:**

["NO or VERY MINOR ethics concerns only"]

**Final Justification:**

As extensively discussed, I still lean slightly towards rejection due to an overall lack of full scientific rigour in the text.
Some of issues have been addressed after being discussed explicitely, but the the pervasiveness of such issues in a paper with strong practical claims removes my confidence in recommending acceptance.

My original question was fully answered by the authors (and turned out to be a confusion on my side).

I highlight that, even though I could understand the theoretical side of the work, I lack the enough familiarity with the literature to properly assess its significance.

**Limitations:**

yes

**Paper Formatting Concerns:**

I couldn't find any major formatting issues.

**Quality:**

3

**Strengths And Weaknesses:**

The proposed method is very simple and easy to implement, which is a clear strength.
This is especially valuable given that it depends on only two hyperparameters, $L_\mathrm{min}$ and $L_\mathrm{max}$, not relying on any hard-to-compute parameters, like many previous methods.
Besides the theoretical guarantees, which are clearly the focus of the paper, the authors also provide some empirical validation of the method, which indicates that it works well in practice.
Overall, the main text of the paper is fairly clear and can be followed without much effort.

Moving to more nuanced points, my main issue unfortunately lies with the experimental validation.
I believe those experiments are far from sufficient to support the reasonably strong claims made by the authors.
Unfortunately, understanding how the performance of the method compares with other methods demands some statistical investigation that cannot be done with such small experiments (only 3 seeds, little hyperparameter tuning, a limited set of datasets and architectures).
In particular, the objective claims of better stability and faster convergence could and should be made rigorous (associated with precise metrics) and evaluated statistically.
I doubt it is possible to extract any deep conclusions from those claims from the mere plots provided in the paper.
I do, however, understand how expensive such experiments can be, but I'm afraid this is an unavoidable price to pay if we are to properly support practical promises.

The theoretical work is, therefore, the main contribution of the paper.
The proofs are a computational tour de force.
I have tried to dive into them and some of the literature to come up with a fair assessment of the theoretical contribution, both in terms of novelty and significance.
Alas, I'm afraid I didn't have much success, **which should be noted by the AC**.
While this, of course, reflects some mismatch in expertise, the fact that such evaluation is that hard also says something about the authors' contextualization of their contributions.
In particular, it seemed important to have a good understanding of the significance of an analysis compatible with multiplicative noise, as this appeared to be the main technical addition to the literature.
Accordingly, the authors should consider significantly expanding their discussion of this point since, without proper advertisement, it would be fair to assume that the contribution is not so important.

For a less critical point, arguing against the dependence of previous methods (mainly [16] and [17]) on hard-to-compute parameters is not so immediate as the authors suggest.
The authors would have to argue that those parameters cannot be efficiently estimated by usual tuning methods (perhaps the method is quite robust to the choice of those parameters).
Similarly, arguing for the easy tunability of $L_\mathrm{min}$ and $L_\mathrm{max}$ has to involve some consideration of the robustness of the method to their choice.

Overall, I appreciate the work (mainly for the theoretical side) but I don't feel confident to properly assess its significance and originality (which is reflected in my confidence score).
I'll lean slightly towards rejection due to the concerns raised above.

---

> ### Author Rebuttal · Authors · 2025-07-30
>
> We thank the reviewer for their time and effort in reviewing our paper. We appreciate their thoughtful and constructive comments and agree that one of the key advantages of our saddle-point escape algorithm lies in its reliance on only two hyperparameters for implementation in neural networks.
>
> **Need for Experimental Evaluation**
>
> he reviewer has raised concerns with respect to three major aspects in our experimental evaluation: number of seeds, limited hyperparameter tuning, and limited number of datasets and architectures. We address each of them below.
>
> - Number of seeds: Our experimental setting was developed with reference to our baselines. Our choice for the number of seeds (three) is based on [A]. A cornerstone paper in this field, [B] uses 5 seeds, but only for the implementation of their own algorithm, and not the baselines. We have now implemented 2 additional seeds for our LR scheduler based on your suggestion for one of our architectures owing to limited time. The results over 5 runs are provided below
>
> | Architecture | Dataset  | Max test acc | Average acc (S.D) |
> |--------------|----------|--------------|--------------------|
> | WRN-28-10    | CIFAR-10 | 93.76        | 92.02 (1.24)       |
>
> - Additional datasets and architectures: We have now performed simulations on ImageNet-1K (on ResNet-50 architecture) and BERT-large finetuned on SQuAD v1.1 dataset [C].
> For BERT fine-tuning, we apply the proposed PLRS with the AdamW optimizer, using $L_{\text{min}}$ = 2e-5 and $L_{\text{max}}$ = 3e-5. All other training settings follow [A]. We achieve an F1 score of 87.69, with an average accuracy of 87.55 and a standard deviation of 0.117 across 3 runs, which is comparable to the state-of-the-art F1 score reported (88.66) using the Knee LR scheduler [A].
> We also train on the ImageNet-1K dataset for 80 iterations using the SGD optimizer without momentum or weight decay. With $L_{\text{min}} = 0.05$ and $L_{\text{max}} = 0.11$, and a batch size of 256, we achieve a test accuracy of **68.25**, outperforming the Knee LR scheduler by 1.03 under similar settings.
> This demonstrates that PLRS generalizes well across diverse tasks, including both small- and large-scale image classification, as well as transformer-based text models.
>
> - Hyperparameter tuning: Our initial intention was to highlight that the proposed scheduler works well even with minimal tuning. Now, in response to the reviewer’s comments, we have performed a hyperparameter sweep using the wandb library on a range of values for $L_{\text{min}}$ (0.01, 0.03, 0.05, 0.07, 0.09) and $L_{\text{max}}$ (0.1, 0.2, 0.3, 0.4, 0.5) for WRN-28-10 with CIFAR-10 dataset. We achieve a maximum test accuracy of 94 for $L_{\text{min}} = 0.09$ and $L_{\text{max}} = 0.1$, which is higher than our previously mentioned result and also higher in comparison to all other learning rate schedulers, as is evident from Table 1 of our paper.
>
> **Quantifying Faster and More Stable Convergence**
>
> We would like to point out that our main claim is that PLRS can escape saddle points with provable convergence. We do not claim that it is universally faster or more stable than other schedulers theoretically. Having said that, we do consistently **observe** faster and stable convergence across multiple datasets and architectures in our **empirical evaluation**, and we note that in our discussion of results. Such an observation regarding the convergence of the algorithm using the training loss curves is in line with literature [D,E,F].
>
> **Expanding Discussion on Multiplicative Noise**
>
> We would like to draw attention to the final paragraph preceding Section 1.1 of our paper where we briefly discuss the significance of multiplicative noise. We further elaborate on the characteristics of the said multiplicative noise based on the reviewer’s suggestion.
>
> In the small-gradient regime, such as near a local minimum or a saddle point, the intuition is that occasional sampling of a high learning rate (close to $L_{\text{max}}$) causes the stochastic gradient to be scaled significantly, enabling the iterate to move away from the saddle point. Conversely, when the learning rate sampled from the uniform distribution $U[L_{\text{min}}, L_{\text{max}}]$ is low, the iterate may undergo only minor changes. However, due to the randomness in the learning rate, repeated application of this multiplicative noise results, with high probability, in the iterate escaping the saddle point. By bringing in multiplicative noise using a probabilistic LR scheduler, in this work we analytically show that under certain assumptions, the iterate escapes a saddle point region with high probability.
>
> We will include this discussion of multiplicative noise in the final version of the paper.
>
> **On the Tunability of Hard-to-Compute Parameters in Existing Literature**
>
> In [16], setting the learning rate as per their algorithm requires knowledge of the parameters gradient Lipschitz constant, Hessian Lipschitz constant, and \epsilon to converge to an \epsilon second order stationary point (see equation (4) of [16]). While \epsilon is a tunable hyperparameter, we would like to point out that the other parameters (gradient and Hessian Lipschitz constants) **are typically meant to be theoretically computed, and not tuned.** While tuning is possible in practice, doing so deviates from the original algorithmic intent proposed by the authors and can undermine theoretical guarantees. Additionally,  it can be intractable to  compute the gradient Lipschitz and Hessian Lipschitz constants, especially for neural networks owing to their complicated optimization landscapes [G].
>
> **References in Lines 60–61**
>
> The references in lines 60–61 (references [2, 9, 16, 28, 33]) do not contradict, but in fact, support our contribution 2. All of the references in lines 60–61 cannot be implemented for neural networks since they require computation of various constants such as gradient and Hessian Lipschitz constants, which are typically mathematically intractable to compute for deep neural networks. Therefore, all these works have focussed exclusively on non-neural network settings in their empirical evaluation.
> Since we show that one can use multiplicative noise to escape saddle points, we can cast our solution as a probabilistic learning rate scheduler and hence, leverage the LR range test used extensively by deterministic learning rate schedulers for deep neural networks to compute the parameters we require, namely, $L_{\text{min}}$ and $L_{\text{max}}$.
>
> **References**
>
> [A] Iyer, Nikhil, et al. "Wide-minima density hypothesis and the explore-exploit learning rate schedule." Journal of Machine Learning Research 24.65 (2023): 1-37.
>
> [B] Smith, Leslie N. "Cyclical learning rates for training neural networks." 2017 IEEE Winter Conference on Applications of Computer Vision (WACV). IEEE, 2017.
>
> [C] Rajpurkar, Pranav, et al. "Squad: 100,000+ questions for machine comprehension of text." arXiv preprint arXiv:1606.05250 (2016).
>
> [D] Loshchilov, Ilya, and Frank Hutter. "SGDR: Stochastic Gradient Descent with Warm Restarts." International Conference on Learning Representations. 2017.
>
> [E] Vaswani, Sharan, Francis Bach, and Mark Schmidt. "Fast and faster convergence of SGD for over-parameterized models and an accelerated perceptron." The 22nd International Conference on Artificial Intelligence and Statistics. PMLR, 2019.
>
> [F] Qin, Tiancheng, S. Rasoul Etesami, and Cesár A. Uribe. "Faster Convergence of Local SGD for Over-Parameterized Models." Transactions on Machine Learning Research 2024 (2024).
>
> [G] Virmaux, Aladin, and Kevin Scaman. "Lipschitz regularity of deep neural networks: analysis and efficient estimation." Advances in Neural Information Processing Systems 31 (2018).

---

> > ### Comment · Reviewer_KReu · 2025-08-04
> >
> > I thank the authors for their detailed reply.
> > I felt that some aspects of my review might not have been fully clear, so I'll expand on some of the points to try to get us on the same track.
> >
> > **Need for Experimental Evaluation**
> >
> > I'll once more highlight that I recognise and respect the theoretical focus of the paper, as I acknowledge that the final paragraphs of my review might have given the opposite impression.
> > Nonetheless, the expanded experiments are a nice addition to the work.
> >
> > Turning to the *hyperparameter tuning* subsection, it seems to contain a good illustration of the type of approach that is concerning me here.
> > When the authors say
> > > Our initial intention was to highlight that the proposed scheduler works well even with minimal tuning...
> >
> > A scientific way to do so would be to make the concept rigorous and design experiments to statistically evaluate it.
> > A simple example would be to take note of the accuracies obtained for the different combinations of parameters ($L\_\mathrm{min}$, $L\_\mathrm{max}$) and show a statistical study of the hypothesis.
> > For a simplified example, showing that with high confidence the difference between the maximum and minimum accuracy encountered is very small or that it is always above the performance of competing methods would be a scientific conclusion indicating that "the proposed scheduler works well even with minimal tuning".
> > This approach also has the big advantage of generating more data that might also reveal interesting behaviours (things could go against our hypotheses).
> > For example, in these experiments, the best accuracy was found with the "least random" parameter choice (when $L\_\mathrm{min}$ and $L\_\mathrm{max}$ are the closest), which is precisely when the proposed method most resembles a simple deterministic choice of LR. (But don't worry, I'm regarding this as an artefact of the *relatively* simple loss landscape in this experiment)
> >
> >
> > **Quantifying Faster and More Stable Convergence**
> >
> > Here we see the phenomenon I'm trying to highlight.
> > It's not totally clear what the authors mean when they say that they
> > > consistently observe faster and stable convergence
> >
> > Without a detailed (ideally, quantitative) definition of "stable convergence", it seems to me that the authors simply look at the accuracy plots and infer from eyeballing that the curve seems "less noisy".
> >
> >
> > **References in Lines 60–61**
> >
> > It seems there is some misunderstanding here.
> > Consider the sentence (line 60)
> > > ...several works prove the convergence of noisy stochastic gradient descent in the additive noise setting [33, 16, 2, 28].
> >
> > If it holds true, then the following should be false (line 81)
> > > To the best of our knowledge, we are the first to theoretically prove the convergence of SGD with an LR scheduler that does not conform to constant or monotonically decreasing rate.
> >
> > The conclusion should be valid as long as the "noise" mentioned in the first quote is enough to make SGD "not conform to constant or monotonically decreasing rate" for at least one of the references.
> >
> > From the authors' reply, it seems that some further qualification is needed in the claim of contribution 2.
> >
> > ---
> >
> > Overall, I respect and appreciate the mathematical side of the work and have little to no issue with it.
> > However, I do have a problem with the scientific side of the work.
> > This, in part, comes from the small improvements of the method, but mostly from the inherent difficulty of empirically evaluating claims about such generic tools as LR schedulers.
> > This combination calls for excellent experimental design and statistical rigour to support any practical claims, which I don't see in this work.
> > Thus, I don't feel comfortable recommending acceptance.
> > Still, I wouldn't strongly oppose it (also, notice my low confidence score).

---

> ### Author Response · Authors · 2025-08-05
>
> We appreciate the reviewer’s commendation of the theoretical aspects of our work and the experimental additions made during the rebuttal. We take the opportunity given by the reviewer to provide further clarification.
>
> **Hyperparameter tuning**: We conducted precisely the type of scientific study suggested by the reviewer. During the hyperparameter sweep, we conducted a sensitivity analysis for one of our configurations in response to Reviewer YdRg. We provide the details here and would like to add it to the paper.
>
> On WRN-28-10 architecture using CIFAR-10 dataset, for combinations of the following values for L_min​ and L_max​ - (0.01, 0.03, 0.05, 0.07, 0.09) and (0.1, 0.2, 0.3, 0.4, 0.5), respectively, we observe average test accuracy of 93.42, with a standard deviation of 0.47 and an interquartile range of 0.385. The lower value of standard deviation and interquartile range suggests that the accuracies are not spread out for this range of L_min and L_max values, thereby demonstrating robustness of the algorithm to the choice of these parameters.
>
> **Faster and stable convergence**: We respectfully argue that visually inspecting training loss (or accuracy) plots is a standard widely accepted method to claim faster convergence in this area. We support our argument with the prominent references below:
>
> * In [A], the authors make a similar claim in the first paragraph of Section 6.1 simply based on studying their training loss curves. We quote: “According to Figure 1, we found that Adam yields similar convergence as SGD with momentum and both converge faster than Adagrad.”
> * In [B], the authors claim that they are able to train their network faster with the proposed one-cycle LR scheduler using the test accuracy plots given in Fig. 6. Quoting from the caption of Fig. 6, “Training ResNet and inception architectures on the ImageNet dataset with the standard learning rate policy versus a 1cycle policy that displays super-convergence; illustrates that deep neural networks can be trained much faster (20 versus 100 epochs) than by using standard training methods”.
>
> By “stable convergence,” we refer to the intuitive, non-technical notion of stability; specifically, a consistent decrease in training loss with fewer spikes or oscillations, rather than formal mathematical stability. We agree that this could be better clarified in the text and would be happy to either (i) explicitly define this usage in the paper, or (ii) replace the three instances of “stable convergence” with “smoother loss curves”.
>
> **References in Lines 60–61**: We would like to make the important distinction that while these references prove convergence of noisy SGD algorithms, where noise is added to the gradient, we analyse the LR scheduler paradigm, where noise is added to the learning rate. Elaborating,
>
> (a) In additive noise mechanisms (the references in lines 60-61), the gradient is perturbed by an i.i.d noise vector which is independent of the gradient, and then a learning rate (which is typically either a constant or assumed to be monotonically decreasing with iterations) is multiplied to the perturbed gradient. If $x_t$ is the SGD training iterate at time step $t$, $g(x_t)$ is the stochastic gradient, the noisy SGD update equation is,
> $$x_{t+1}=x_t - \eta (g(x_t)+n)= x_t - \eta g(x_t) - n \eta,$$
> where $n$, the noise vector is sampled from a known distribution such as standard normal or unit sphere and $\eta$ is the learning rate.
>
> (b) In the multiplicative noise setting (our work), we do not perturb the stochastic gradient by  i.i.d noise, but rather, multiply it with a probabilistic learning rate. The SGD update equation in this case is given by (from equation (3) of our paper),
> $$x_{t+1}=x_t - (\eta+u_{t+1}) g(x_t)= x_t - \eta g(x_t) - u_{t+1}g(x_t),$$
> where $u_{t+1}$ is sampled from a uniform distribution between L_min and L_max and $\eta$ is a constant.
>
> For (a) to be equivalent to (b), the $n \eta$ term in the equation of (a)  has to have a component which is of the form of the stochastic gradient multiplied by a random variable which violates the assumption that $n$ is independent of $g(x_t)$. Further, the noise in (b) stems from the randomness due to the learning rate (which is multiplicative by nature) while in (a) it is due to the additive component $n$ which cannot alter the learning rate in any way.
>
> In summary, the literature on convergence analysis of additive noise does not subsume the non-monotonically decreasing learning rate paradigm which we put forth in our work. We hope that this discussion helps clarify the merit of the claim in contribution 2.
>
> [A] Kingma, Diederik P. and Jimmy Ba. “Adam: A Method for Stochastic Optimization.” arXiv preprint arXiv:1412.6980 1412.6 (2014).
>
> [B] Smith, Leslie N., and Nicholay Topin. "Super-convergence: Very fast training of neural networks using large learning rates." Artificial intelligence and machine learning for multi-domain operations applications. Vol. 11006. SPIE, 2019.

---

> > ### Comment · Reviewer_KReu · 2025-08-06
> >
> > (I apologise for sounding somewhat adversarial here. I don't mean it, but no phrasing I tried removed this tone.)
> >
> > Thanks for the clarifications.
> >
> > **Hyperparameter tuning:**
> > I apologise for missing that part of the reply.
> > I actually read it, but with the many simultaneous discussions, it escaped my mind by the time I was writing the reply.
> > Nonetheless, that was an illustration of the principle that worried me in the text. I was alluding to the fact that providing some scientific substation for claims does not seem like a "natural reflex" in the text, meaning that even claims that could be addressed easily are given not even a minimal scientific treatment.
> >
> > **Faster and stable convergence:**
> > I do believe faster convergence can be assessed via direct inspection (e.g., once reduced to the comparison of two values).
> > It was for this reason that the wording in my comment was crafted to avoid implying otherwise.
> > Nonetheless, this instance might be a good opportunity to highlight that the appeal to standard practice is never a solid argument.
> > If the standard is good, then there is an underlying explanation that ensures its correctness, and it's this explanation that should be provided as the argument.
> > If the standard is bad, appealing to it only perpetuates the problem.
> > This issue can poison an entire field.
> >
> > Still, I'm glad that the authors acknowledge the issue with the use of "eyeballing" to infer stability of convergence and are willing to take steps to mitigate it.
> > Moreover, the offer of option (i) (to mention explicitly that this is an informal, non-reproducible heuristic) illustrates that the authors grasped my point quite well: it would be fine if the authors added anything to overwrite the "this is a serious scientific claim" that implicitly precedes **every sentence** in a scientific paper.
> >
> > **References in Lines 60–61**: I now understand it. Thank you. The authors should consider revising the phrasing around that. I might have been entirely on my side, but it seems that the chance of confusion could be significantly reduced with a few extra words.
> >
> > ---
> > I again apologise if the tone came off as adversarial in any way.
> >
> > Regardless, I maintain my position:
> > *Mathematics is not a science.
> > It appears that this paper was written as a mathematics one, with experiments being added as a secondary matter (which is common and fine).
> > However, the existence of empirical evaluations claims for scientific rigour, but, in this work, much of it only seems to be observed by the authors in an ad hoc fashion, making me lean **ever so slightly** towards rejection.*
> >
> > To change my assessment would take going through the entire text, aiming to deliver the entire empirical side of the paper with absolute scientific rigour.
> > I hope the authors will see that this is not an unfair demand, as it seems that they try to deliver the mathematical side of the work with absolute logical rigour (also given the prestige of this venue).
> > I understand that giving the full work (and experiments) a pass to deliver full scientific rigour could mean a significant amount of work, and that it might not be worth it for the authors since my position shouldn't impact the final decision in any meaningful way.

---

### Official Review · Reviewer_YDrg · 2025-07-02

**Clarity:** 3
**Significance:** 2
**Originality:** 3
**Rating:** 4
**Confidence:** 3

**Summary:**

The authors propose a probabilistic learning-rate scheduler in which the step size is drawn independently and identically distributed from a uniform distribution at every iteration. The idea is that if you run stochastic gradient descent but choose the step size at each iteration by randomly sampling it from a fixed interval with a lower and upper bound, the resulting algorithm still converges. The paper presents experiments with the proposed approach on several CNN architectures (VGG-16, WRN-28-10, ResNet-110, DenseNet-40-12, ResNet-50) and image-classification datasets (CIFAR-10, CIFAR-100, Tiny-ImageNet). The obtained accuracy is close to or slightly better than popular deterministic schedules such as cosine annealing, one-cycle, knee, multi-step, and constant learning rates. They also show training-loss curves that show convergence more smoothly than traditional schedules. Supplemental material also presents proofs and justification for the chosen hyperparameters.

**Questions:**

Since multiplicative noise can alter implicit regularisation, is it possible that PLRS might converge to sharper minima on some tasks?

If I understand correctly, the range test adds an extra pass over the data and still leaves two hyperparameters to sweep. How robust is PLRS to mis-specification of these bounds?

**Ethical Concerns:**

["NO or VERY MINOR ethics concerns only"]

**Final Justification:**

After rebuttal from the authors, I increased my evaluation.

**Limitations:**

One point I would like to consider is that the proof assumes vanilla SGD, but the real-world training we are dealing with now almost always uses momentum or Adam. I believe the paper lacks both theory and experiments in that setting. The authors should include adaptive optimisers, such as AdamW, etc., because I think users may prefer adaptive methods unless PLRS shows clear advantages.

I also suggest some sensitivity analysis 𝐿_min and 𝐿_max, as they are usually selected via a range test.

I don't get why authors believe uniform sampling is sufficient and do not test other distributions that might yield better exploration–convergence trade-offs. I think there is potential for performance improvement.

**Quality:**

3

**Strengths And Weaknesses:**

I liked the simplicity of requiring only two hyperparameters (𝐿_min, 𝐿_max) and it is possible the proposed approach can be dropped into any SGD code-base without architectural changes or extra forward passes.

The authors suggest it bridges a long-standing theory–practice gap, the formal convergence proof for a non-monotonic learning rate schedule in non-convex SGD. They also show that multiplicative (step-size) noise can be analysed under standard β-smooth/ρ-Hessian-Lipschitz and strict-saddle assumptions.

Besides the potential applicability, the experiments are limited as they only consider image datasets with images of small dimensions. I was expecting some experiments on large-scale datasets such as ImageNet-1K, or maybe NLP or transformer. I am not convinced about the generalisation to modern training regimes.

I found the proposal very interesting, but the performance is modest. The improvements over classical approaches, such as cosine/knee are minimal and disappear in some experiments. More comments on the benefits of the added tuning cost are necessary.

---

> ### Author Rebuttal · Authors · 2025-07-30
>
> We thank the reviewer for their thorough and thoughtful review. Their insightful questions and comments have helped us clarify key aspects of our work and improve the overall quality of the paper.
>
> **Experiments on large-scale datasets:**
>
> We thank the reviewer for their comments and agree on the simplicity of porting our probabilistic learning rate scheduler in any architecture. In response to their comments on training large-scale datasets and using modern architectures, we have now performed experiments on both ImageNet-1K (on ResNet-50) and BERT fine tuning on SQuAD V1.1 dataset. On training ImageNet-1K dataset for 80 epochs with batch size of 256, $L_{\text{min}}$ and $L_{\text{max}}$ values of 0.05 and 0.11 respectively, we obtain a top-1 test accuracy of 68.25% and top-5 accuracy of 88.11% using the SGD optimizer without momentum and weight decay.
> We fine-tuned BERT\-large with base parameters set similar to [A], $L_{\text{min}}$ and $L_{\text{max}}$ values of 2e-5 and 3e-5 respectively. We obtain an F1 macro of 87.69 which is comparable to the F1 macro scores reported in [A]. We would also like to point out to the Reviewer and Area chair that these results were obtained with the available time and computational resources. With more tuning, the results may improve further.
>
> **On performance improvement compared to baselines:**
>
> Our focus in this work was to provide provable convergence guarantees for a learning rate scheduler which could also be ported into modern neural network architectures seamlessly while being amenable to theoretical analysis due to its probabilistic nature
> We have now conducted a hyper parameter sweep of $L_{\text{min}}$ and $L_{\text{max}}$ for the WRN-28-10 architecture trained on CIFAR-10 dataset, and got a maximum test accuracy of 94 for $L_{\text{min}}$ and $L_{\text{max}}$ values of 0.09 and 0.1 respectively, which is an improvement of 1.96 from the result of state-of-the-art Knee LR scheduler (refer Table 1 of our paper). For further details regarding the hyper parameter tuning please refer to our response to Reviewer KReu.
>
> **Cost of tuning:**
>
> There is no extra tuning cost in comparison to state-of-the-art deterministic LR schedulers since all LR schedulers such as cosine, knee, cyclic, require an LR range test to set the parameters. Specifically, cosine LR scheduler requires the parameters minimum learning rate, frequency of restarts and a multiplicative factor; cyclic LR scheduler requires a base learning rate, maximum learning rate, mode of operation and the number of iterations to reach the maximum learning rate; knee LR scheduler requires the peak learning rate, number of explore iterations and the number of warmup iterations. In comparison, for our proposed probabilistic learning rate scheduler, we only require $L_{\text{min}}$ and $L_{\text{max}}$.
>
> **Possible convergence to sharper minimum:**
>
> We really appreciate the reviewer’s insights on multiplicative noise converging to a sharper minima at times. We also observed this effect when computing the Keskar sharpness metric (as referenced in paper [D] and used in [A]). In our experiments, the PLRS approach led to a marginally higher Keskar sharpness value compared to the knee learning‑rate scheduler. A higher value of the Keskar metric generally indicates convergence to a sharper minimum, as highlighted in [D]. We plan to study this phenomenon in greater depth in our future research.
>
> **Clarification regarding the range test:**
>
> We set both our hyper parameters $L_{\text{min}}$ and $L_{\text{max}}$ using the LR range test for the combination of architecture and dataset being used. In the LR range test, we linearly increase the learning rate up to a maximum value, say 0.5 and observe the training loss across the epochs. The training loss curve will show a decreasing trend initially following which there will be a sudden/sharp increase. We note down the learning rate at which it increases and set the value of $L_{\text{max}}$ slightly below it, while the training loss is still in descent and then tune $L_{\text{min}} < L_{\text{max}}$. As stated previously, we do not require any additional tuning other than what existing LR schedulers require. The LR range test was first proposed by the landmark cyclic LR paper [B] and later adopted by one-cyclic LR [C] as well as the Knee LR schedulers [A] subsequently.
>
> We would also like to point the reviewer to the second paragraph of Section 5 (Empirical evaluation) where we already have a condensed version of the above explanation regarding the selection of $L_{\text{min}}$ and $L_{\text{max}}$ values.
>
> **Use of adaptive optimizers:**
>
> The theoretical analysis of SGD with an LR scheduler was quite mathematically challenging due to the multiplicative noise component introduced by the scheduler. Extending these results to SGD with momentum hence adds another layer of complexity. Hence, in order to align with the theory, we have provided empirical results for SGD without momentum. However, as stated in the footnote 2 (on Page 6), we do have better empirical results on SGD with momentum compared with other LR schedulers. For example with $L_{\text{min}}$ and $L_{\text{max}}$ values of 0.07 and 0.1 respectively, we obtain a maximum test accuracy of 90.34 with momentum 0.6 in VGG-16 on the CIFAR-10 dataset, while we obtain 89.68 with the knee LR scheduler with similar settings. We also train BERT-large on SQuAD V1.1 dataset with the AdamW optimizer with our proposed LR scheduler and obtain an F1 score of 87.69 which is on par with the baseline, knee LR scheduler in the same setting.
>
> **Sensitivity analysis:**
>
> In order to determine how sensitive the maximum test accuracy is to the choice of $L_{\text{min}}$ and $L_{\text{max}}$, we conducted a hyper parameter sweep (as in [A]) across a range of values for $L{\text{min}}$ (0.01, 0.03, 0.05, 0.07, 0.09) and $L{\text{max}}$ (0.1, 0.2, 0.3, 0.4, 0.5) for WRN-28-10 on the CIFAR-10 dataset. The average test accuracy was 93.42 with a standard deviation of 0.47 and an inter-quartile range of 0.385, indicating that the values are not spread out. The maximum test accuracy is relatively insensitive to $L_{\text{min}}$ and $L_{\text{max}}$ and tuning them, while recommended, may not be critical.
>
> **Choice of uniform sampling:**
>
> Our choice of uniform sampling was motivated by the feasibility of analysis, as the predominant aim of our work was to provide theoretical convergence guarantees. We agree with the reviewer that sampling from other (bounded) distributions may facilitate better exploration-convergence trade-offs. However providing theoretical guarantees for the same will not be a trivial extension of this work. We believe this could be an excellent future direction for both theoretically and empirically This can be pursued as a future direction through empirical validation.
>
> **References**
>
> [A]  Iyer, Nikhil, et al. "Wide-minima density hypothesis and the explore-exploit learning rate schedule." Journal of Machine Learning Research 24.65 (2023): 1-37.
> [B] Smith, Leslie N. "Cyclical learning rates for training neural networks." 2017 IEEE winter conference on applications of computer vision (WACV). IEEE, 2017.
> [C] Smith, Leslie N., and Nicholay Topin. "Super-convergence: Very fast training of neural networks using large learning rates." Artificial intelligence and machine learning for multi-domain operations applications. Vol. 11006. SPIE, 2019.
> [D] Keskar, Nitish Shirish, et al. "On large-batch training for deep learning: Generalization gap and sharp minima." arXiv preprint arXiv:1609.04836 (2016).

---

> > ### Comment · Reviewer_YDrg · 2025-08-03
> >
> > Thanks to the authors for the detailed rebuttal and clarifications. This is one of those papers with interesting theoretical contributions and results following the right direction, but the experimental evidence still does not reach confirmation.
> >
> > I don't think the results can support the SOTA or "better stability" claims yet. Looking at the results, the gains are small and sometimes absent. I miss a standard large-scale experimentation setup, such as ImageNet-1k with momentum/WD or transformers with a widely used configuration. What is the sensitivity across multiple datasets and architectures? robustness to mis-specification?
> >
> > I will increase my assessment and wait for the other reviewers' vision of the paper.

---

> > > ### Author Response · Authors · 2025-08-04
> > >
> > > We sincerely thank the reviewer for their thoughtful feedback and for increasing their assessment.
> > >
> > > Regarding their question about sensitivity analysis, we performed a hyperparameter sweep for one of our configurations during the rebuttal period due to time constraints, where we included a sensitivity analysis showing robustness across a range of $L_{min}$​ and $L_{max}$​. While we agree broader evaluations would be valuable, we prioritized depth over breadth within the available time.
> > >
> > > Our ImageNet-1K experiments were conducted without momentum to align with our theoretical setup. In this configuration, our method outperforms Knee Scheduler [A]  by 1.03% accuracy.
> > >
> > > For experiments with momentum as well as transformer-based fine-tuning, we included BERT fine-tuning on SQuAD v1.1 with standard AdamW settings. We apologize for not explicitly referencing the optimizer configuration in the main rebuttal. For BERT fine-tuning on the SQuAD v1.1 dataset, we used the following base optimizer parameters, consistent with [A]: $\beta_1$ and $\beta_2$ were set to 0.9 and 0.999, respectively (corresponding to momentum parameters). The model is implemented using HuggingFace Transformers version 4.55.0.dev0.
> > >
> > > We appreciate the reviewer’s constructive suggestions and hope the current results help demonstrate the promise of our approach.

---

### Official Review · Reviewer_Smwc · 2025-07-03

**Clarity:** 3
**Significance:** 3
**Originality:** 3
**Rating:** 4
**Confidence:** 1

**Summary:**

The paper proposes a new probabilistic scheduler that achieves state-of-the-art performance and has provable convergence guarantees.

**Questions:**

NA.

**Ethical Concerns:**

["NO or VERY MINOR ethics concerns only"]

**Final Justification:**

I am satisfied with the author's response, and I maintain my already positive score. The authors addressed my concerns mentioned in the review. As mentioned by other reviewers, the experimental evidence (which I tried to follow up with the authors during the rebuttal period) is reasonable, if not groundbreaking. Hence, I think "Borderline accept" is the right rating (my "final updated rating") for this paper, mirroring other reviewers' opinions too.

**Limitations:**

Yes.

**Quality:**

3

**Strengths And Weaknesses:**

Strengths:

1. Solid work showing an efficient and effective scheduler. This work especially tackles multiplicative noise appearing during training.
2. This is also the first work showing provable guarantees for SGD with an LR scheduler. They have a good coverage of theory that shows provable guarantees on the scheduler.
3. Focusing more on the experimental part of the paper, I think the paper applies their LR on a decent spread of datasets/tasks. Results are good enough on CIFAR-100 and Tiny ImageNet datasets. It would have been interesting to see results on a language modeling task, too.
4. Given that the proposer LR has provable guarantees and in practice it shows better or comparable performance with other schedulers, I think there is potential for this to be adopted over other heuristic-based schedulers. This is why I am leaning toward accepting.

Weaknesses:

1. It would have been nice to show the scheduler's effectiveness on larger tasks. Like I mentioned, they could have tried a language modeling task with a potentially larger corpus (e.g., enwiki8) and model (character transformer).
2. Indeed, as mentioned in the limitation, it could come in a big way when this LR is used in larger networks with a complex loss function. I am curious, do you think the assumptions will hold for a language modeling loss?

---

> ### Author Rebuttal · Authors · 2025-07-30
>
> We thank the reviewer for their feedback, which motivated us to further evaluate the effectiveness of our proposed method on larger tasks, particularly in the language modeling domain.
>
> **Effectiveness on larger tasks**
>
> In response to the reviewer’s comment, we have now implemented our proposed LR scheduler for finetuning BERT-large on the SQuAD v1.1 dataset with the AdamW optimizer. With all base parameters set similar to [A], including the number of steps, we obtain an F1 score of **87.69** with $L_{\text{min}}$ = 2e-5 and  $L_{\text{max}}$ = 3e-5. Compared with the F1 score obtained by [A], we get competitive performance with very minimal tuning.
>
> **Validity of assumptions on language modeling tasks**
>
> While our theoretical guarantees rely on certain assumptions about the optimization function, we believe that the algorithm itself has wide applicability, as is observed with other LR schedulers and optimization algorithms. Our observed performance on the experiment described above (BERT) is another testament to its applicability.
>
> [A]  Iyer, Nikhil, et al. "Wide-minima density hypothesis and the explore-exploit learning rate schedule." Journal of Machine Learning Research 24.65 (2023): 1-37.

---

> > ### Comment · Area_Chair_2Hhs · 2025-08-08
> >
> > Dear Reviewer Smwc,
> >
> > As the author-reviewer discussion period is approaching its end, please review the rebuttal and engage in the discussion promptly. A note confirming your concerns are resolved is critical. Also, non-participating reviewers may face penalties under the Responsible Reviewing Initiative, affecting future invitations.
> >
> > Thanks,
> >
> > AC

---

### Comment · Area_Chair_2Hhs · 2025-08-04
**Engage in Author-Reviewer Discussions**

Dear reviewers,

If you haven't done so already, please click the 'Mandatory Acknowledgement' button and actively participate in the rebuttal discussion with the authors after carefully reading all other reviews and the author responses.

Thanks,
AC

---

### Note · Authors · 2025-08-13

Advancing trustworthy AI, which is aligned with the principles of NeurIPS, requires not only scrutinizing models and training data, but also bringing interpretability to the optimization algorithms themselves. Acting as a meaningful step in that direction, our work provides the first provable convergence result for non-monotone learning rates. We do so by proposing a probabilistic learning rate scheduler and hence leveraging randomness to provide rigorous theoretical guarantees. Based on the reviews and the following discussion, we believe that all the reviewers agree with the significance, impact, and rigor of our theoretical contributions, which is the major focus of our work.

We further applied our proposed scheduler to a variety of datasets and architectures, and showed that a work grounded in theory can perform well in practice. Almost all of the reviewers’ comments and suggestions were focused on empirical verification; we highlight the important ones here. Reviewer YDrg and KReu suggested including large datasets, for which we implemented ImageNet-1K in ResNet-50 and observed better performance than a state-of-the-art learning rate scheduler. In response to Reviewers Smwc, YDrg, and KReu’s comment about including a language model, we performed experiments on BERT fine-tuning on the SQuAD v1.1 dataset and noticed comparable performance to a state-of-the-art learning rate scheduler. Another important common question was regarding the sensitivity to hyperparameters, raised by Reviewers YDrg and KReu, for which we performed a hyperparameter sweep and corresponding sensitivity analysis and noted that our method does not rely heavily on the choice of hyperparameters. We believe that addressing these concerns has significantly improved our experimental validation. Further, we have also addressed Reviewer 69TR’s queries regarding our proof mechanism using martingale theory. We have also elaborated on the multiplicative noise mechanism we have adopted in response to Reviewers 69TR and KReu's queries.

We further would like to note that we have addressed every comment raised by the reviewers, with most reviewers confirming that our response is up to their satisfaction. With regard to reviewer KReu, we have addressed all of their actionable concerns. We believe that we have provided sufficient empirical evidence demonstrating the practical utility of our theoretical work across multiple architectures and datasets, thus bridging theory and practice.

---

### Decision · Program_Chairs · 2025-09-17

**Decision:**

Reject

**Comment:**

This paper proposes a simple probabilistic learning rate (LR) scheduler with SGD in which the learning rate for each step is sampled from a uniform random variable with two hyperparameters conrresponding to a lower and upper bound. In particular, it is theoretically proved for convergence even without a restriction to constant or monotonically decreasing rates. Empirical results on small image classficiation datasets with somewhat small networks show that the proposed LR schedulaer, PLRS, improves stability and enables faster convergence compared with the existing LR schedulers.

Overall, the theoretical proof of convergence of the proposed LR scheduler is interesting and would be the main contribution this paper. Actually, experimental results show that the proposed LR schedulaer does not demonstrate consistent and significant improvements in terms of both the test accuracy and convergence rate. Moreover, the empirical validation especially from the perspective of the practicality is still limited, even considering the additional results on ImageNet-1K with ResNet-50 and on SQuAD V1.1 with BERT during the rebuttal phase. Regarding the issues on the tuning of the two critical hyperparameters, L_min and L_max, a more thorough (sensitivity) analysis is also necessary particularly by considering not only test accuracies but also convergence rates across diverse models and datasets, since these are basically tuned by a somewhat ad-hoc method with some forward and backward passes.

Therefore, while the theoretical convergence proof is meaningful, the lack of sufficient empirical validation and sensitivity analysis makes it difficult to consider this paper as meeting the acceptance bar of NeurIPS.